# UNDERSTANDING TRANSFORMERS FOR TIME SERIES: RANK STRUCTURE, FLOW-OF-RANKS, AND COMPRESSIBILITY

**Annan Yu,**[1,*]   **Danielle C. Maddix,**[2,†]   **Boran Han,**[2]   **Xiyuan Zhang,**[2]
**Abdul Fatir Ansari,**[2]   **Oleksandr Shchur,**[2]   **Christos Faloutsos,**[3]
**Andrew Gordon Wilson,**[4]   **Michael W. Mahoney,**[4]   **Yuyang Wang**[2]
[1] Center for Applied Mathematics, Cornell University
[2] Amazon Web Services   [3] Amazon Selling Partner Services
[4] Amazon Supply Chain Optimization Technologies

## ABSTRACT

Transformers are widely used across data modalities, and yet the principles distilled from text models often transfer imperfectly to models trained to other modalities. In this paper, we analyze Transformers through the lens of rank structure. Our focus is on the time series setting, where the structural properties of the data differ remarkably from those of text or vision. We show that time-series embeddings, unlike text or vision, exhibit sharply decaying singular value spectra: small patch sizes and smooth continuous mappings concentrate the data into low-rank subspaces. From this, we prove that the associated $Q/K/V$ projections admit accurate low-rank approximations, and that attention layers become compressible in proportion to the decay of the embedding spectrum. We introduce the concept of *flow-of-ranks*, a phenomenon by which nonlinear mixing across depth inflates the rank, explaining why early layers are most amenable to compression and why ranks grow with depth. Guided by these theoretical and empirical results, we use these insights to compress Chronos, a large time series foundation model, achieving a reduction of $65\%$ in inference time and $81\%$ in memory, without loss of accuracy. Our findings provide principled guidance for allocating width, depth, and heads in time series foundation models, and for exploiting their inherent compressibility. Our code is available at https://github.com/amazon-science/tsfm-compression.

## 1 INTRODUCTION

Transformers, originally designed for language (Lewis et al., 2020; Achiam et al., 2023), are now widely deployed, e.g., time series (Ansari et al., 2024a; Das et al., 2024; Shi et al., 2025; Wolff et al., 2025), images (Liu et al., 2021; Dosovitskiy et al., 2020), molecules (Maziarka et al., 2020; Leon et al., 2024), and DNA sequences (Ji et al., 2021; Le et al., 2021; Nguyen et al., 2024). A common approach to apply Transformers to these other data modalities is to directly transfer architectural parameters (e.g., width, heads, depth) from text-based models, on the assumption that what works for text should generalize. However, this assumption is fragile. As an example, we show that time series differ fundamentally from language in how signals are tokenized and embedded. This leads to the more general question: how well do community insights on pretraining and hyperparameter tuning, based largely on Transformers applied to text data, port to Transformers applied to other data modalities? Understanding the answer to this question is particularly important when Transformers are applied in domains where data are less abundant than in the text domain.

Here, we address this question in the context of time-series data and time-series forecasting. A priori, the answer to this question is not obvious: time series data have similarities with text data (e.g., they have obvious sequential properties), but they also have many differences (e.g., it is not clear that they

---

*Work done during an internship at AWS.
†Correspondence to: Danielle C. Maddix <dmmaddix@amazon.com>.

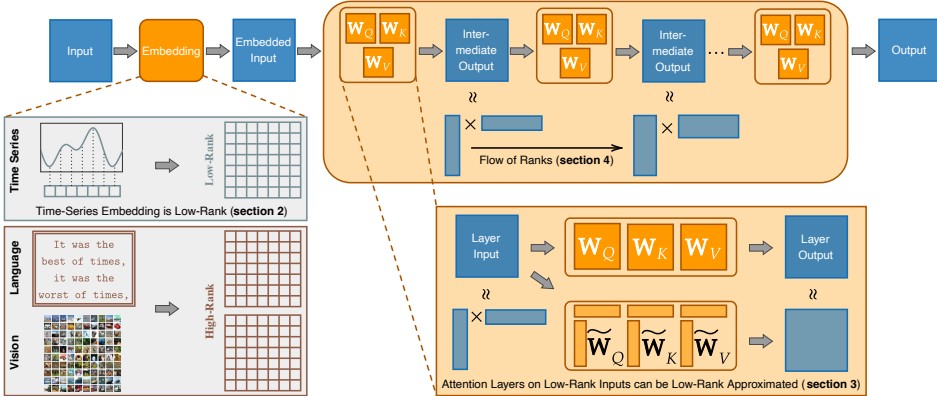

**Figure 1:** Overview of our results. We show that the embedded inputs of Transformers trained with time-series data have much lower ranks than those of other modalities, including Vision Transformers and Transformers trained with language data (see section 2); we prove that attention matrices on low-rank inputs are well-approximated by low-rank matrices (see section 3); and we introduce and demonstrate a concept called flow-of-ranks, describing how attention matrices in earlier layers are more compressible than those in later layers (see section 4).

are well-modelable by a discrete set of tokens). The question, however, is timely: time series data are ubiquitous in many domains, including scientific (Abhishek et al., 2012; Zhang & Gilpin, 2025; Lai et al., 2025), industrial (Hong et al., 2016), and financial applications (Zhang et al., 2001), where they facilitate critical tasks, e.g., forecasting (Hyndman & Athanasopoulos, 2018), imputation (Yoon et al., 2018), and anomaly detection (Blázquez-García et al., 2021). In addition, there has been recent growing interest in developing so-called time series foundation models (TSFMs) (Brown et al., 2020; Radford et al., 2021; Ansari et al., 2024a). These models are large pretrained models, designed with the hope of providing a foundation (Bommasani et al., 2021), to be adaptable to a wide range of domains and time series tasks. As with other models that aim to provide such a "foundation," TSFMs are appealing because they reduce the need for task-specific architectures and parameter computation, thereby enabling the transfer to new settings with relatively little effort.

In this paper, we develop a framework for analyzing design decisions in Transformers; see Figure 1 for an overview and Appendix A for related work. Our view is that these decisions are guided by an understanding of the structural properties of the data modality, which are then inherited by properties of the model. Our framework enables a detailed linear algebraic analysis of the attention layers within a Transformer, which consist of three linear transformations to form the queries, keys, and values. Our approach is general, and we apply it to the time series domain. Lastly, we show that we can use our approach as a practical tool for compressing TSFMs. We summarize the flow of our paper and our main contributions as follows:

1. **Data modality and rank structure.** We compare Transformers trained to different data modalities through the lens of numerical rank, and we demonstrate that time-series data lead to a particularly low-rank structure, at the level of inputs. To the best of our knowledge, our work is the first to directly study which data modality features make Transformer models well-approximated via a truncated singular value decomposition (SVD), and why. We also show how standard time-series embeddings preserve low-rank structure in the hidden space, which differs from large-vocabulary text and other modalities. (See section 2.)

2. **From low-rank inputs to low-rank attention.** Next, we provide the first general theoretical results that connect low-rank embeddings to low-rank attention matrices, and we make clear how width and the number of heads control the quality of low-rank approximations. While these results apply generally to any low-rank embeddings, we illustrate them in the case of time series. In addition, our results are sharp: we show that high-rank embedded inputs, which appear in data modalities such as text and vision, lead to incompressible attention matrices. (See section 3.)

3. **Flow-of-ranks.** We introduce the "flow-of-ranks" concept to describe how the numerical rank changes across layers in a deep Transformer. This extends our earlier analysis from a single attention layer to the setting of a deep Transformer, where nonlinear activations, residual mixing, and normalization gradually increase the rank of a representation. We note that "flow-of-ranks" explains why early layers are often better approximated by SVD than later ones. (See section 4.)

**Table 1:** Comparison of embedding strategies and patch sizes across TSFMs.

| Model | Chronos | WaveToken | TOTEM | Time-MOE | Chronos-Bolt | TimesFM |
|---|---|---|---|---|---|---|
| **Strategy** | Quantization | Quantization | Quantization | Continuous | Continuous | Continuous |
| **Patch Size** | 1 | 1 | 1 | 1 | 16 | 32 |

4. **Compressibility of real-world TSFMs.** Finally, we illustrate one application of our insights: compressing a real-world TSFM. We demonstrate that the same set of hyperparameters (i.e., width, depth, and number of heads) more severely over-parameterizes TSFMs than it does LLMs. This leads us to develop two complementary compression strategies: compressing a pretrained model and pretraining a model that is compressed by design. In particular, we show that by compressing a Chronos model, we can reduce the inference time by $65.4\%$ and memory usage by $81.4\%$, at no cost of predictive performance. Overall, our results demonstrate and explain why state-of-the-art TSFMs are highly compressible, in practice, compared to (language-trained) LLMs of the same size. (See section 5.)

Additional supporting material may be found in the appendices.

## 2 DATA MODALITY AND RANK STRUCTURE OF EMBEDDING

In this section, we investigate the structure of time-series embeddings and compare them to embeddings from other data modalities. Our goal is to understand, from both a theoretical and empirical perspective, why time-series inputs often look low-rank after embedding.

We consider univariate time series. In particular, let $\mathbf{x} = (x_1, \ldots, x_T) \in \mathbb{R}^{1 \times T}$ be a univariate input of length $T$. Note that, unlike other data modalities, the input $\mathbf{x}$ is a rank-1 matrix.[1] The first step in a TSFM is to map $\mathbf{x}$ into a high-dimensional sequence via an embedding function $\mathbf{\Phi} : \mathbb{R}^{1 \times T} \to \mathbb{R}^{d \times L}$, where $d$ denotes the hidden dimension of the model and $L$ denotes the new sequence length, possibly different from $T$ due to patching. This embedding is typically constructed by applying a trainable function $\phi : \mathbb{R}^k \to \mathbb{R}^d$ to disjoint patches of size $k$. Assuming $L = T/k$ is an integer, this gives:

$$\mathbf{\Phi}(\mathbf{x}) = (\phi(x_1, \ldots, x_k), \phi(x_{k+1}, \ldots, x_{2k}), \ldots, \phi(x_{(L-1)k+1}, \ldots, x_{Lk} = x_T)).$$

These patch embedding functions $\phi$ fall into two main categories (see Figure 8 in Appendix B):

- A **quantization-based embedding** partitions the input space $\mathbb{R}^k$ into $V$ disjoint regions, $\mathbb{R}^k = \biguplus_{i=1}^{V} R_i$. Each region $R_i$ is mapped to a unique trainable vector $\mathbf{u}_i \in \mathbb{R}^d$, so that $\phi(\mathbf{x}) = \sum_{i=1}^{V} \mathbb{1}_{\{\mathbf{x} \in R_i\}} \mathbf{u}_i$. This approach is used in several TSFMs, e.g., Chronos (Ansari et al., 2024a), WaveToken (Masserano et al., 2025), and TOTEM (Talukder et al., 2024).

- A **continuous embedding** uses a parameterized function, typically a neural network $\phi(\cdot; \boldsymbol{\theta})$, to map patches directly. This strategy is also used in many TSFMs, e.g., Chronos-Bolt (Ansari et al., 2024b), Moirai (Woo et al., 2024), TimesFM (Das et al., 2024), and Time-MoE (Shi et al., 2025).

Table 1 summarizes the design choices for these prominent TSFMs.

In most TSFMs, the patch size is significantly smaller than the hidden dimension (i.e., $k \ll d$), meaning $\phi$ maps from a low-dimensional space to a higher-dimensional one. Intuitively, if $\phi$ is well-behaved, it should embed the low-dimensional space $\mathbb{R}^k$ into a corresponding low-dimensional submanifold $\phi(\mathbb{R}^k) \subset \mathbb{R}^d$. To formalize this notion of dimensionality, we use singular values. Let $\mathbf{U}$ be a linear operator between some Hilbert spaces and $\sigma_1 \geq \cdots \geq \sigma_n \geq 0$ be its singular values, where $1 \leq n \leq \infty$. While the algebraic rank of an object $\mathbf{U}$, i.e., the number of its non-zero singular values, is a strict measure, real-world data is noisy; therefore, we use the more practical concept of numerical rank. For a tolerance $\varepsilon > 0$, the $\varepsilon$-rank of $\mathbf{U}$ denotes the number of its singular values that are significant relative to the largest one:

$$\text{rank}_{\varepsilon}(\mathbf{U}) = |\{j \mid \sigma_j(\mathbf{U})/\sigma_1(\mathbf{U}) > \varepsilon\}|. \tag{1}$$

---

[1]In the case of a few-variate time series, $\mathbf{x}$ becomes a few-rank matrix, making the analysis in the paper directly generalizable by viewing the number of variates as another patch dimension. We note that in some time-series applications with a large number of input variables, the embedded inputs may not be low-rank, and our analysis may not directly apply in such cases.

A low numerical rank implies $\mathbf{U}$ is well-approximated by an operator $\tilde{\mathbf{U}}$ with $\mathrm{rank}(\tilde{\mathbf{U}}) = \mathrm{rank}_\varepsilon(\mathbf{U})$.

Our central hypothesis is that for a large corpus of input patches $\{\mathbf{x}^{(i)}\}_{i=1}^N$, the resulting embedded matrix, via quantization or a continuous embedding, $\mathbf{U} = \begin{bmatrix} \phi(\mathbf{x}^{(1)}) & \cdots & \phi(\mathbf{x}^{(N)}) \end{bmatrix}$ has a low numerical rank, which is smaller than the ambient dimension $d$. To test this hypothesis, we sample thousands of patches from diverse signals (e.g., sinusoids, exponential functions, and white noise), and we compute the singular value decay of their embeddings. For contrast, we perform the same analysis on a tabular foundation model (TFM) (Mitra) (Zhang & Robinson, 2025) using 1000 synthetic tables, a T5 LLM processing text from Dickens' *A Tale of Two Cities*, and a ViT processing 1000 randomly sampled images from CIFAR-10.

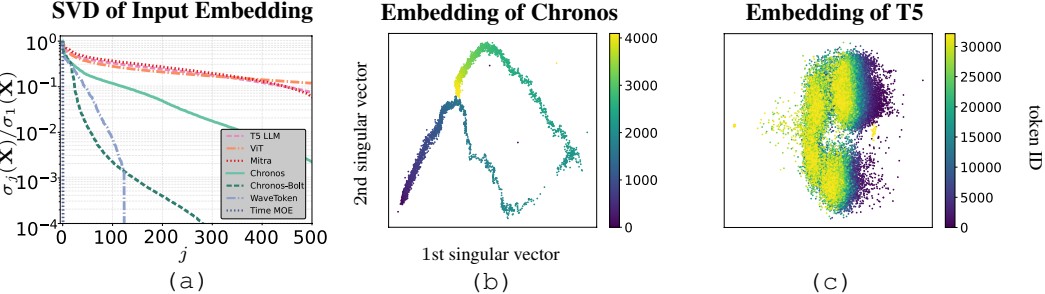

**Figure 2:** (a): Singular values of the embedded input matrices from many different TSFMs, a TFM, a ViT, and an LLM. (b,c): Embedding space of Chronos and a T5 LLM, respectively, visualized by projecting them onto the leading two singular vectors of the embedding matrix.

As shown in Figure 2(a), the singular values of TSFM embeddings decay dramatically faster than those from the tabular and language models, which confirms their significantly lower numerical rank. Figure 2(b,c) visualize the embedding spaces of Chronos (quantized) and T5 (language tokens) by projecting them onto their top two singular vectors. The Chronos embedding, mapping a quantized real line, reveals a clear low-dimensional structure, whereas the T5 embedding, likely due to its vocabulary properties, appears far less well-structured.

One may wonder: why is it that Chronos-Bolt ($k = 16$) produces a lower-rank embedding than Chronos ($k = 1$)? This seeming surprise arises from their different embedding mechanisms. A quantization-based model like Chronos initially maps adjacent values to random, unstructured vectors; it must learn the geometry of the real line during training. In contrast, a continuous embedding like Chronos-Bolt uses a smooth neural network $\phi(\cdot; \boldsymbol{\theta})$. This architectural choice imposes smoothness from the start, ensuring that even a randomly initialized model maps the low-dimensional patch space $\mathbb{R}^k$ to a low-dimensional submanifold in $\mathbb{R}^d$ (see Appendix H).

Although understanding a quantization-based embedding requires looking into the training dynamics, which is beyond the scope of this paper, we theoretically analyze how a continuous embedding preserves low-rank structures. The following theorem formalizes this intuition.

**Theorem 1.** Given any hidden dimension $d > 1$, let $\boldsymbol{\phi} : [-1, 1] \to \mathbb{R}^d$ be a function that embeds $[-1, 1]$ into $\mathbb{R}^d$. Given $L$ arbitrary points $x_1, \ldots, x_L$ sampled from $[-1, 1]$, define

$$\mathbf{\Xi} = \begin{bmatrix} - & \phi_1(x) & - \\ & \vdots & \\ - & \phi_d(x) & - \end{bmatrix} \in \mathbb{R}^{d \times [-1,1]}, \qquad \mathbf{\Psi} = \begin{bmatrix} \phi_1(x_1) & \cdots & \phi_1(x_L) \\ \vdots & \ddots & \vdots \\ \phi_d(x_1) & \cdots & \phi_d(x_L) \end{bmatrix} \in \mathbb{R}^{d \times L}.$$

Let $s_1 \geq \cdots \geq s_d \geq 0$ and $\sigma_1 \geq \cdots \geq \sigma_d \geq 0$ be the singular values of the quasimatrix $\mathbf{\Xi}$ and matrix $\mathbf{\Psi}$, respectively. Then, the following statement holds:

1. If, for some $V > 0$, $\nu \geq 1$, and every $1 \leq i \leq d$, we have $\phi_i$ and its derivative through $\phi_i^{(\nu)}$ are absolutely continuous on $[-1, 1]$ and $\phi_i^{(\nu)}$ is of bounded variation $V$, then we have

$$s_{j+1} \leq \frac{4V\sqrt{d}}{\pi\nu(j-1-\nu)^\nu} = \mathcal{O}(j^{-\nu}\sqrt{d}), \quad \sigma_{j+1} \leq \frac{2V\sqrt{dL}}{\pi\nu(j-1-\nu)^\nu} = \mathcal{O}(j^{-\nu}\sqrt{dL}), \quad \nu+1 < j \leq d-1.$$

2. If, for some $M > 0$ and every $1 \leq i \leq d$, $\phi_i$ has an analytic continuation to the Bernstein ellipse of radius $\rho > 1$ (see Trefethen (2019)), whose infinity norm is no greater than $M$, then we have

$$s_{j+1} \leq \frac{4M\sqrt{d}\rho^{-j+1}}{\rho - 1} = \mathcal{O}(\rho^{-j}\sqrt{d}), \quad \sigma_{j+1} \leq \frac{2M\sqrt{dL}\rho^{-j+1}}{\rho - 1} = \mathcal{O}(\rho^{-j}\sqrt{dL}), \quad 0 \leq j \leq d - 1.$$

We refer interested readers to Townsend & Trefethen (2015) for the precise definition of a quasimatrix (informally, it is a "matrix" in which one of the dimensions is discrete as usual but the other is continuous). Theorem 1 guarantees that for univariate patches ($k = 1$), a smooth embedding function yields singular values with guaranteed decay rates: polynomial decay of order $\nu$ for functions with $\nu$ continuous derivatives, and exponential decay for analytic functions.

See Appendix B for a proof of Theorem 1 using classic univariate polynomial approximation techniques. Using this result, we can directly explain the low-rank structure observed in models like Time-MoE (see Figure 2 and Corollary 2). While multivariate polynomial approximation results enable us to extend Theorem 1, the size of the polynomial basis up to a fixed degree increases exponentially with $k$, which makes it less practically relevant for a larger patch size $k$. Instead, for Chronos-Bolt, where $k = 16$, we seek an ad hoc result that works when the embedding $\phi(\cdot; \boldsymbol{\theta})$ is an MLP (see Appendix C for a proof of Theorem 2 below).

**Theorem 2.** Consider the input embedding defined by a two-layer residual MLP:

$$\boldsymbol{\Phi}(\mathbf{X}) = \mathbf{W}_3\mathbf{X} + \mathbf{W}_2\,\omega(\mathbf{W}_1\mathbf{X}), \quad \mathbf{X} \in \mathbb{R}^{k \times L}, \quad \mathbf{W}_1 \in \mathbb{R}^{d_f \times k}, \quad \mathbf{W}_2 \in \mathbb{R}^{d \times d_f}, \quad \mathbf{W}_3 \in \mathbb{R}^{d \times k},$$

where $k$ denotes the patch size, $L$ denotes the number of patches, $d_f$ denotes the hidden-layer dimension in an MLP, $d > k$ denotes the hidden dimension of the Transformer, and $\omega$ denotes any activation function satisfying that $|\omega(x)| \leq |x|$ for every $x \in \mathbb{R}$. Then, for any $\varepsilon > 0$, we have

$$|\{j \mid \sigma_j(\boldsymbol{\Phi}(\mathbf{X})) > \varepsilon\|\mathbf{W}_2\|_2\|\mathbf{W}_1\mathbf{X}\|_2\}| \leq \min\{d, (1 + \varepsilon^{-2})k\}.$$

Theorem 2 states that the numerical rank of the continuous embedding is bounded by a quantity dependent linearly on the patch size $k$, not the much larger ambient dimension $d$. The term $\|\mathbf{W}_2\|_2\|\mathbf{W}_1\mathbf{X}\|_2$ reflects the natural scaling of $\sigma_1(\boldsymbol{\Phi}(\mathbf{X}))$ in the definition of rank$_\varepsilon$. In practice, we have $k \ll d$, meaning that MLP embeds an input patch into a low-dimensional subspace in $\mathbb{R}^d$.

To illustrate this theorem, we pretrain 6 Chronos-Bolt models of different patch sizes, and we compute the singular values of their embeddings. Figure 3(a) shows that the numerical rank of the embedding increases with the patch size $k$. We also perform an in-depth analysis, where for each pair of embedded inputs $\phi(\mathbf{x}^{(i)})$ and $\phi(\mathbf{x}^{(j)}) \in \mathbb{R}^d$, we compute the angle between them as follows:

$$\theta(\phi(\mathbf{x}^{(i)}), \phi(\mathbf{x}^{(j)})) = |\phi(\mathbf{x}^{(i)})^\top \phi(\mathbf{x}^{(j)})| / (\|\phi(\mathbf{x}^{(i)})\|_2\|\phi(\mathbf{x}^{(j)})\|_2) \in [0, \pi/2]. \tag{2}$$

The larger the $\theta$, the more linearly independent the two vectors are. We illustrate this in Figure 3(b), where brighter the heatmaps correspond to higher rank matrices. We see that the embedded input, $\boldsymbol{\Phi}(\mathbf{X})$ from Chronos-Bolt, which is a subset of $\mathbb{R}^d = \mathbb{R}^{768}$, spans a subspace of significantly smaller dimension, i.e., the image of $\mathbb{R}^k = \mathbb{R}^{16}$ under $\phi$.

## 3 FROM LOW-RANK INPUTS TO LOW-RANK ATTENTION MATRICES

Let $\mathbf{U} \in \mathbb{R}^{d \times L}$ be an input embedded in the hidden space. Recall that in section 2 we showed that for TSFMs, $\mathbf{U}$ often has a low numerical rank. This immediately implies $\mathbf{U}$ can be expressed in a low-rank format: $\mathbf{U} \approx \mathbf{U}_1\mathbf{U}_2$, where $\mathbf{U}_1 \in \mathbb{R}^{d \times \tilde{d}}$ and $\mathbf{U}_2 \in \mathbb{R}^{\tilde{d} \times L}$ for some $\tilde{d} \ll d$. This yields faster matrix-matrix products with $\mathbf{U}$, but a limitation is that this representation requires an expensive rank-revealing matrix factorization (Damle et al., 2024), which adds overhead, particularly during backpropagation. If so, how do we leverage the low-rank structure of TSFM embeddings?

From basic linear algebra, it is known that for a linear operator $\mathbf{T} : \mathbb{R}^d \to \mathbb{R}^d$ to act on an $r$-dimensional subspace, one only needs to specify the operator in $r$ directions, in which case the operator can then be well-approximated by a low-rank matrix $\tilde{\mathbf{T}}$ whose rank scales with $r$ instead of the full width $d$ (Damle et al., 2024; Ipsen & Saibaba, 2024). An attention layer, defined by

$$\text{Attention}(\mathbf{U}; \mathbf{W}_Q, \mathbf{W}_K, \mathbf{W}_V) = \mathbf{W}_V\mathbf{U}\,\text{softmax}\left(\frac{\mathbf{U}^\top\mathbf{W}_Q^\top\mathbf{W}_K^\top\mathbf{U}}{\sqrt{\tilde{d}}}\right), \quad \mathbf{W}_Q, \mathbf{W}_K, \mathbf{W}_V \in \mathbb{R}^{d \times d},$$

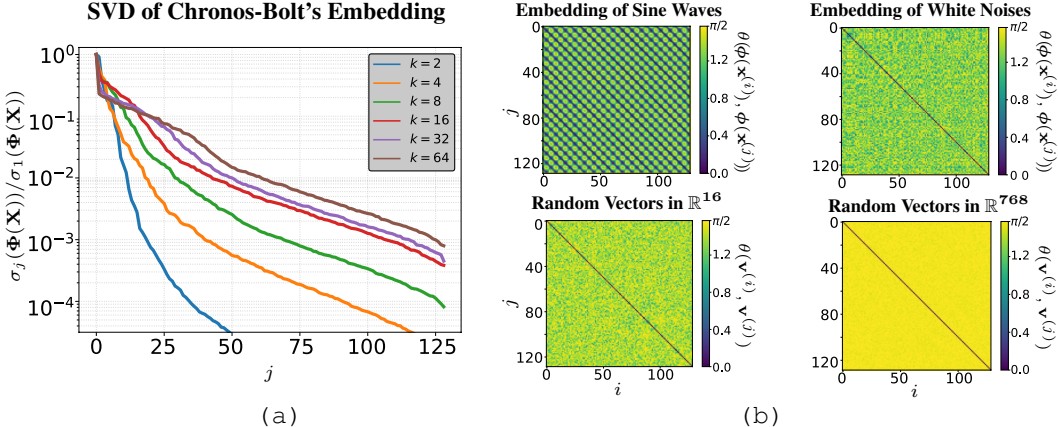

**Figure 3:** (a): Singular values of the embedded input matrices from Chronos-Bolt models pre-trained with different patch sizes $k$. (b): Angles between Chronos-Bolt's embedded vectors in $\mathbb{R}^d = \mathbb{R}^{768}$ defined in eq. (2), where the patches $\mathbf{x}^{(i)}$ are from a sinusoidal wave and Gaussian white noises, respectively. We also plot the angles between i.i.d. random Gaussian vectors in $\mathbb{R}^k = \mathbb{R}^{16}$ and $\mathbb{R}^d = \mathbb{R}^{768}$ for comparison.

while nonlinear, contains three linear transformations: namely, the queries, the keys, and the values. In this section, we establish Theorem 3, which supports the low-rank representations of $\mathbf{W}_Q$, $\mathbf{W}_K$, and $\mathbf{W}_V$ given a low-rank $\mathbf{U}$ (see Appendix D for the proof).

**Theorem 3.** Let $C > 0$ be a constant. Let $\mathbf{\Xi} = [\mathbf{x}_1 \ \cdots \ \mathbf{x}_N] \in \mathbb{R}^{d \times N}$ be given for some $d, N \geq 1$, $\mathbf{x}_j \in \mathbb{R}^d$, and $\|\mathbf{x}_j\|_2 \leq C$ for all $1 \leq j \leq N$. Let $\mathbf{W}_Q, \mathbf{W}_K, \mathbf{W}_V \in \mathbb{R}^{d \times d}$ be matrices such that $\|\mathbf{W}_Q^\top \mathbf{W}_K\|_2 \leq C\sqrt{d}$, and $\|\mathbf{W}_V\|_2 \leq C\sqrt{d}$. The following two statements hold:

1. **(Attention matrices are compressible on low-rank inputs.)** For any $\tilde{d} < d$ such that $\sigma_{\tilde{d}+1} := \sigma_{\tilde{d}+1}(\mathbf{\Xi}) \leq 1$, there exist $\tilde{\mathbf{W}}_Q, \tilde{\mathbf{W}}_K, \tilde{\mathbf{W}}_V \in \mathbb{R}^{d \times d}$ with $\mathrm{rank}(\tilde{\mathbf{W}}_Q) = \mathrm{rank}(\tilde{\mathbf{W}}_K) = \mathrm{rank}(\tilde{\mathbf{W}}_V) = \tilde{d}$, $\|\tilde{\mathbf{W}}_Q^\top \tilde{\mathbf{W}}_K\|_2 \leq \|\mathbf{W}_Q^\top \mathbf{W}_K\|_2$, and $\|\tilde{\mathbf{W}}_V\|_2 \leq \|\mathbf{W}_V\|_2$, such that given any matrix $\mathbf{U} \in \mathbb{R}^{d \times L}$ for any $L \geq 1$, where each column of $\mathbf{U}$ is a column of $\mathbf{\Xi}$, we have that

$$\left\|\mathrm{Attention}(\mathbf{U}; \mathbf{W}_Q, \mathbf{W}_K, \mathbf{W}_V) - \mathrm{Attention}(\mathbf{U}; \tilde{\mathbf{W}}_Q, \tilde{\mathbf{W}}_K, \tilde{\mathbf{W}}_V)\right\|_F \leq \mathcal{O}\left(\sqrt{d}\,\sigma_{\tilde{d}+1}\right), \quad (3)$$

where the constant in the $\mathcal{O}$-notation only depends on $C$.

2. **(Attention matrices are incompressible on high-rank inputs.)** The upper bound in eq. (3) is tight up to a factor of $\sqrt{d}$. That is, fix some $d \geq 1$, $L \geq d$ and $1 \geq \sigma_1 \geq \sigma_2 \geq \cdots \geq \sigma_d > 0$. There exist $\mathbf{U} \in \mathbb{R}^{d \times L}$ with $\sigma_j(\mathbf{U}) = \sigma_j$ for all $1 \leq j \leq d$, and $\mathbf{W}_Q, \mathbf{W}_K \in \mathbb{R}^{d \times d}$ such that for any $\tilde{d} < d$, any orthogonal matrix $\mathbf{W}_V \in \mathbb{R}^{d \times d}$, and any rank-$\tilde{d}$ matrix $\tilde{\mathbf{W}}_V \in \mathbb{R}^{d \times d}$, we have

$$\left\|\mathrm{Attention}(\mathbf{U}; \mathbf{W}_Q, \mathbf{W}_K, \mathbf{W}_V) - \mathrm{Attention}(\mathbf{U}; \mathbf{W}_Q, \mathbf{W}_K, \tilde{\mathbf{W}}_V)\right\|_F \geq \frac{1}{4}\sigma_{\tilde{d}+1}. \quad (4)$$

Note that Theorem 3 is a purely numerical statement that holds for any low-rank embedding $\mathbf{\Xi}$. It applies to time series because our analysis in section 2 reveals the low-rank nature of $\mathbf{\Xi}$, but it nonetheless works for other modalities provided that $\mathbf{\Xi}$ is low-rank. The first statement of Theorem 3 says, at a high level, that if the inputs $\mathbf{U}$ come from a low-rank vocabulary $\mathbf{\Xi}$, i.e., $\sigma_{\tilde{d}+1}$ is small, then one only needs low-rank attention matrices $\tilde{\mathbf{W}}_Q, \tilde{\mathbf{W}}_K, \tilde{\mathbf{W}}_V$ on $\mathbf{U}$. Here, it is important that we fix the low-rank embedded space $\mathbf{\Xi}$ and prove a uniform bound eq. (3) that holds for all input $\mathbf{U}$, so that our low-rank approximation $\tilde{\mathbf{W}}_Q, \tilde{\mathbf{W}}_K, \tilde{\mathbf{W}}_V$ is not input-dependent. Since TSFMs have a low-rank vocabulary $\mathbf{\Xi}$ (see section 2), high-rank attention matrices in TSFMs can be approximated by low-rank ones. See Appendix J for a numerical experiment. We emphasize that this low-rank property does not depend on the temporal simplicity of a particular time series (e.g., being constant), but rather on the intrinsic low-dimensionality of the input embedding space $\mathbf{\Xi}$ itself, which makes $\mathbf{U}$ low-rank-sufficient for any time-series input. For other data modalities than time-series, e.g., TFMs, ViTs, and LLMs (see Figure 2), where $\mathbf{U}$ has a higher rank, the second statement of Theorem 3 suggests that the attention matrices are less compressible, as we will show in section 5.

To illustrate the concepts in Theorem 3 on TSFMs, we take pretrained Chronos models and T5 LLMs of different hidden dimensions $d$. These two models have the same Transformer size and architecture, and differ only in the pretraining data modality. For each model and a fixed $\varepsilon > 0$, we compute the averaged $\varepsilon$-rank of all query projection matrices $\mathbf{W}_Q$ from the model. Figure 4(a,b) show that as the size of $\mathbf{W}_Q$ increases, the matrix $\mathbf{W}_Q$ stays low-rank in a Chronos model, and its rank scales almost linearly with $d$ in a T5 LLM. For Chronos models, where the vocabulary is embedded in a low-rank subspace, low-rank attention matrices suffice to capture most information in the inputs. This empirical observation goes beyond Theorem 3, which proves the sufficiency of low-rank $\mathbf{W}_{Q/K/V}$, but it does not guarantee their appearance via training. Figure 4(a) shows that training, while independent from expressiveness, gives rise to low-rank weights (see Appendix G for more insights). This provides motivation for pretraining a compressed model and for compressing a pretrained one (see section 5). In Appendix G, we show an analysis of the distribution of the singular values of attention matrices, in pretrained foundation models and also during training.

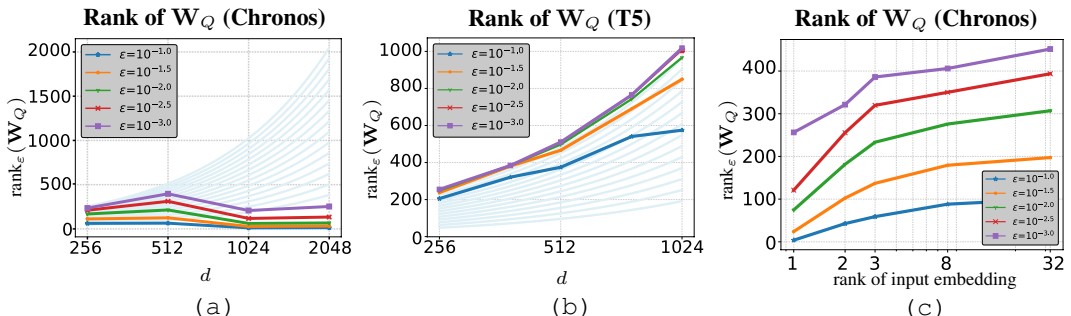

**Figure 4:** The averaged $\varepsilon$-rank of query projection matrices $\mathbf{W}_Q$ in pretrained Chronos models and T5 LLMs. In (a,b), we vary the hidden dimension $d$. The light blue curves are contours of the ratio between the horizontal and the vertical axes in the semilog-x scale. In (c), we fix the hidden dimension $d = 512$ and change the rank of the fixed input embedding $\mathbf{\Xi}$ (see Appendix I).

To further corroborate the role of the rank $\tilde{d}$ in Theorem 3, we pretrain Chronos models with a fixed vocabulary. We increase the (algebraic) rank of $\mathbf{\Xi}$ from 1 to 32 (see Appendix I), where we observe that the numerical ranks of attention matrices increase with rank($\mathbf{\Xi}$) (see Figure 4(c)).

While our discussion in this section assumes a single head in the attention layer, in Appendix E we extend the result to the multi-head case. From the standpoint of representation complexity, Theorem 3 applies equally to multi-head attention, and it is independent of the number of heads (see Theorem 5 in Appendix D). In practice, however, we observe that the $\mathbf{W}_{Q/K/V}$ matrices in pretrained multi-head layers tend to have higher numerical ranks (see Figure 9). This effect can be understood through the lens of numerical linear algebra, via a concept called "sketching" (see Appendix E). It helps explain why additional heads can improve robustness and training stability, even though the underlying complexity result is head-agnostic (see Theorem 6 in Appendix E).

## 4 FLOW-OF-RANKS: MOVING THROUGH A TRANSFORMER

Our prior discussions in section 3 focused on compressing a single attention layer, but a Transformer in a TSFM has many layers. While we showed that the input into the first attention layer is low-rank (see section 2), one may wonder: what about the inputs into a later layer? If you apply a linear transformation to a low-rank input, then it at most preserves, if not decreases, the rank of the input; however, each layer in a Transformer is generally nonlinear. Here, we demonstrate that these nonlinear layers can increase the rank of the input.

We refer to the phenomenon of the increasing rank of the input as it goes deeper into the layers of a model as the *flow-of-ranks*. Our contribution here is twofold: we quantify this rank growth across layers in a deep Transformer by proving Theorem 4 and then show what that means to Transformers applied to time-series data. To see the "flow-of-ranks" in practice, we show the numerical ranks of attention matrices per layer in a Chronos model and a T5 LLM in Figure 5(a,b). The numerical ranks of attention matrices become higher for deeper layers because of the flow-of-ranks (see Figure 5(c)). For a T5 LLM, only with a large $\varepsilon$ do we observe an increase in the $\varepsilon$-rank, because the $\varepsilon$-rank with a small $\varepsilon$ is already saturated (i.e., close to the matrix's dimension) to cap-

ture the high-rank vocabulary $\Xi$. To formalize this discussion, we prove how an attention layer increases the small singular values of a low-rank input matrix. Conceptually, this "flow-of-ranks" can be viewed as a mechanism by which the model lifts a low-dimensional but complex signal into a higher-dimensional space where the underlying autocorrelation becomes simpler. This is analogous to the Koopman operator framework, which similarly transforms nonlinear dynamics into a higher-dimensional representation with approximately linear evolution.

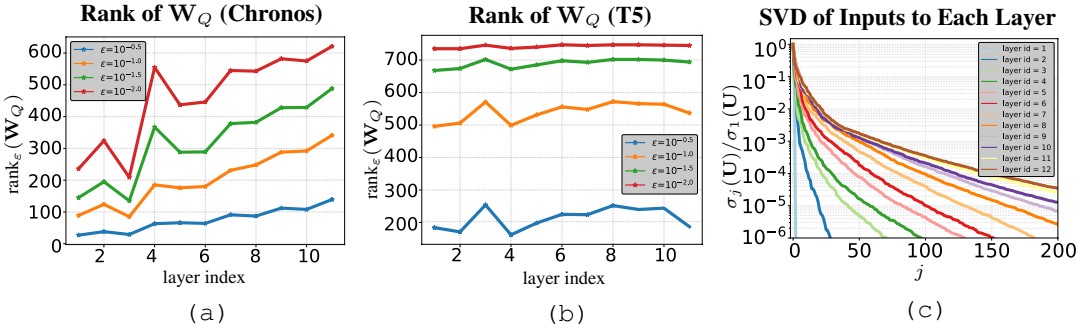

**Figure 5:** (a,b): The $\varepsilon$-rank of every query projection $\mathbf{W}_Q$ in the encoder of a Chronos model and a T5 model of the same size, respectively. (c): Singular values of the input matrix to each Chronos' encoder layer, starting with a constant signal $(x_1, \ldots, x_T) = (0, \ldots, 0)$.

**Theorem 4.** Given positive integers $d \leq L$, let $\mathbf{U} \in \mathbb{R}^{d \times L}$ be an input matrix with singular values $1 = \sigma_1 \geq \cdots \geq \sigma_d > 0$. Let $D \geq 1$ be the number of layers of the model. Let $h$ be the number of heads and $d_h$ be the per-head dimension so that $d = h \times d_h$. The following statements hold:

1. Suppose $\sqrt{D} \geq 2e^2 h$. For every $1 \leq i \leq h$, let $\|\mathbf{W}_Q^{(i)\top} \mathbf{W}_K^{(i)}\|_2 \leq \sqrt{d_h}$, and $\|\mathbf{W}_V^{(i)}\|_2 \leq 1$ be any attention matrices (see eq. (18) for the notation). For any $1 \leq k \leq d$, we have

$$\sigma_k(\mathbf{Z})/\sigma_1(\mathbf{Z}) \leq 2 \min_{1 \leq j \leq k} \left( \sigma_{k-j+1} + \frac{e^2 h}{\sqrt{D}} \sigma_{\lfloor (j-1)/h \rfloor + 1} \right), \tag{5}$$

where $\mathbf{Z} = \mathbf{U} + \mathbf{Y}/\sqrt{D}$ and $\mathbf{Y} = \text{MH-Attention}(\mathbf{U}; \mathbf{W}_Q, \mathbf{W}_K, \mathbf{W}_V, h)$.

2. The upper bound above is tight. More precisely, given any $d, L, N, h$, and singular values $1 = \sigma_1 \geq \cdots \geq \sigma_d > 0$, there exist an input $\mathbf{U} \in \mathbb{R}^{d \times L}$ with singular values $\sigma_1, \ldots, \sigma_d$, and matrices $\mathbf{W}_Q$, $\mathbf{W}_K$, and $\mathbf{W}_V$ with $\|\mathbf{W}_V^{(i)}\|_2 \leq 1$, such that for any $1 \leq k \leq d$, we have

$$\sigma_k(\mathbf{Z})/\sigma_1(\mathbf{Z}) \geq \frac{1}{4} \left( \sigma_{(i-1)d_h + \lceil k/i \rceil} + \frac{1}{\sqrt{D}} \sigma_{\lceil k/i \rceil} \right), \tag{6}$$

for every $i \leq h$ such that $\lceil k/i \rceil < d_h$.

See Appendix F for the proof of Theorem 4 and Appendix J for a numerical experiment. The matrix $\mathbf{Z}$ can be interpreted as the output of a residual attention layer given input $\mathbf{U}$, where the scaling factor $1/\sqrt{D}$ is commonly used in a Transformer to stabilize and accelerate training (De & Smith, 2020). The upper bound in eq. (5) provides a guarantee that the rank of the output cannot be arbitrarily large if the input is low-rank. In this bound, there are two terms balancing each other: $\sigma_{k-j+1}$, which increases as $j$ increases, and $\sigma_{\lfloor (j-1)/h \rfloor + 1}$, which decreases as $j$ increases. Without knowing the precise distribution of the singular values, the $j$ which minimizes eq. (5) cannot be determined. To understand why the upper bound is tight, we prove a corollary in a simplified one-layer setting.

**Corollary 1.** Using the notations in Theorem 4 and assuming $D = 1$, we have $\sigma_k(\mathbf{Z}) \leq \mathcal{O}(h)\sigma_{\lfloor (k+1)/(h+1) \rfloor + 1}$. In addition, the lower bound eq. (6) satisfies that $\sigma_k(\mathbf{Z}) \geq \mathcal{O}(1)\sigma_{\lceil k/h \rceil}$ for every $k \leq d - d_h$. The constants in both $\mathcal{O}$-notations are universal.

The corollary states that when $\mathbf{U}$ goes through an attention layer, the $k^{\text{th}}$ singular value of the output $\mathbf{Z}$ can be increased to at most the order of magnitude of the $\lfloor k/h \rfloor^{\text{th}}$ singular value of $\mathbf{U}$. The lower bound suggests that this can be achieved by some inputs, which proves the sharpness of the upper bound. Interestingly, this result says that the number of heads $h$ plays an important role in increasing the ranks of an input matrix. The term to note here is not the $\mathcal{O}(h)$ factor multiplied to the upper

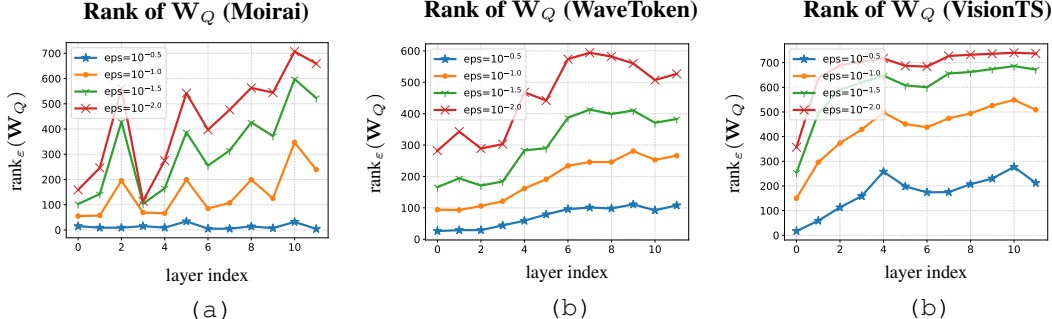

**Figure 6:** Flow-of-ranks is also observed for many TSFMs other than Chronos, including Moirai-1.0-R-base (Woo et al., 2024), WaveToken (base) (Masserano et al., 2025), and VisionTS (Chen et al., 2025). The hidden dimension $d$ in all three models are 768.

bound but the division by $h+1$ in the subindex $\lfloor(k+1)/(h+1)\rfloor+1$. When $h$ is large and the singular values decay fast, $\sigma_{\lfloor(k+1)/(h+1)\rfloor+1}(\mathbf{U})$ can be significantly larger than $\sigma_k(\mathbf{U})$ (see Figure 17).

In Figure 6, we also verify the flow-of-ranks in many TSFMs beyond Chronos. In general, we observe that the rank of an attention weight matrix in many TSFMs grows as a function of the layer index. The three models compared in Figure 6 also have distinct input space dimensions, with that of VisionTS significantly larger than those of Moirai and WaveToken, which is also reflected by the overall larger numerical ranks of the weight matrices of VisionTS.

## 5 USING THESE INSIGHTS: HOW TO COMPRESS A TSFM?

In this section, we apply our theoretical framework to compress large TSFMs. We pursue two approaches: first, motivated by the observation that attention matrices in many pretrained TSFMs are already low rank, we apply truncated SVD to each attention matrix of a pretrained model. Second, to achieve stronger compression, we design architectures that parameterize attention matrices in low rank from the start and pretrain them from scratch, using a layer-dependent hyperparameter schedule that accounts for the "flow-of-ranks." We present results on compressing Chronos, and we provide more results on another TSFM in Appendix K to support the broad applicability of our methods. Unlike LoRA's low-rank fine-tuning updates, we factorize our attention weights themselves, yielding inference-time efficiency and enabling pretraining with a compressed backbone.

**Compressing a Pretrained Model.** In Figure 4, we see that attention matrices in a large pretrained model usually do not have a full numerical rank. That means we can well-approximate a large matrix with a low-rank one. More precisely, let $\mathbf{W} \in \mathbb{R}^{d\times d}$ be an attention matrix and let $\mathbf{W} = \sum_{j=1}^{d} \sigma_j \mathbf{u}_i \mathbf{v}_i^\top$ be the singular value decomposition (SVD) of $\mathbf{W}$. Then, for a fixed $\varepsilon > 0$, the truncated SVD satisfies a relative error bound: $\left\| \mathbf{W} - \sum_{j=1}^{\mathrm{rank}_\varepsilon(\mathbf{W})} \sigma_j \mathbf{u}_i \mathbf{v}_i^\top \right\|_2 \leq \varepsilon \|\mathbf{W}\|_2$ (see eq. (1)). For a fixed $\varepsilon > 0$, we apply the truncated SVD to every attention matrix of the pretrained Chronos model without fine-tuning, which reduces the number of parameters. We compute the WQL and MASE losses (see Ansari et al. (2024a)) and their geometric means relative to the original model. A score below 1 means the reduced model performs better while it performs worse otherwise.

Table 2 shows that we can compress the attention matrices up to around 23.7% of the original size without any loss of performance. Reducing the size further takes away key information in attention matrices and results in a rapid performance deterioration, leading us to the next compression method.

**Pretraining a Compressed Model.** Table 2 shows a hard limit of compression: compressing the size to below about 20% makes the performance significantly worse. If one is given enough budget, a more robust method to obtain a smaller TSFM is by pretraining a compressed model. We parameterize a $d \times d$ attention matrix by a rank-$\tilde{d}$ representation. Driven by "flow-of-ranks" from section 4, we choose to let $\tilde{d}$ be layer-dependent. In the $i$th layer of the model, we use a $\tilde{d}_i = \tilde{d}(i) = \lceil \tilde{d}_0(1+i)^\alpha \rceil$, where $\tilde{d}_0 > 0$ and $\alpha \geq 0$ are two hyperparameters. This design is motivated by the fact that the numerical rank of an input to a layer increases as the layer index, and we need attention matrices of higher ranks to capture them (see Theorem 3). In Figure 7, we see how pretraining a compressed model allows us to expand the time–accuracy Pareto frontier of TSFMs, which show that pretrain-

**Table 2:** Results of compressing a pretrained Chronos or T5 model. We apply truncated SVD with a specific $\varepsilon$, which results in a model whose attention matrices are compressed to the "Ratio" of the original size. We compare the performance scores relative to the original pretrained model, where "LPPL" stands for the logarithm of the perplexity. Overlap is obtained by selecting the top-10 tokens from the original model and the compressed one, and computing their Jaccard overlap.

| Ratio | Chronos In-Domain ↓ WQL | MASE | Zero-Shot ↓ WQL | MASE | Overlap ↑ | T5 LPPL ↓ | Overlap ↑ |
|---|---|---|---|---|---|---|---|
| 0.073 | 4.409 | 4.095 | 3.562 | 3.435 | 0.308 | 3.313 | 0.227 |
| 0.151 | 1.991 | 2.412 | 1.566 | 1.576 | 0.717 | 2.530 | 0.301 |
| 0.237 | 1.053 | 1.005 | 1.030 | 1.011 | 0.883 | 1.652 | 0.345 |
| 0.393 | 1.009 | 1.024 | 0.990 | 0.994 | 0.979 | 1.544 | 0.568 |
| 0.569 | 1.003 | 0.952 | 0.945 | 0.995 | 0.997 | 1.290 | 0.730 |
| 0.755 | 1.007 | 1.021 | 1.027 | 1.000 | 0.999 | 1.085 | 0.871 |
| 0.889 | 1.000 | 0.960 | 1.016 | 1.001 | 0.999 | 1.028 | 0.954 |
| 1.000 | 1.000 | 1.000 | 1.000 | 1.000 | 1.000 | 1.000 | 1.000 |

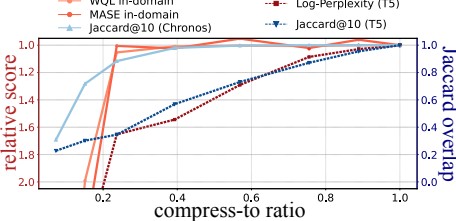

ing a compressed model is more robust than compressing a pretrained one. In fact, in Table 3 of Appendix K, we show that pretraining a compressed Chronos-Bolt allows us to outperform even traditional local methods on *both* time and accuracy.

| $\tilde{d}_0$ | $\alpha$ | Size Ratio | Inference Time | Space | Embedding From Scratch In-Domain WQL ↓ | MASE ↓ | Zero-Shot WQL ↓ | MASE ↓ | Reuse Embedding In-Domain WQL ↓ | MASE ↓ | Zero-Shot WQL ↓ | MASE ↓ |
|---|---|---|---|---|---|---|---|---|---|---|---|---|
| 3 | 0.27 | 0.075 | 0.346 | 0.186 | 1.034 | 0.988 | 0.966 | 0.982 | 1.031 | 1.047 | 1.087 | 1.084 |
| 5 | 0.35 | 0.150 | 0.398 | 0.312 | 1.048 | 0.982 | 1.080 | 1.055 | 1.025 | 0.930 | 0.936 | 0.960 |
| 7 | 0.40 | 0.250 | 0.494 | 0.440 | 1.021 | 0.949 | 0.996 | 1.019 | 0.999 | 1.007 | 0.949 | 0.943 |
| 64 | 0.00 | 1.000 | 1.000 | 1.000 | 1.000 | 1.000 | 1.000 | 1.000 | 1.000 | 1.000 | 1.000 | 1.000 |

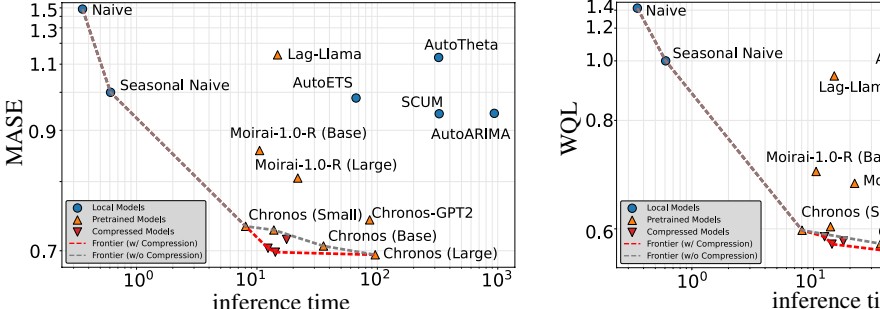

**Figure 7:** Results of pretraining a compressed Chronos model. We compare the performance scores *relative to the original pretrained model*. We show prediction losses for both models whose embedding matrix is randomly initialized and models whose embedding matrix is inherited from the original pretrained model. The last row is the baseline.

To show the generality of our analysis in this paper, we also pretrain compressed Moirai-1.0-R-base models. We show the results in Table 4 of Appendix K. It clearly shows the benefit of the flow-of-ranks design: given a fixed compression ratio, a model with a different reduced rank per layer performs significantly better than one whose rank remains layer-independent.

## 6 CONCLUSION

We have developed a data/modality-dependent framework via the lens of rank structure for analyzing the structure of and design decisions for Transformers, and we have applied it to Chronos and Chronos-Bolt, two popular TSFMs. Our results highlight how properties of the model depend on and interact with properties of the input data, and they lead to concrete principles for the design of models, parameters, and hyperparameters. We illustrate our proof-of-principle results by showing the compressibility of TSFMs in comparison to Transformers trained on text data.

### ACKNOWLEDGMENTS

The authors would like to thank Yongjun He, Junming Yin, Dmitry Efimov, and Boris Oreshkin for many fruitful discussions on this work.

REPRODUCIBILITY STATEMENT.

Code used to produce the experiments in this paper, along with scripts for data processing and figure generation, is available at https://github.com/amazon-science/tsfm-compression.

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

# A    RELATED WORK

## A.1    TIME-SERIES MODELING

Time series data are ubiquitous in many domains, including scientific (Abhishek et al., 2012; Zhang & Gilpin, 2025; Lai et al., 2025), industrial (Hong et al., 2016), and financial applications (Zhang et al., 2001), where they facilitate critical tasks, such as forecasting (Hyndman & Athanasopoulos, 2018), imputation (Yoon et al., 2018), and anomaly detection (Blázquez-García et al., 2021). Classical time-series forecasting (Hyndman & Athanasopoulos, 2018) has deep roots in traditional methods such as ARIMA (Makridakis & Hibon, 1997) and exponential smoothing (Gardner Jr, 1985). Many modern approaches involve deep learning; for example, sequence models such as DeepAR (Salinas et al., 2020) popularized probabilistic forecasting at scale, while Lim et al. (2021) combined the attention mechanism with interpretability for multi-horizon tasks. There are other neural forecasters without attention, such as N-BEATS (Oreshkin et al., 2019) and N-HiTS (Challu et al., 2023). An evolving modern alternative that targets long contexts efficiently is the recently developed state-space models (Gu et al., 2021) and Mamba (Gu & Dao, 2023). These neural-network-based methods are usually considered task-specific — that is, given a task, one trains a model to obtain a specific set of weights tailored to that task.

## A.2    TIME-SERIES FOUNDATION MODELS

Recent work pretrains general-purpose time series models across domains and tasks. Examples include TimesFM (Das et al., 2024), Chronos (Ansari et al., 2024a), Moirai (Woo et al., 2024), and Time MOE (Shi et al., 2025). These models differ in tokenization choices, training corpora, and zero-shot protocols, but they share the goal of one model that transfers across datasets and horizons.

## A.3    TIME-SERIES TOKENIZATION AND EMBEDDING

Designing the input representation is the key to leveraging the high flexibility and expressive power of large Transformers. Patching turns small motifs into tokens, as in Nie et al. (2022) and Das et al. (2024). Discrete tokenization via quantization has been explored by Ansari et al. (2024a) and Talukder et al. (2024), and other discrete designs include HDT (Feng et al., 2025) and vector-quantized methods (Gui et al., 2024). Another line of research is by using frequency-based tokenizers, such as WaveToken (Masserano et al., 2025). Lately, traditional language embedding strategies such as Byte-Pair Encoding are also considered (Götz et al., 2025).

## A.4    LOW-RANK STRUCTURES IN DEEP LEARNING

Low-rank structure shows up in the deep learning community as both an inductive bias (Martin & Mahoney, 2021; Wilson, 2025; Yu et al., 2024) and a compression tool (Sharma et al., 2023; Gu et al., 2022; Yu et al., 2023). There is also a line of research that looks into parameter-efficient finetuning called LoRA (Hu et al., 2022), which learns low-rank updates to weight matrices. We note that LoRA is different from our low-rank model compression strategy because LoRA essentially learns a high-rank-plus-low-rank expression of the weight matrices, and does not facilitate storage or inference of a model. Earlier work reduced cost via matrix or tensor factorizations, especially in the CNN community (Denton et al., 2014; Jaderberg et al., 2014; Lebedev et al., 2014). These ideas motivate studying when low-rank operators suffice for sequence models, while our paper provides clean justifications for why low-rank operators suffice for a time-series foundation model.

## A.5    LOW-RANK STRUCTURES IN TRANSFORMERS

Low-rank structure in Transformers and attention has been extensively exploited for both efficiency and interpretability. A first line of work directly approximates the quadratic self-attention matrix with low-rank kernels: Linformer (Wang et al., 2020) projects keys and values to a low-dimensional subspace, yielding self-attention with linear complexity in sequence length, while Nyströmformer (Xiong et al., 2021) uses a Nyström approximation of the softmax kernel based on a small set of landmark tokens. Random-feature methods such as Performer (Choromanski et al., 2021) further view softmax attention as a kernel and approximate it with low-rank random feature

maps, obtaining linear-time and memory. Scatterbrain (Chen et al., 2021) analyzes when sparse versus low-rank attention yields better approximations and proposes a unified sparse+low-rank estimator that attains lower error than either component alone. Complementary approaches parameterize the weight matrices themselves in low rank: LoRA (Hu et al., 2022) learns low-rank updates on top of frozen full-rank weights for parameter-efficient fine-tuning of large language models, and Sharma et al. (2023) perform layer-selective rank reduction to probe and improve the reasoning behavior of Transformers. These works treat low rank primarily as an architectural or algorithmic assumption used to accelerate or regularize attention. By contrast, our analysis starts from the observed low numerical rank of time-series embeddings, proves when attention matrices over such embeddings are provably compressible, and then uses this data-driven perspective to design TSFMs whose attention layers are low rank by construction and compressible, which accelerates both pretraining and inference.

## B  PROOF OF THEOREM 1 AND COROLLARY 2

Before we go into the proofs of Theorem 1 and Corollary 2, we present a figure to illustrate the differences between the quantization-based embedding and continuous embedding discussed in Section 2.

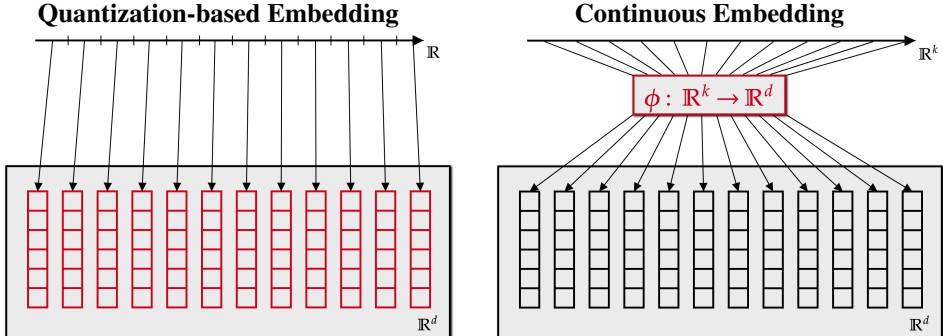

**Figure 8:** A comparison of the quantization-based embeddings and continuous embeddings. In a quantization-based one, we discretize the input domain into a number of regions and map each of them to a trainable vector. In a continuous embedding, we use a trainable function (usually an MLP) to map each element into the hidden space $\mathbb{R}^d$. The red parts in the figure highlight the trainable components.

The proof of Theorem 1 relies on classic polynomial approximation results. Intuitively, if a function $\phi$ is well-approximated by a low-degree polynomial, then the low-degree polynomial can be represented with a small basis, spanning a low-dimensional subspace. We now formalize this notion by providing the proof.

*Proof of Theorem 1.* Fix some $1 \leq j \leq d$. Let $P_{i,j}$ be the best degree-$(j-1)$ polynomial approximation of $\phi_i$ in the infinity norm, and let

$$\delta_j = \max_{1 \leq i \leq d} \|\phi_i - P_{i,j}\|_{L^\infty([-1,1])}.$$

Since $P_{i,j}$ is a polynomial of degree $\leq j - 1$, we can write it into

$$P_{i,j}(x) = \sum_{k=0}^{j-1} a_{i,k} T_k(x),$$

where $T_k$ is the degree-$k$ Chebyshev polynomial. Then, we can write $\Xi$ into

$$\Xi = \begin{bmatrix} — P_{1,j}(x) — \\ \vdots \\ — P_{d,j}(x) — \end{bmatrix} + \underbrace{\begin{bmatrix} — (\phi_1 - P_{1,j})(x) — \\ \vdots \\ — (\phi_d - P_{d,j})(x) — \end{bmatrix}}_{\mathbf{E}_j} = \underbrace{\begin{bmatrix} a_{1,0} & \cdots & a_{1,j-1} \\ \vdots & \ddots & \vdots \\ a_{d,0} & \cdots & a_{d,j-1} \end{bmatrix} \begin{bmatrix} — T_0(x) — \\ \vdots \\ — T_{j-1}(x) — \end{bmatrix}}_{\Xi_j} + \mathbf{E}_j.$$

Since $\Xi_j$ is a quasimatrix of rank at most $j$, by (Townsend & Trefethen, 2015, Thm. 4.2), we have that

$$s_{j+1} \leq \|\mathbf{E}_j\|_F = \sqrt{\sum_{i=1}^{d} \|\phi_i - P_{i,j}\|_{L^2([-1,1])}^2} \leq 2\sqrt{\sum_{i=1}^{d} \|\phi_i - P_{i,j}\|_{L^\infty([-1,1])}^2} \leq 2\sqrt{d}\,\delta_j. \quad (7)$$

Similarly, note that

$$
\mathbf{\Psi} = \begin{bmatrix} P_{1,j}(x_1) & \cdots & P_{1,j}(x_L) \\ \vdots & \ddots & \vdots \\ P_{d,j}(x_1) & \cdots & P_{d,j}(x_L) \end{bmatrix} + \underbrace{\begin{bmatrix} (\phi_1 - P_{1,j})(x_1) & \cdots & (\phi_1 - P_{1,j})(x_L) \\ \vdots & \ddots & \vdots \\ (\phi_d - P_{d,j})(x_1) & \cdots & (\phi_d - P_{d,j})(x_L) \end{bmatrix}}_{\mathbf{F}_j}
$$

$$
= \underbrace{\begin{bmatrix} a_{1,0} & \cdots & a_{1,j-1} \\ \vdots & \ddots & \vdots \\ a_{d,0} & \cdots & a_{d,j-1} \end{bmatrix} \begin{bmatrix} T_0(x_1) & \cdots & T_0(x_L) \\ \vdots & \ddots & \vdots \\ T_{j-1}(x_1) & \cdots & T_{j-1}(x_1) \end{bmatrix}}_{\mathbf{\Psi}_j} + \mathbf{F}_j.
$$

Since the rank of $\mathbf{\Psi}_j$ is at most $j$, by the Eckart–Young inequality, we have that

$$
\sigma_{j+1} \leq \|\mathbf{F}_j\|_2 \leq \|\mathbf{F}_j\|_F \leq \sqrt{dL}\delta_j. \tag{8}
$$

From (Trefethen, 2019, Thm. 7.2 & 8.2), we have that

$$
\delta_j \leq \frac{2V}{\pi\nu(j-1-\nu)^\nu} \qquad \text{and} \qquad \delta_j \leq \frac{2M\rho^{-j+1}}{\rho - 1} \tag{9}
$$

when $\phi_i$ satisfies the condition in the first and the second statement of the theorem, respectively. Hence, the two statement are proved by combining eq. (7), (8), and (9). $\qquad\square$

The activation function $\text{swish}_\beta$ used in Time MOE is analytic, and its domain of analyticity increases as $\beta \to 0^+$. Hence, we can use Theorem 1 to prove that Time MOE's embedding is low-rank.

**Corollary 2.** The embedding used in Time-MoE, defined by $\phi_i(x) = \text{swish}_\beta(w_i x) \cdot (v_i x) = (w_i v_i x^2)/(1 + e^{-\beta x})$, where we assume $|w_i v_i| \leq 1$ and $\beta > 0$, satisfies that

$$
\sigma_{j+1} = \mathcal{O}\left(\sqrt{dL}(\beta + \beta^{-1})(1 + \pi/(2\beta))^{-j+1}\right),
$$

where $\sigma_{j+1}$ is defined in Theorem 1 for any $\mathbf{\Psi}$ and the constant in the $\mathcal{O}$-notation is universal.

*Proof.* The function $\phi_i(z)$ is a meromorphic function with poles at $\{z \mid \exp(-\beta z) = -1\} = \{(2k+1)\pi i/\beta \mid k \in \mathbb{Z}\}$. Set $\rho = 1 + \pi/(2\beta)$. Then, within the Bernstein ellipse $E_\rho$ of radius $\rho$, we have that

$$
\text{Re}\left(e^{-\beta x}\right) > -c \Rightarrow |\phi_i(x)| \leq \frac{|x|^2}{1-c} = \mathcal{O}(1 + \beta^{-2}), \qquad x \in E_\rho,
$$

where $0 < c < 1$ is a universal constant and the constant in the $\mathcal{O}$-notation is also universal. The corollary follows from Theorem 1. $\qquad\square$

## C   PROOF OF THEOREM 2

The intuition in Theorem 2 is that if the MLP does not have a nonlinear activation function, then the linear MLP clearly maps a low-rank matrix to a low-rank matrix. While a nonlinear activation function complicates things, a contractive function does not increase the Frobenius norm of a matrix, which is equivalent to the $\ell^2$-norm of all singular values.

*Proof of Theorem 2.* Since $\mathbf{X}$ is a rank-$k$ matrix, the rank of $\mathbf{W}_1\mathbf{X}$ is at most $k$. Hence, the Frobenius norm of the matrix satisfies that

$$\|\mathbf{W}_1\mathbf{X}\|_F^2 = \sum_{j=1}^{k} \sigma_j(\mathbf{W}_1\mathbf{X})^2 \leq k\,\sigma_1(\mathbf{W}_1\mathbf{X})^2.$$

Given that $|\omega(x)| \leq |x|$ for every $x \in \mathbb{R}$, we have

$$k\,\sigma_1(\mathbf{W}_1\mathbf{X})^2 \geq \|\mathbf{W}_1\mathbf{X}\|_F^2 \geq \|\omega(\mathbf{W}_1\mathbf{X})\|_F^2 = \sum_{j=1}^{\min(d_f,L)} \sigma_j(\omega(\mathbf{W}_1\mathbf{X}))^2.$$

Using the singular value inequalities, we have that

$$\sum_{j=1}^{\min(d_f,L)} \sigma_j(\mathbf{W}_2\,\omega(\mathbf{W}_1\mathbf{X}))^2 \leq \sigma_1(\mathbf{W}_2)^2 \sum_{j=1}^{\min(d_f,L)} \sigma_j(\omega(\mathbf{W}_1\mathbf{X}))^2$$
$$\leq k\,\sigma_1(\mathbf{W}_2)^2\,\sigma_1(\mathbf{W}_1\mathbf{X})^2 = k\|\mathbf{W}_2\|_2^2\|\mathbf{W}_1\mathbf{X}\|_2^2.$$

This is, we can control the number of large singular values of $\mathbf{W}_2\,\omega(\mathbf{W}_1\mathbf{X})$ by

$$|\{j \,|\, \sigma_j(\mathbf{W}_2\,\omega(\mathbf{W}_1\mathbf{X})) > \varepsilon\|\mathbf{W}_2\|_2\|\mathbf{W}_1\mathbf{X}\|_2\}| \leq \varepsilon^{-2}k.$$

Moreover, the rank of the matrix $\mathbf{W}_3\mathbf{X}$ is at most $k$. Using the singular value inequality that

$$\sigma_{i+j-1}(\mathbf{A}+\mathbf{B}) \leq \sigma_i(\mathbf{A}) + \sigma_j(\mathbf{B}), \qquad \mathbf{A},\mathbf{B} \in \mathbb{R}^{d\times L}, \quad i+j-1 \leq \min(d,L),$$

we have

$$\sigma_{k+j}(\phi(\mathbf{X})) \leq \sigma_{k+1}(\mathbf{W}_3\mathbf{X}) + \sigma_j(\mathbf{W}_2\,\omega(\mathbf{W}_1\mathbf{X})) = \sigma_j(\mathbf{W}_2\,\omega(\mathbf{W}_1\mathbf{X})), \quad j \leq \min(d,L) - k.$$

This gives us

$$|\{j \,|\, \sigma_j(\phi(\mathbf{X})) > \varepsilon\|\mathbf{W}_2\|_2\|\mathbf{W}_1\mathbf{X}\|_2\}| \leq |\{j \,|\, \sigma_j(\mathbf{W}_2\,\omega(\mathbf{W}_1\mathbf{X})) > \varepsilon\|\mathbf{W}_2\|_2\|\mathbf{W}_1\mathbf{X}\|_2\}| + k$$
$$\leq (1 + \varepsilon^{-2})k,$$

which proves the result. $\qquad\square$

# D  PROOF OF THEOREM 3

To prove the upper bound in Theorem 3, we will prove a stronger result concerning not only a single-head attention but also a multi-head one.

**Theorem 5.** Let $C > 0$ be a constant. Let $\mathbf{\Xi} = [\mathbf{x}_1 \quad \cdots \quad \mathbf{x}_N] \in \mathbb{R}^{d \times N}$ be an embedding matrix, where $d$ is the hidden dimension, $N$ is the vocabulary size, $\mathbf{x}_j \in \mathbb{R}^d$, and $\|\mathbf{x}_j\|_2 \leq C$ for all $1 \leq j \leq N$. Let $\mathbf{W}_Q, \mathbf{W}_K, \mathbf{W}_V \in \mathbb{R}^{d \times d}$ be multi-head attention matrices with $h$ heads, such that $\|(\mathbf{W}_Q^{(i)})^\top \mathbf{W}_K^{(i)}\|_2 \leq C\sqrt{d_h}$, and $\|\mathbf{W}_V^{(i)}\|_2 \leq C\sqrt{d_h}$ for every $1 \leq i \leq h$. For any $\tilde{d} < d$ such that $\sigma_{\tilde{d}+1} := \sigma_{\tilde{d}+1}(\mathbf{\Xi}) \leq 1$, there exists a stable low-rank approximation $\tilde{\mathbf{W}}_Q, \tilde{\mathbf{W}}_K, \tilde{\mathbf{W}}_V \in \mathbb{R}^{d \times d}$ with $\operatorname{rank}(\tilde{\mathbf{W}}_Q) = \operatorname{rank}(\tilde{\mathbf{W}}_K) = \operatorname{rank}(\tilde{\mathbf{W}}_V) = \tilde{d}$, $\|\tilde{\mathbf{W}}_Q^\top \tilde{\mathbf{W}}_K\|_2 \leq \|\mathbf{W}_Q^\top \mathbf{W}_K\|_2$, and $\|\tilde{\mathbf{W}}_V\|_2 \leq \|\mathbf{W}_V\|_2$, such that given any input matrix $\mathbf{U} \in \mathbb{R}^{d \times L}$ for any $L \geq 1$, where each column of $\mathbf{U}$ is a column of $\mathbf{\Xi}$, we have that the low-rank attention matrices uniformly approximate the original one:

$$\left\| \text{MH-Attention}(\mathbf{U}; \mathbf{W}_Q, \mathbf{W}_K, \mathbf{W}_V, h) - \text{MH-Attention}(\mathbf{U}; \tilde{\mathbf{W}}_Q, \tilde{\mathbf{W}}_K, \tilde{\mathbf{W}}_V) \right\|_F \leq \mathcal{O}\left(\sqrt{d}\, \sigma_{\tilde{d}+1}\right), \tag{10}$$

where the constant in the $\mathcal{O}$-notation only depends on $C$.

*Proof.* Fix a $\tilde{d} < d$. For the sake of simplicity, we assume, without loss of generality, that $C = 1$. Let

$$\mathbf{\Xi} = \mathbf{U}_{\tilde{d}} \mathbf{S}_{\tilde{d}} \mathbf{V}_{\tilde{d}}^\top, \qquad \mathbf{U}_{\tilde{d}} \in \mathbb{R}^{d \times \tilde{d}}, \quad \mathbf{S}_{\tilde{d}} \in \mathbb{R}^{\tilde{d} \times \tilde{d}}, \quad \mathbf{V}_{\tilde{d}} \in \mathbb{R}^{N \times \tilde{d}}$$

be the truncated SVD of $\mathbf{\Xi}$. Let $\mathbf{Q}_{\tilde{d}} = \mathbf{S}_{\tilde{d}} \mathbf{V}_{\tilde{d}}^\top \in \mathbb{R}^{\tilde{d} \times N}$. Therefore, we have

$$\|\mathbf{\Xi} - \mathbf{U}_{\tilde{d}} \mathbf{Q}_{\tilde{d}}\|_2 \leq \sigma_{\tilde{d}+1}(\mathbf{\Xi}).$$

Define $\tilde{\mathbf{W}}_Q, \tilde{\mathbf{W}}_K,$ and $\tilde{\mathbf{W}}_V$ by

$$\tilde{\mathbf{W}}_Q = \mathbf{W}_Q \mathbf{U}_{\tilde{d}} \mathbf{U}_{\tilde{d}}^\top \in \mathbb{R}^{d \times d}, \qquad \tilde{\mathbf{W}}_K = \mathbf{W}_K \mathbf{U}_{\tilde{d}} \mathbf{U}_{\tilde{d}}^\top \in \mathbb{R}^{d \times d}, \qquad \tilde{\mathbf{W}}_V = \mathbf{W}_V \mathbf{U}_{\tilde{d}} \mathbf{U}_{\tilde{d}}^\top \in \mathbb{R}^{d \times d}.$$

Since $\mathbf{U}_{\tilde{d}}$ has orthonormal columns, we have that $\|\mathbf{U}_{\tilde{d}}\|_2 = \|\mathbf{U}_{\tilde{d}}^\top\|_2 = 1$. By the sub-multiplicity of the spectral norm, we have that

$$\|\tilde{\mathbf{W}}_Q^\top \tilde{\mathbf{W}}_K\|_2 \leq \|\mathbf{W}_Q^\top \mathbf{W}_K\|_2, \qquad \|\tilde{\mathbf{W}}_V\|_2 \leq \|\mathbf{W}_V\|_2,$$

which proves the stability of the low-rank representation. Let $\mathbf{Q}_U \in \mathbb{R}^{\tilde{d} \times L}$ be the matrix defined by the condition that the $i$th column of $\mathbf{Q}_U$ is the $j$th column of $\mathbf{Q}_{\tilde{d}}$ if and only if the $i$th column of $\mathbf{U}$ is the $j$th column of $\mathbf{\Xi}$, and let

$$\mathbf{\Delta}_U = \mathbf{U}_{\tilde{d}} \mathbf{Q}_U - \mathbf{U}.$$

Since every column of $\mathbf{\Delta}_U$ is a column of $\mathbf{\Xi} - \mathbf{U}_{\tilde{d}} \mathbf{Q}_{\tilde{d}}$, its norm is no greater than $\sigma_{\tilde{d}+1}(\mathbf{\Xi})$, i.e.,

$$\begin{aligned}
\|\mathbf{\Delta}_U(:, j)\|_2 &\leq \sigma_{\tilde{d}+1}(\mathbf{\Xi}), \qquad 1 \leq j \leq L, \\
\|\mathbf{\Delta}_U\|_2 &\leq \|\mathbf{\Delta}_U\|_F \leq \sqrt{L} \sigma_{\tilde{d}+1}(\mathbf{\Xi}).
\end{aligned} \tag{11}$$

Next, we have

$$\begin{aligned}
\|\mathbf{\Delta}_V^{(i)}\|_2 &\leq \|\mathbf{\Delta}_U^{(i)}\|_2 \leq \sqrt{d_h L}\, \sigma_{\tilde{d}+1}(\mathbf{\Xi}), \\
\mathbf{\Delta}_V^{(i)} &:= \mathbf{W}_V^{(i)} \mathbf{U} - \tilde{\mathbf{W}}_V^{(i)} \mathbf{U} = \mathbf{W}_V^{(i)} \mathbf{U} - \mathbf{W}_V^{(i)} \mathbf{U}_{\tilde{d}} \mathbf{U}_{\tilde{d}}^\top \mathbf{U} \\
&= \mathbf{W}_V^{(i)} \mathbf{U} - \mathbf{W}_V^{(i)} \mathbf{U}_{\tilde{d}} \mathbf{U}_{\tilde{d}}^\top \mathbf{U}_{\tilde{d}} \mathbf{Q}_U \\
&= \mathbf{W}_V^{(i)} \mathbf{U} - \mathbf{W}_V^{(i)} \mathbf{U}_{\tilde{d}} \mathbf{Q}_U = -\mathbf{W}_V^{(i)} \mathbf{\Delta}_U,
\end{aligned}$$

where the first inequality comes from the sub-multiplicity of the spectral norm. Moreover, every column of $\mathbf{\Delta}_V^{(i)}$ is a column of $-\mathbf{W}_V^{(i)} \mathbf{\Delta}_U$ and must satisfy that

$$\|\mathbf{\Delta}_V^{(i)}(:, j)\|_2 \leq \sqrt{d_h}\, \sigma_{\tilde{d}+1}(\mathbf{\Xi}), \qquad 1 \leq j \leq L. \tag{12}$$

Similarly, we have that

$$\|\boldsymbol{\Delta}_Q^{(i)}\|_2 \le \sqrt{d_h}\, L(2\sigma_{\tilde{d}+1}(\boldsymbol{\Xi}) + \sigma_{\tilde{d}+1}(\boldsymbol{\Xi})^2),$$

$$\boldsymbol{\Delta}_Q^{(i)} := \mathbf{U}^\top (\mathbf{W}_Q^{(i)})^\top \mathbf{W}_K^{(i)} \mathbf{U} - \mathbf{U}^\top (\tilde{\mathbf{W}}_Q^{(i)})^\top \tilde{\mathbf{W}}_K^{(i)} \mathbf{U}$$

$$= \mathbf{U}^\top (\mathbf{W}_Q^{(i)})^\top \mathbf{W}_K^{(i)} \mathbf{U} - \mathbf{U}^\top \mathbf{U}_{\tilde{d}} \mathbf{U}_{\tilde{d}}^\top (\mathbf{W}_Q^{(i)})^\top \mathbf{W}_K^{(i)} \mathbf{U}_{\tilde{d}} \mathbf{U}_{\tilde{d}}^\top \mathbf{U}$$

$$= \mathbf{U}^\top (\mathbf{W}_Q^{(i)})^\top \mathbf{W}_K^{(i)} \mathbf{U} - \mathbf{Q}_X^\top \mathbf{U}_{\tilde{d}}^\top (\mathbf{W}_Q^{(i)})^\top \mathbf{W}_K^{(i)} \mathbf{U}_{\tilde{d}} \mathbf{Q}_U$$

$$= -\boldsymbol{\Delta}_U^\top (\mathbf{W}_Q^{(i)})^\top \mathbf{W}_K^{(i)} \mathbf{U} - \mathbf{U}^\top (\mathbf{W}_Q^{(i)})^\top \mathbf{W}_K^{(i)} \boldsymbol{\Delta}_U - \boldsymbol{\Delta}_U^\top (\mathbf{W}_Q^{(i)})^\top \mathbf{W}_K^{(i)} \boldsymbol{\Delta}_U,$$

where the inequality is obtained by recalling that $\|\mathbf{U}\|_2 \le \sqrt{L}$ and $\|\boldsymbol{\Delta}_U\|_2 \le \sqrt{L}\, \sigma_{\tilde{d}+1}(\boldsymbol{\Xi})$. Moreover, since we have $\|\mathbf{U}(:,j)\|_2 \le 1$ and $\|\boldsymbol{\Delta}_U(:,j)\|_2 \le \sigma_{\tilde{d}+1}(\boldsymbol{\Xi})$ for all $1 \le j \le N$, every entry of $\boldsymbol{\Delta}_Q^{(i)}$ satisfies that

$$\|\boldsymbol{\Delta}_Q^{(i)}(t,j)\|_2 \le \sqrt{d_h}(2\sigma_{\tilde{d}+1}(\boldsymbol{\Xi}) + \sigma_{\tilde{d}+1}(\boldsymbol{\Xi})^2), \qquad 1 \le t \le L, \quad 1 \le j \le L. \tag{13}$$

For each fixed $1 \le i \le h$, define the notations

$$\mathbf{G}^{(i)} = \frac{\mathbf{U}^\top (\mathbf{W}_Q^{(i)})^\top \mathbf{W}_K^{(i)} \mathbf{U}}{\sqrt{d_h}}, \qquad \tilde{\mathbf{G}}^{(i)} = \frac{\mathbf{U}^\top (\tilde{\mathbf{W}}_Q^{(i)})^\top \tilde{\mathbf{W}}_K \mathbf{U}}{\sqrt{d_h}},$$

$$\boldsymbol{g}_j^{(i)} = \mathbf{G}(:,j), \qquad \tilde{\boldsymbol{g}}_j^{(i)} = \tilde{\mathbf{G}}(:,j).$$

For simplicity, we drop the superscripts $(i)$ on $\mathbf{G}, \tilde{\mathbf{G}}, \boldsymbol{g}$, and $\tilde{\boldsymbol{g}}$. Since we assumed $\|\mathbf{u}_j\|_2 \le 1$ for all $1 \le j \le N$ and $\|(\mathbf{W}_Q^{(i)})^\top \mathbf{W}_K^{(i)}\|_2 \le \sqrt{d_h}$, we have that

$$\|\boldsymbol{g}_j\|_{\max} \le 1, \qquad \|\tilde{\boldsymbol{g}}_j\|_{\max} \le 1, \qquad 1 \le j \le L.$$

Denote by $g_j^t$ and $\tilde{g}_j^t$ the $t$th entry of $\boldsymbol{g}_j$ and $\tilde{\boldsymbol{g}}_j$, respectively. For every fixed $1 \le j \le L$ and $1 \le t \le L$, we have

$$|\underbrace{\text{softmax}(\boldsymbol{g}_j)^t - \text{softmax}(\tilde{\boldsymbol{g}}_j)^t}_{d_j^t}| = \left| \frac{\exp(g_j^t)}{\sum_{k=1}^L \exp(g_j^k)} - \frac{\exp(\tilde{g}_j^t)}{\sum_{k=1}^L \exp(\tilde{g}_j^k)} \right|$$

$$= \frac{\left| \sum_{k=1}^L \left( \exp(g_j^t)\exp(\tilde{g}_j^k) - \exp(\tilde{g}_j^t)\exp(g_j^k) \right) \right|}{\left| \left( \sum_{k=1}^L \exp(g_j^k) \right) \left( \sum_{k=1}^L \exp(\tilde{g}_j^k) \right) \right|} \le \frac{L \max_k \left| \exp(g_j^t)\exp(\tilde{g}_j^k) - \exp(\tilde{g}_j^t)\exp(g_j^k) \right|}{L^2 \exp(-2)}$$

$$= \frac{L \max_k \left| \exp(g_j^t)(\exp(g_j^k) + (\exp(\tilde{g}_j^k) - \exp(g_j^k))) - (\exp(g_j^t) + (\exp(\tilde{g}_j^t) - \exp(g_j^t)))\exp(g_j^k) \right|}{L^2 \exp(-2)}$$

$$\le \frac{\max_k \left( \exp(1) \left( |\exp(g_j^k) - \exp(\tilde{g}_j^k)| + |\exp(g_j^t) - \exp(\tilde{g}_j^t)| \right) + |\exp(g_j^k) - \exp(\tilde{g}_j^k)| |\exp(g_j^t) - \exp(\tilde{g}_j^t)| \right)}{L \exp(-2)}$$

$$\le \underbrace{\frac{2\exp(1)(2\sigma_{\tilde{d}+1}(\boldsymbol{\Xi}) + \sigma_{\tilde{d}+1}(\boldsymbol{\Xi})^2) + (2\sigma_{\tilde{d}+1}(\boldsymbol{\Xi}) + \sigma_{\tilde{d}+1}(\boldsymbol{\Xi})^2)^2}{L\exp(-2)}}_{D_Q},$$

$$\tag{14}$$

where the last inequality follows from eq. (13). Hence, for each fixed $1 \le j \le L$ and $1 \le i \le h$, we have that

$$\left\| \mathbf{W}_V^{(i)} \mathbf{U}\, \text{softmax}(\boldsymbol{g}_j^{(i)}) - \tilde{\mathbf{W}}_V^{(i)} \mathbf{U}\, \text{softmax}(\tilde{\boldsymbol{g}}_j^{(i)}) \right\|_2$$

$$= \left\| \sum_{i=1}^L \left( \mathbf{W}_V^{(i)} \mathbf{U}(:,i)\, \text{softmax}(g_j^t) - \tilde{\mathbf{W}}_V^{(i)} \mathbf{U}(:,i)\, \text{softmax}(\tilde{g}_j^t) \right) \right\|_2$$

$$\le \sqrt{L} \max_{1 \le t \le L} \left\| \left( \mathbf{W}_V^{(i)} \mathbf{U}(:,t)\, \text{softmax}(g_j^t) - (\mathbf{W}_V^{(i)} \mathbf{U}(:,t) - \boldsymbol{\Delta}_V^{(i)}(:,t)) (\text{softmax}(g_j^t) - d_j^t) \right) \right\|_2$$

$$\le \sqrt{L} \max_{1 \le t \le L} \left( \|\boldsymbol{\Delta}_V^{(i)}(:,t)\|_2 |\text{softmax}(\tilde{g}_j^t)| + \|\mathbf{W}_V^{(i)} \mathbf{U}(:,t)\|_2\, D_Q \right)$$

$$\le \sqrt{L} \left( \sqrt{d_h}\, \sigma_{\tilde{d}+1} \Theta(1/L) + \sqrt{d_h}\, \mathcal{O}\left( \left( \sigma_{\tilde{d}+1} + \sigma_{\tilde{d}+1}^2 \right)/L \right) \right) = \mathcal{O}\left( \sqrt{d_h}\, \sigma_{\tilde{d}+1}/\sqrt{L} \right),$$

where the last inequality follows from eq. (12) and (14). Since $\mathbf{W}_V^{(i)}\mathbf{U}\ \mathrm{softmax}(\boldsymbol{g}_j^{(i)}) - \tilde{\mathbf{W}}_V^{(i)}\mathbf{U}\,\mathrm{softmax}(\tilde{\boldsymbol{g}}_j^{(i)})$ is the $j$th column of

$$\mathrm{Attention}(\mathbf{U};\mathbf{W}_Q^{(i)},\mathbf{W}_K^{(i)},\mathbf{W}_V^{(i)}) - \mathrm{Attention}(\mathbf{U};\tilde{\mathbf{W}}_Q^{(i)},\tilde{\mathbf{W}}_K^{(i)},\tilde{\mathbf{W}}_V^{(i)}),$$

we have that

$$\|\mathrm{MH\text{-}Attention}(\mathbf{U};\mathbf{W}_Q,\mathbf{W}_K,\mathbf{W}_V,h) - \mathrm{Attention}(\mathbf{U};\tilde{\mathbf{W}}_Q,\tilde{\mathbf{W}}_K,\tilde{\mathbf{W}}_V,h)\|_F$$

$$= \sqrt{\sum_{i=1}^{h}\|\mathrm{Attention}(\mathbf{U};\mathbf{W}_Q^{(i)},\mathbf{W}_K^{(i)},\mathbf{W}_V^{(i)}) - \mathrm{Attention}(\mathbf{U};\tilde{\mathbf{W}}_Q^{(i)},\tilde{\mathbf{W}}_K^{(i)},\tilde{\mathbf{W}}_V^{(i)})\|_F^2}$$

$$\leq \sqrt{\sum_{i=1}^{h}\sum_{j=1}^{L}\left\|\mathbf{W}_V^{(i)}\mathbf{U}\,\mathrm{softmax}(\boldsymbol{g}_j^{(i)}) - \tilde{\mathbf{W}}_V^{(i)}\mathbf{U}\,\mathrm{softmax}(\tilde{\boldsymbol{g}}_j^{(i)})\right\|_2^2}$$

$$= \sqrt{hL}\mathcal{O}\left(\sqrt{d_h}\,\sigma_{\tilde{d}+1}/\sqrt{L}\right) = \mathcal{O}\left(\sqrt{d}\,\sigma_{\tilde{d}+1}\right).$$

The proof is complete. $\qquad\square$

Theorem 5 immediately proves the upper bound in Theorem 3. The lower bound needs a separate argument.

*Proof of Theorem 3.* The upper bound follows immediately from Theorem 5 by setting $h = 1$. To prove the lower bound, let $\mathbf{U}' \in \mathbb{R}^{d\times\tilde{d}}$ and $\mathbf{U} \in \mathbb{R}^{d\times L}$ be such that

$$\mathbf{U} = [\ \mathbf{U}'\ |\ \mathbf{0}\ ], \qquad \mathbf{U}' = \mathrm{diag}(\sigma_1,\ldots,\sigma_d) \in \mathbb{R}^{d\times d}, \quad \mathbf{0} \in \mathbb{R}^{d\times(L-d)}.$$

Clearly, the singular values of $\mathbf{U}$ are $\sigma_1,\ldots,\sigma_d$. Define $\mathbf{W}_Q$ and $\mathbf{W}_K$ such that

$$\mathbf{W}_Q^\top\mathbf{W}_K = \log(4d)\sigma_d^{-2}\sqrt{d}\,\mathbf{I}_d.$$

Then, we have

$$\mathbf{T} := \frac{\mathbf{U}^\top\mathbf{W}_Q^\top\mathbf{W}_K\mathbf{U}}{\sqrt{d}} = \log(4d)\sigma_d^{-2}\left[\begin{array}{c|c}\mathrm{diag}(\sigma_1^2,\ldots,\sigma_d^2) & \mathbf{0}_{d\times(L-d)} \\ \hline \mathbf{0}_{(L-d)\times d} & \mathbf{0}_{(L-d)\times(L-d)}\end{array}\right].$$

Let $\mathbf{G} = \mathrm{softmax}(\mathbf{T})$ and let $\boldsymbol{g}_j$ be the $j$th column of $\mathbf{G}$. Then, for every $1 \leq j \leq d$, we have

$$\boldsymbol{g}_j = [g_j^1,\ldots,g_j^L]^\top, \qquad g_j^i = \begin{cases} \dfrac{1}{(L-1) + \exp\left(\log(4d)\sigma_d^{-2}\sigma_j^2\right)} \leq \dfrac{1}{4d}, & i \neq j, \\[4mm] \dfrac{\exp\left(\log(4d)\sigma_d^{-2}\sigma_j^2\right)}{(L-1) + \exp\left(\log(4d)\sigma_d^{-2}\sigma_j^2\right)} \geq \dfrac{1}{2}, & i = j. \end{cases} \tag{15}$$

Write $\mathbf{G}$ into

$$\mathbf{G} = \left[\begin{array}{c|c}\mathbf{G}_{11} & \mathbf{G}_{12} \\ \hline \mathbf{G}_{21} & \mathbf{G}_{22}\end{array}\right],$$

where

$$\mathbf{G}_{11} = \mathbf{G}' \in \mathbb{R}^{d\times d}, \quad \mathbf{G}_{12} \in \mathbb{R}^{d\times(L-d)}, \quad \mathbf{G}_{21} \in \mathbb{R}^{(L-d)\times d}, \quad \mathbf{G}_{22} \in \mathbb{R}^{(L-d)\times(L-d)},$$

and write $\mathbf{G}_{11}$ into the sum of its diagonal part and off-diagonal part:

$$\mathbf{G}_{11} = \mathbf{G}_{\mathrm{diag}} + \mathbf{G}_{\mathrm{off\text{-}diag}},$$

where $\mathbf{G}_{\mathrm{diag}}$ is a diagonal matrix and $\mathbf{G}_{\mathrm{off\text{-}diag}}$ is a matrix with zero diagonal entries. Then, by eq. (15), we have

$$\sigma_d(\mathbf{G}_{\mathrm{diag}}) = \min_{1\leq j\leq d}g_j^j \geq \frac{1}{2}, \qquad \|\mathbf{G}_{\mathrm{off\text{-}diag}}\|_2 \leq \|\mathbf{G}_{\mathrm{off\text{-}diag}}\|_F \leq d\frac{1}{4d} = \frac{1}{4}.$$

Hence, by Weyl's inequality, we have

$$\sigma_d(\mathbf{G}_{11}) \geq \sigma_d(\mathbf{G}_{\text{diag}}) - \|\mathbf{G}_{\text{off-diag}}\|_2 \geq \frac{1}{4}. \tag{16}$$

Using the results above, we have

$$\mathbf{W}_V \mathbf{U} \operatorname{softmax}\left(\frac{\mathbf{U}^\top \mathbf{W}_Q^\top \mathbf{W}_K \mathbf{U}}{\sqrt{d}}\right) - \tilde{\mathbf{W}}_V \mathbf{U} \operatorname{softmax}\left(\frac{\mathbf{U}^\top \mathbf{W}_Q^\top \mathbf{W}_K \mathbf{U}}{\sqrt{d}}\right)$$

$$= (\mathbf{W}_V \mathbf{U} - \tilde{\mathbf{W}}_V \mathbf{U})\mathbf{G} = \left[\ \mathbf{W}_V \mathbf{U}' - \tilde{\mathbf{W}}_V \mathbf{U}' \ \middle| \ \mathbf{0}_{d\times(L-d)}\ \right] \left[\begin{array}{c|c} \mathbf{G}_{11} & \mathbf{G}_{12} \\ \hline \mathbf{G}_{21} & \mathbf{G}_{22} \end{array}\right]$$

$$= \left[\ \left(\mathbf{W}_V \mathbf{U}' - \tilde{\mathbf{W}}_V \mathbf{U}'\right)\mathbf{G}_{11} \ \middle| \ \left(\mathbf{W}_V \mathbf{U}' - \tilde{\mathbf{W}}_V \mathbf{U}'\right)\mathbf{G}_{12}\ \right],$$

but $\tilde{\mathbf{W}}_V \mathbf{U}'$ is a rank-$\tilde{d}$ matrix so we must have

$$\|\mathbf{W}_V \mathbf{U}' - \tilde{\mathbf{W}}_V \mathbf{U}'\|_2 \geq \sigma_{\tilde{d}+1}(\mathbf{W}_V \mathbf{U}') = \sigma_{\tilde{d}+1}. \tag{17}$$

Hence, by the singular value inqualities, we have

$$\left\|\mathbf{W}_V \mathbf{U} \operatorname{softmax}\left(\frac{\mathbf{U}^\top \mathbf{W}_Q^\top \mathbf{W}_K \mathbf{U}}{\sqrt{d}}\right) - \tilde{\mathbf{W}}_V \mathbf{U} \operatorname{softmax}\left(\frac{\mathbf{U}^\top \mathbf{W}_Q^\top \mathbf{W}_K \mathbf{U}}{\sqrt{d}}\right)\right\|_2$$

$$\geq \left\|\left(\mathbf{W}_V \mathbf{U}' - \tilde{\mathbf{W}}_V \mathbf{U}'\right)\mathbf{G}_{11}\right\|_2 \geq \sigma_1\left(\mathbf{W}_V \mathbf{U}' - \tilde{\mathbf{W}}_V \mathbf{U}'\right)\sigma_d(\mathbf{G}_{11}) \geq \frac{1}{4}\sigma_{\tilde{d}+1},$$

where the last inequality is obtained by combining eq. (16) and (17). The proof is complete. $\qquad\square$

## E    COMPRESSING A MULTI-HEAD ATTENTION LAYER

Theorem 3 concerns the compressibility of a single-head attention layer. In practice, most TSFMs use multi-head attention instead:[2]

$$\text{MH-Attention}(\mathbf{U}; \mathbf{W}_Q, \mathbf{W}_K, \mathbf{W}_V, h) = \begin{bmatrix} \text{Attention}(\mathbf{U}; \mathbf{W}_Q^{(1)}, \mathbf{W}_K^{(1)}, \mathbf{W}_V^{(1)}) \\ \vdots \\ \text{Attention}(\mathbf{U}; \mathbf{W}_Q^{(h)}, \mathbf{W}_K^{(h)}, \mathbf{W}_V^{(h)}) \end{bmatrix}, \tag{18}$$

$$\mathbf{W}_{Q/K/V} = \begin{bmatrix} (\mathbf{W}_{Q/K/V}^{(1)})^\top & \cdots & (\mathbf{W}_{Q/K/V}^{(h)})^\top \end{bmatrix}^\top, \quad \mathbf{W}_{Q/K/V}^{(i)} \in \mathbb{R}^{d_h \times d}, \quad d_h = d/h.$$

This leads to the following question: does the number of heads have an effect on the numerical ranks of the attention matrices in trained Chronos models? The answer to this is positive, but there are some important subtleties. If we look at the left panel of Figure 9, then we see that if we pretrain Chronos models with fixed hidden dimension $d = 1024$ and only increase the number of heads, the numerical rank of attention matrices increases. At first, it may seem that this happens because a multi-head attention is less compressible than a single-head one, but this is not the case. It is straightforward to show that Theorem 3 holds as well for multi-head attentions (see Theorem 5 in Appendix D). In particular, if $\Xi$ has an algebraic rank of $\tilde{d}$, then all multi-head attention matrices can be compressed to rank-$\tilde{d}$ without affecting the output.

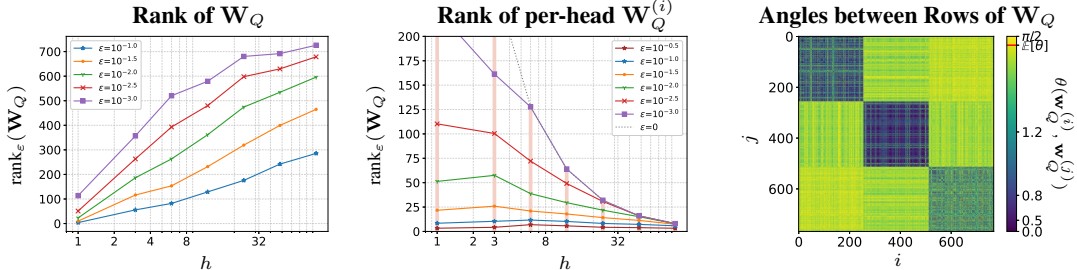

**Figure 9:** The left panel shows the averaged $\varepsilon$-rank of query projection matrices $\mathbf{W}_Q$ in pretrained Chronos models with varying number of heads. The middle panel shows the averaged $\varepsilon$-rank of query projection submatrices $\mathbf{W}_Q^{(i)}$ for every head. The right panel shows the angle between every pair of rows of $\mathbf{W}_Q$ in the first layer of a 3-head pretrained Chronos model.

If Theorem 5 holds for multi-head attention, then why does a multi-head attention exhibit a higher numerical rank? To understand this, we need to understand the mechanism of a low-rank weight matrix. We use $\mathbf{W}_Q$ for illustration, and the same analogous applies to $\mathbf{W}_K$ and $\mathbf{W}_V$. Let $\mathbf{W}_Q \in \mathbb{R}^{d \times d}$ be approximated by a rank-$\tilde{d}$ matrix, i.e., $\mathbf{W}_Q \approx \mathbf{W}_1 \mathbf{W}_2$, where $\mathbf{W}_1 \in \mathbb{R}^{d \times \tilde{d}}$ and $\mathbf{W}_2 \in \mathbb{R}^{\tilde{d} \times d}$. We view the action $\mathbf{W}_Q \mathbf{U}$, which maps every column of $\mathbf{U}$ from $\mathbb{R}^d$ into $\mathbb{R}^d$, as a two-step process. First, we multiply $\mathbf{W}_2$ to $\mathbf{U}$ to drop the columns of $\mathbf{U}$ from $\mathbb{R}^d$ to $\mathbb{R}^{\tilde{d}}$. In numerical linear algebra, this is known as "sketching," (Drineas & Mahoney, 2016) i.e., $\mathbf{W}_2$ "sketches" a $\tilde{d}$-dimensional subspace $R(\mathbf{W}_2 \mathbf{U})$ in $\mathbb{R}^L$, the row space of $\mathbf{W}_2 \mathbf{U}$, from a $d$-dimensional subspace $R(\mathbf{U})$ in $\mathbb{R}^L$.[3] If $\sigma_{\tilde{d}+1}(\mathbf{U})$ is small, then there exists a sketching matrix $\mathbf{W}_2$ so that $R(\mathbf{W}_2 \mathbf{U}) \approx R(\mathbf{U}) \supset R(\mathbf{W}_Q \mathbf{U})$; hence, we can apply a matrix $\mathbf{W}_1$ to lift the columns of $\mathbf{W}_2 \mathbf{U}$ from $\mathbb{R}^{\tilde{d}}$ back to $\mathbb{R}^d$ and have $\mathbf{W}_1(\mathbf{W}_2 \mathbf{U}) \approx \mathbf{W}_Q \mathbf{U}$.

If we apply a low-rank approximation to a multi-head attention matrix $\mathbf{W}_Q$, then we obtain:

$$\begin{bmatrix} \mathbf{W}_Q^{(1)} \\ \vdots \\ \mathbf{W}_Q^{(h)} \end{bmatrix} = \mathbf{W}_Q \approx \mathbf{W}_1 \mathbf{W}_2 = \begin{bmatrix} \mathbf{W}_1^{(1)} \\ \vdots \\ \mathbf{W}_1^{(h)} \end{bmatrix} \mathbf{W}_2 = \begin{bmatrix} \mathbf{W}_1^{(1)} \mathbf{W}_2 \\ \vdots \\ \mathbf{W}_1^{(h)} \mathbf{W}_2 \end{bmatrix}, \quad \mathbf{W}_1^{(i)} \in \mathbb{R}^{d_h \times \tilde{d}}. \tag{19}$$

---

[2]In eq. (18), the $1/\sqrt{d}$ normalization factor in an attention is changed into $1/\sqrt{d_h}$.

[3]In numerical linear algebra (NLA), sketching is usually considered as an operation on the column space. To adapt it to our framework, we sketch the row space instead. This is achieved by taking the transpose of the matrix $\mathbf{U}$ and sketching its column space as usually done in NLA.

When we apply $\mathbf{W}_Q$ to $\mathbf{U}$, for each head $1 \leq i \leq h$, $\mathbf{W}_2$ sketches a $\tilde{d}$-dimensional subspace from the row space of $\mathbf{U}$ while $\mathbf{W}_1^{(i)}$ forms $d_h$ rows from this $\tilde{d}$-dimensional space. This makes the problem clear: in an "optimal" low-rank approximation of $\mathbf{W}_Q$, the sketching matrix $\mathbf{W}_2$ is shared across all heads. Since each head is independent from the others in a multi-head attention, there is no guarantee that, in a pretrained model, the sketching matrices will be shared. In the right panel of Figure 9, we visualize the angles (see eq. (2)) between every pair of rows of $\mathbf{W}_Q$. If $\mathbf{W}_2$ is shared across all heads, then all rows of $\mathbf{W}_Q$ should exhibit some linear dependency. We only see low-rank structures within each head $\mathbf{W}_Q^{(i)}$ (i.e., the dark $d_h \times d_h$ blocks along the diagonal). This shows that instead of eq. (19), $\mathbf{W}_Q$ from a pretrained Chronos model looks more like

$$\begin{bmatrix} \mathbf{W}_Q^{(1)} \\ \vdots \\ \mathbf{W}_Q^{(h)} \end{bmatrix} = \mathbf{W}_Q \approx \begin{bmatrix} \mathbf{W}_1^{(1)} \mathbf{W}_2^{(1)} \\ \vdots \\ \mathbf{W}_1^{(h)} \mathbf{W}_2^{(h)} \end{bmatrix}, \qquad \mathbf{W}_1^{(i)} \in \mathbb{R}^{d_h \times \tilde{d}}, \qquad \mathbf{W}_2^{(i)} \in \mathbb{R}^{\tilde{d} \times d}, \qquad (20)$$

which leverages a head-dependent sketching $\mathbf{W}_2^{(i)}$. The rank of the right-hand side matrix in eq. (20), which consists of $h$ rank-$\tilde{d}$ submatrices, can be as large as $h\tilde{d}$. This finding explains the phenomenon we observed at the beginning of this section, i.e., while the numerical rank $\tilde{d}$ of the input stays the same, the rank of the attention matrices increases with the number of heads $h$. The middle panel of Figure 9 confirms this numerically. We see that the numerical rank of the projection matrix $\mathbf{W}_Q^{(i)}$ in each head remains relatively constant for reasonably large $\varepsilon$ as we increase the number of heads.

Our observation in Figure 9 is empirical and explanatory: it says in a pretrained TSFM, the attention matrices $\mathbf{W}_{Q/K/V}$ are essentially doing the head-dependent sketching (see eq. (20)). From a methodological point of view, if we train a compressed model (i.e., the one that is paramaterized by $\mathbf{W}_1$ and $\mathbf{W}_2$ instead of $\mathbf{W}_Q$) from scratch, should we adopt a parameter-efficient head-independent sketching in eq. (19) or a head-dependent one, which requires more parameters? If we know the numerical rank $\tilde{d}$ of $\boldsymbol{\Xi}$ so that $\sigma_{\tilde{d}+1}(\boldsymbol{\Xi})$ is tiny, then from an approximation theory perspective, there is little loss using eq. (19). The only problem is: we do not know $\tilde{d}$ a priori. If we choose $\tilde{d}$ small enough that $\sigma_{\tilde{d}+1}(\boldsymbol{\Xi})$ is still relatively large, then the following theorem (see Appendix E for the proof) shows the benefit of using a head-dependent sketching $\mathbf{W}_2^{(h)}$ in eq. (20).

**Theorem 6.** Fix some $d = h \times d_h$ for two positive integers $h$ and $d_h$ and $L \geq d$. Let $C \geq 1$ be any stability bound. Let $1 \geq \sigma_1 \geq \sigma_2 \geq \cdots \geq \sigma_d > 0$ be any fixed sequence. There exist an input $\mathbf{U} \in \mathbb{R}^{d \times L}$ with $\sigma_j(\mathbf{U}) = \sigma_j$ for all $1 \leq j \leq d$ and three matrices $\mathbf{W}_Q, \mathbf{W}_K, \mathbf{W}_V \in \mathbb{R}^{d \times d}$, such that the following statements hold for any $\tilde{d} < d_h$:

1. Given any rank-$\tilde{d}$ matrix $\tilde{\mathbf{W}}_V \in \mathbb{R}^{d \times d}$ with $\|\tilde{\mathbf{W}}_V\|_F \leq C\|\mathbf{W}_V\|_F$, we have

$$\left\| \text{MH-Attention}(\mathbf{U}; \mathbf{W}_Q, \mathbf{W}_K, \mathbf{W}_V, h) - \text{MH-Attention}(\mathbf{U}; \mathbf{W}_Q, \mathbf{W}_K, \tilde{\mathbf{W}}_V, h) \right\|_2 \geq \frac{1}{2}\sigma_{\tilde{d}+1}.$$

2. There exist low-rank matrices $\tilde{\mathbf{W}}_Q, \tilde{\mathbf{W}}_K, \tilde{\mathbf{W}}_V \in \mathbb{R}^{d \times d}$ with $\|\tilde{\mathbf{W}}_Q\|_F \leq \|\mathbf{W}_Q\|_F$, $\|\tilde{\mathbf{W}}_K\|_F \leq \|\mathbf{W}_K\|_F$, and $\|\tilde{\mathbf{W}}_V\|_F \leq \|\mathbf{W}_V\|_F$, such that

$$\tilde{\mathbf{W}}_{Q/K/V} = \begin{bmatrix} \mathbf{W}_{Q/K/V,1}^{(1)} \mathbf{W}_{Q/K/V,2}^{(1)} \\ \hline \vdots \\ \hline \mathbf{W}_{Q/K/V,1}^{(h)} \mathbf{W}_{Q/K/V,2}^{(h)} \end{bmatrix}, \quad \mathbf{W}_{Q/K/V,1}^{(i)} \in \mathbb{R}^{d_h \times (\tilde{d}+1)}, \quad \mathbf{W}_{Q/K/V,2}^{(i)} \in \mathbb{R}^{(\tilde{d}+1) \times d},$$

where every row of $\mathbf{W}_{Q/K/V,2}^{(i)}$, except the last row of $\mathbf{W}_{K,2}^{(i)}$, contains at most $d_h$ non-zero entries, and satisfies that

$$\left\| \text{MH-Attention}(\mathbf{U}; \mathbf{W}_Q, \mathbf{W}_K, \mathbf{W}_V, h) - \text{MH-Attention}(\mathbf{U}; \tilde{\mathbf{W}}_Q, \tilde{\mathbf{W}}_K, \tilde{\mathbf{W}}_V, h) \right\|_2 \leq 4\sqrt{h}\sigma_{h\tilde{d}+1}.$$

Theorem 6 highlights an "error-correcting" mechanism of the head-dependent low-rank design in eq. (20) in the following sense: if we choose a $\tilde{d}$ where $\sigma_{\tilde{d}+1}$ is large, then the first statement

shows that the shared sketching in eq. (19) inevitably limits the expressiveness of the reduced multi-head attention layer. Using a head-dependent low-rank representation relaxes the error to the order of $\sigma_{h\tilde{d}+1}$, which can be significantly smaller than $\sigma_{\tilde{d}+1}$ when the number of heads $h$ is large and the singular values decay fast. While using eq. (20) requires a larger number of parameters, Theorem 6 suggests the potential of a sparse parameterization of the sketching matrices $\mathbf{W}^{(i)}_{Q/K/V,2}$. We leave this as a promising future direction.

### E.1 Proof of Theorem 6

We prove that a sparse but head-dependent sketching performs better in theory than a head-independent sketching, especially when $\tilde{d}$ is lower than the numerical rank of the input matrix. The intuition is that if $\tilde{d}$ is too small, then the sketching performed by a single sketching matrix must be bad, but if we use head-dependent sketchings, then we can leverage multiple of them to still obtain a lot from the input matrix.

*Proof of Theorem 6.* Set $\mathbf{W}_V = \mathbf{I}_d$. Interleave the singular values $\sigma_1, \dots, \sigma_d$ as follows:

$$\boldsymbol{\sigma}^{(1)} = \left(\sigma_1^{(1)}, \sigma_2^{(1)} \dots, \sigma_{d_h}^{(1)}\right) = \left(\sigma_1, \sigma_{h+1}, \dots, \sigma_{(d_h-1)h+1}\right),$$

$$\boldsymbol{\sigma}^{(2)} = \left(\sigma_1^{(2)}, \sigma_2^{(2)} \dots, \sigma_{d_h}^{(2)}\right) = \left(\sigma_2, \sigma_{h+2}, \dots, \sigma_{(d_h-1)h+2}\right),$$

$$\vdots$$

$$\boldsymbol{\sigma}^{(h)} = \left(\sigma_1^{(h)}, \sigma_2^{(h)} \dots, \sigma_{d_h}^{(h)}\right) = \left(\sigma_h, \sigma_{2h}, \dots, \sigma_{d_h \cdot h}\right).$$

Define the input matrix by

$$\mathbf{U} = \left[\ \underbrace{\operatorname{diag}(\operatorname{diag}(\boldsymbol{\sigma}^{(1)}), \dots, \operatorname{diag}(\boldsymbol{\sigma}^{(h)}))}_{\mathbf{U}_D}\ \middle|\ \mathbf{0}_{d\times(L-d)}\ \right].$$

For every $1 \le i \le h$ and $1 \le j \le d_h$, let $\mathbf{v}_j^{(i)} \in \mathbb{R}^{d_h}$ be a unit vector satisfying that

$$\left\|\mathbf{v}_j^{(i)\top}\mathbf{v}_{j'}^{(i')}\right\|_2 < 1, \qquad (i,j) \ne (i',j').$$

Then, consider the following matrix:

$$\overline{\mathbf{W}}_Q = \overline{\mathbf{W}}_K = \left[\ \mathbf{v}_1^{(1)}/\sigma_1^{(1)} \quad \cdots \quad \mathbf{v}_{d_h}^{(1)}/\sigma_{d_h}^{(1)} \ \middle|\ \cdots \ \middle|\ \mathbf{v}_1^{(h)}/\sigma_1^{(h)} \quad \cdots \quad \mathbf{v}_{d_h}^{(h)}/\sigma_{d_h}^{(h)}\ \right] \in \mathbb{R}^{d_h \times d}.$$

Then, by our construction, the matrix $\mathbf{U}^\top \overline{\mathbf{W}}_Q^\top \overline{\mathbf{W}}_K \mathbf{U}$ satisfies that

$$\left(\mathbf{U}^\top \overline{\mathbf{W}}_Q^\top \overline{\mathbf{W}}_K \mathbf{U}\right)(j,j) = 1, \qquad \left|\left(\mathbf{U}^\top \overline{\mathbf{W}}_Q^\top \overline{\mathbf{W}}_K \mathbf{U}\right)(i,j)\right| < 1, \qquad i \ne j.$$

Set

$$\varepsilon = \min\left\{(1+C)^{-1}\sigma_{\tilde{d}+1}/2,\ \min_{1\le i\le h}\sigma_{\tilde{d}+1}^{(i)}\big/\left(2\,\|\mathbf{W}_V\mathbf{U}\|_2 + 2\sigma_1\right), 1\right\}.$$

By choosing a sufficiently large $\alpha > 0$, we guarantee that

$$\left\|\underbrace{\operatorname{softmax}\left(\frac{\mathbf{U}^\top(\alpha\overline{\mathbf{W}}_Q^\top)(\alpha\overline{\mathbf{W}}_K)\mathbf{U}}{\sqrt{d_h}}\right)}_{\mathbf{G}} - \underbrace{\left[\begin{array}{c|c} \mathbf{I_d} & L^{-1}\mathbf{1}_{d\times(L-d)} \\ \hline \mathbf{0}_{(L-d)\times d} & L^{-1}\mathbf{1}_{(L-d)\times(L-d)} \end{array}\right]}_{\tilde{\mathbf{G}}}\right\|_F \le \varepsilon. \qquad (21)$$

Let query and key matrices $\mathbf{W}_Q$ and $\mathbf{W}_K$ be defined in eq. (18) using submatrices $\mathbf{W}_Q^{(i)} = \alpha\overline{\mathbf{W}}_Q$ and $\mathbf{W}_K^{(i)} = \alpha\overline{\mathbf{W}}_K$, respectively, for all $1 \le i \le h$. We use these matrices $\mathbf{W}_Q, \mathbf{W}_K, \mathbf{W}_V$, and $\mathbf{U}$ to prove the two claims.

**Claim 1: No low-rank parameterization gets to an accuracy below $\sigma_{\tilde{d}+1}$.** Let $\tilde{\mathbf{W}}_V \in \mathbb{R}^{d \times d}$ be any rank-$\tilde{d}$ matrix. We can write the approximation error as

$$\left\| \text{MH-Attention}(\mathbf{U}; \mathbf{W}_Q, \mathbf{W}_K, \mathbf{W}_V, h) - \text{MH-Attention}(\mathbf{U}; \mathbf{W}_Q, \mathbf{W}_K, \tilde{\mathbf{W}}_V, h) \right\|_2$$

$$= \left\| \begin{bmatrix} \mathbf{W}_V^{(1)}\mathbf{U}\mathbf{G} \\ \vdots \\ \hline \mathbf{W}_V^{(h)}\mathbf{U}\mathbf{G} \end{bmatrix} - \begin{bmatrix} \tilde{\mathbf{W}}_V^{(1)}\mathbf{U}\mathbf{G} \\ \vdots \\ \hline \tilde{\mathbf{W}}_V^{(h)}\mathbf{U}\mathbf{G} \end{bmatrix} \right\|_2$$

$$\geq \underbrace{\left\| \begin{bmatrix} \mathbf{W}_V^{(1)}\mathbf{U}\tilde{\mathbf{G}} \\ \vdots \\ \hline \mathbf{W}_V^{(h)}\mathbf{U}\tilde{\mathbf{G}} \end{bmatrix} - \begin{bmatrix} \tilde{\mathbf{W}}_V^{(1)}\mathbf{U}\tilde{\mathbf{G}} \\ \vdots \\ \hline \mathbf{W}_V^{(h)}\mathbf{U}\tilde{\mathbf{G}} \end{bmatrix} \right\|_2}_{E_{\text{lead}}} - \underbrace{\left\| \begin{bmatrix} (\mathbf{W}_V^{(1)} - \tilde{\mathbf{W}}_V^{(1)})\mathbf{U}(\mathbf{G} - \tilde{\mathbf{G}}) \\ \vdots \\ \hline (\mathbf{W}_V^{(h)} - \tilde{\mathbf{W}}_V^{(h)})\mathbf{U}(\mathbf{G} - \tilde{\mathbf{G}}) \end{bmatrix} \right\|_2}_{E_{\text{trail}}}.$$

To control $E_{\text{trail}}$, we note that

$$E_{\text{trail}} \leq \max_{1 \leq i \leq h} \left\| (\mathbf{W}_V^{(i)} - \tilde{\mathbf{W}}_V^{(i)})\mathbf{U}(\mathbf{G} - \tilde{\mathbf{G}}) \right\|_2 \leq \max_{1 \leq i \leq h} \left( \left\| \mathbf{W}_V^{(i)} \right\|_2 + \left\| \tilde{\mathbf{W}}_V^{(i)} \right\|_2 \right) \left\| \mathbf{G} - \tilde{\mathbf{G}} \right\|_2$$

$$\leq (1 + C)\varepsilon = \frac{1}{2}\sigma_{\tilde{d}+1}. \tag{22}$$

To control $E_{\text{lead}}$, we note that

$$\begin{bmatrix} (\mathbf{W}_V^{(1)} - \tilde{\mathbf{W}}_V^{(1)})\mathbf{U}\tilde{\mathbf{G}} \\ \vdots \\ \hline (\mathbf{W}_V^{(h)} - \tilde{\mathbf{W}}_V^{(h)})\mathbf{U}\tilde{\mathbf{G}} \end{bmatrix} = \begin{bmatrix} (\mathbf{W}_V^{(1)} - \tilde{\mathbf{W}}_V^{(1)})\left[ \mathbf{U}_D \mid L^{-1}\mathbf{U}_D\mathbf{1}_{d \times (L-d)} \right] \\ \vdots \\ \hline (\mathbf{W}_V^{(h)} - \tilde{\mathbf{W}}_V^{(h)})\left[ \mathbf{U}_D \mid L^{-1}\mathbf{U}_D\mathbf{1}_{d \times (L-d)} \right] \end{bmatrix}.$$

Hence, we have

$$E_{\text{lead}} = \left\| \begin{bmatrix} (\mathbf{W}_V^{(1)} - \tilde{\mathbf{W}}_V^{(1)})\mathbf{U}\tilde{\mathbf{G}} \\ \vdots \\ \hline (\mathbf{W}_V^{(h)} - \tilde{\mathbf{W}}_V^{(h)})\mathbf{U}\tilde{\mathbf{G}} \end{bmatrix} \right\|_2 \geq \left\| \begin{bmatrix} (\mathbf{W}_V^{(1)} - \tilde{\mathbf{W}}_V^{(1)})\mathbf{U}_D \\ \vdots \\ \hline (\mathbf{W}_V^{(h)} - \tilde{\mathbf{W}}_V^{(h)})\mathbf{U}_D \end{bmatrix} \right\|_2$$

$$= \left\| \mathbf{W}_V\mathbf{U}_D - \tilde{\mathbf{W}}_V\mathbf{U}_D \right\|_2 \geq \sigma_{\tilde{d}+1}, \tag{23}$$

where the last inequality follows from the fact that the singular values of $\mathbf{W}_V\mathbf{U}_D$ are exactly $\sigma_1, \ldots, \sigma_d$ and $\tilde{\mathbf{W}}_V\mathbf{U}_D$ is a matrix whose rank is at most $\tilde{d}$. Combining eq. (22) and (23), we have

$$\left\| \text{MH-Attention}(\mathbf{U}; \mathbf{W}_Q, \mathbf{W}_K, \mathbf{W}_V, h) - \text{MH-Attention}(\mathbf{U}; \mathbf{W}_Q, \mathbf{W}_K, \tilde{\mathbf{W}}_V, h) \right\|_2$$

$$\geq E_{\text{lead}} - E_{\text{trail}} \geq \frac{1}{2}\sigma_{\tilde{d}+1}.$$

**Claim 2: A sparse parameterization gets to an accuracy below $\sigma_{h\tilde{d}+1}$.** For every $1 \leq i \leq h$, define $\mathbf{W}_{V,L}^{(i)} \in \mathbb{R}^{d_h \times \tilde{d}}$ and $\mathbf{W}_{V,R}^{(i)} \in \mathbb{R}^{\tilde{d} \times d}$ as follows:

$$\mathbf{W}_{V,L}^{(i)} = \begin{bmatrix} \mathbf{I}_{\tilde{d}} \\ \mathbf{0}_{(d_h - \tilde{d}) \times \tilde{d}} \end{bmatrix}, \qquad \mathbf{W}_{V,R}^{(i)} = \begin{bmatrix} \mathbf{0}_{\tilde{d} \times d_h(i-1)} \mid \mathbf{I}_{\tilde{d}} \mid \mathbf{0}_{\tilde{d} \times ((d_h - \tilde{d}) + d_h(h-i))} \end{bmatrix}.$$

Then, we have

$$\mathbf{W}_{V,L}^{(i)}\mathbf{W}_{V,R}^{(i)}\mathbf{U} = \begin{bmatrix} \mathbf{0}_{\tilde{d} \times d_h(i-1)} & \text{diag}\left( \sigma_1^{(i)}, \ldots, \sigma_{\tilde{d}}^{(i)} \right) & \mathbf{0}_{\tilde{d} \times ((d_h - \tilde{d}) + d_h(h-i))} \\ \hline \mathbf{0}_{(d_h - \tilde{d}) \times d_h(i-1)} & \mathbf{0}_{(d_h - \tilde{d}) \times \tilde{d}} & \mathbf{0}_{(d_h - \tilde{d}) \times ((d_h - \tilde{d}) + d_h(h-i))} \end{bmatrix},$$

while

$$\mathbf{W}_V^{(i)}\mathbf{U} = \begin{bmatrix} \mathbf{0}_{\tilde{d} \times d_h(i-1)} & \text{diag}\left( \sigma_1^{(i)}, \ldots, \sigma_{d_h}^{(i)} \right) & \mathbf{0}_{\tilde{d} \times d_h(h-i)} \end{bmatrix}.$$

Hence, we have that

$$\|\mathbf{W}_V^{(i)}\mathbf{U} - \mathbf{W}_{V,L}^{(i)}\mathbf{W}_{V,R}^{(i)}\mathbf{U}\|_2 = \sigma_{\tilde{d}+1}^{(i)}. \tag{24}$$

Similarly, let $\mathbf{W}_{Q,L}^{(i)}, \mathbf{W}_{K,L}^{(i)} \in \mathbb{R}^{d_h \times (\tilde{d}+1)}$ and $\mathbf{W}_{Q,R}^{(i)}, \mathbf{W}_{K,R}^{(i)} \in \mathbb{R}^{(\tilde{d}+1) \times d}$ be given by

$$\mathbf{W}_{Q,L}^{(i)} = \mathbf{W}_{K,L}^{(i)} = \alpha \left[ \begin{array}{cccc} \mathbf{v}_1^{(i)}/\sigma_1^{(i)} & \cdots & \mathbf{v}_{\tilde{d}}^{(i)}/\sigma_{\tilde{d}}^{(i)} & \mathbf{v}_1^{(i')} \end{array} \right], \qquad i' = \mathrm{mod}(i, h) + 1,$$

$$\mathbf{W}_{Q,R}^{(i)} = \begin{cases} \left[ \begin{array}{c|c} \mathbf{I}_{\tilde{d}} & \mathbf{0}_{d-\tilde{d}} \\ \hline \mathbf{0}_{1\times\tilde{d}} & [1/\sigma_{\tilde{d}+1}, 0, \ldots, 0] \end{array} \right], & i = 1, \\[2.5em] \left[ \begin{array}{c|c|c} \mathbf{0}_{\tilde{d}\times d_h(i-1)} & \mathbf{I}_{\tilde{d}} & \mathbf{0}_{\tilde{d}\times((d_h-\tilde{d})+d_h(h-i))} \\ \hline [1/\sigma_1, 0, \ldots, 0] & \mathbf{0}_{1\times\tilde{d}} & \mathbf{0}_{1\times((d_h-\tilde{d})+d_h(h-i))} \end{array} \right], & i > 1, \end{cases}$$

$$\mathbf{W}_{K,R}^{(i)} = \left[ \begin{array}{c|c|c} \mathbf{0}_{\tilde{d}\times d_h(i-1)} & \mathbf{I}_{\tilde{d}} & \mathbf{0}_{\tilde{d}\times((d_h-\tilde{d})+d_h(h-i))} \\ \hline [1/\sigma_1, \ldots, 1/\sigma_{d_h(i-1)}] & \mathbf{0}_{1\times\tilde{d}} & [1/\sigma_{d_h(i-1)+\tilde{d}+1}, \ldots, 1/\sigma_d] \end{array} \right].$$

Note that since we have $\|\mathbf{v}_j^{(i)}\|_2$ for every $1 \le i \le h$ and $1 \le j \le d_h$, we clearly have that $\|\mathbf{W}_{Q,L}^{(i)}\mathbf{W}_{Q,R}^{(i)}\|_F \le \|\alpha\overline{\mathbf{W}}_Q\|_F$ and $\|\mathbf{W}_{K,L}^{(i)}\mathbf{W}_{K,R}^{(i)}\|_F = \|\alpha\overline{\mathbf{W}}_K\|_F$ for every $1 \le i \le h$. Hence, we have $\|\tilde{\mathbf{W}}_Q\|_F \le \|\mathbf{W}_Q\|_F$ and $\|\tilde{\mathbf{W}}_K\|_F = \|\mathbf{W}_K\|_F$. From now on, we only argue for the first head, i.e., $i = 1$. The rest follows easily from symmetry. Set $i = 1$, we have

$$\left( \underbrace{\mathbf{U}^\top {\mathbf{W}_{Q,R}^{(1)}}^\top {\mathbf{W}_{Q,L}^{(1)}}^\top \mathbf{W}_{K,L}^{(1)} \mathbf{W}_{K,R}^{(1)} \mathbf{U}}_{\tilde{\mathbf{T}}^{(1)}} \right)(j,k) = \alpha^2 \begin{cases} {\mathbf{v}_j^{(1)}}^\top \mathbf{v}_k^{(1)}, & 1 \le j \le \tilde{d}, & 1 \le k \le \tilde{d}, \\ {\mathbf{v}_j^{(1)}}^\top \mathbf{v}_1^{(2)}, & 1 \le j \le \tilde{d}, & \tilde{d} < k \le d, \\ {\mathbf{v}_1^{(2)}}^\top \mathbf{v}_k^{(1)}, & j = \tilde{d}+1, & 1 \le k \le \tilde{d}, \\ {\mathbf{v}_1^{(2)}}^\top \mathbf{v}_1^{(2)}, & j = \tilde{d}+1, & \tilde{d} < k \le d, \\ 0, & \text{otherwise.} \end{cases}$$

That is, for $1 \le j \le d$, the $j$th column of $\tilde{\mathbf{T}}^{(1)}$ has exactly one entry that equals 1, which is the $j$th element for $1 \le j \le \tilde{d}$ and the $(j+1)$st element for $j > \tilde{d}$. Moreover, for $j > d$, the $j$th column of $\tilde{\mathbf{T}}^{(1)}$ is zero. Hence, from the definition of $\alpha$, we have

$$\left\| \mathrm{softmax}\left(\tilde{\mathbf{T}}^{(1)}\right)(1:\tilde{d},:) - \mathbf{G}(1:\tilde{d},:) \right\|_2 \le \left\| \mathrm{softmax}\left(\tilde{\mathbf{T}}^{(1)}\right)(1:\tilde{d},:) - \tilde{\mathbf{G}}(1:\tilde{d},:) \right\|_2 + \left\| \tilde{\mathbf{G}} - \mathbf{G} \right\|_2 \le 2\varepsilon. \tag{25}$$

Combine eq. (24) and (25). We have

$$\left\| \mathbf{W}_V^{(1)}\mathbf{U}\mathbf{G} - \mathbf{W}_{V,L}^{(1)}\mathbf{W}_{V,R}^{(1)}\mathbf{U}\,\mathrm{softmax}\left(\tilde{\mathbf{T}}^{(1)}\right) \right\|_2$$

$$\le \left\| \mathbf{W}_V^{(1)}\mathbf{U}\mathbf{G} - \mathbf{W}_{V,L}^{(1)}\mathbf{W}_{V,R}^{(1)}\mathbf{U}\mathbf{G} \right\|_2 + \left\| \mathbf{W}_{V,L}^{(1)}\mathbf{W}_{V,R}^{(1)}\mathbf{U}\mathbf{G} - \mathbf{W}_{V,L}^{(1)}\mathbf{W}_{V,R}^{(1)}\mathbf{U}\,\mathrm{softmax}\left(\tilde{\mathbf{T}}^{(1)}\right) \right\|_2$$

$$= \left\| \mathbf{W}_V^{(1)}\mathbf{U}\mathbf{G} - \mathbf{W}_{V,L}^{(1)}\mathbf{W}_{V,R}^{(1)}\mathbf{U}\mathbf{G} \right\|_2 + \left\| \mathbf{W}_{V,L}^{(1)}\mathbf{W}_{V,R}^{(1)}\mathbf{U}\left(\mathbf{G} - \mathrm{softmax}\left(\tilde{\mathbf{T}}^{(1)}\right)\right)(1:\tilde{d},:) \right\|_2$$

$$\le \left\| \mathbf{W}_V^{(1)}\mathbf{U} - \mathbf{W}_{V,L}^{(1)}\mathbf{W}_{V,R}^{(1)}\mathbf{U} \right\|_2 \|\mathbf{G}\|_2 + \left( \left\| \mathbf{W}_V^{(1)}\mathbf{U} \right\|_2 + \sigma_{\tilde{d}+1}^{(1)} \right) \left\| \left(\mathbf{G} - \mathrm{softmax}\left(\tilde{\mathbf{T}}^{(1)}\right)\right)(1:\tilde{d},:) \right\|_2$$

$$\le \sigma_{\tilde{d}+1}^{(1)}(2 + \varepsilon) + 2\varepsilon \left\| \mathbf{W}_{V,L}^{(1)}\mathbf{W}_{V,R}^{(1)}\mathbf{U} \right\|_2 \le 4\sigma_{\tilde{d}+1}^{(1)} \le 4\sigma_{h\tilde{d}+1},$$

where the first inequality follows from the triangle inequality, the first equation follows from the sparsity of $\mathbf{W}_{V,R}^{(1)}$, the second inequality follows from the sub-multiplicity of the spectral norm and eq. (24), the third inequality follows from eq. (24) and eq. (25), the fifth inequality follows from the definition of $\varepsilon$, and the last inequality follows from the definition of $\sigma_{\tilde{d}+1}^{(1)}$. Notably, this inequality holds for every head $1 \le i \le h$. Hence, we have

$$\left\| \mathrm{MH\text{-}Attention}(\mathbf{U}; \mathbf{W}_Q, \mathbf{W}_K, \mathbf{W}_V, h) - \mathrm{MH\text{-}Attention}(\mathbf{U}; \mathbf{W}_Q, \mathbf{W}_K, \tilde{\mathbf{W}}_V, h) \right\|_2$$

$$\le \sqrt{h} \max_{1 \le i \le h} \left\| \mathbf{W}_V^{(i)}\mathbf{U}\mathbf{G} - \mathbf{W}_{V,L}^{(i)}\mathbf{W}_{V,R}^{(i)}\mathbf{U}\,\mathrm{softmax}\left(\tilde{\mathbf{T}}^{(i)}\right) \right\|_2 \le 4\sqrt{h}\,\sigma_{h\tilde{d}+1}.$$

The proof is complete. $\qquad\square$

## F    PROOF OF THEOREM 4 AND COROLLARY 1

In this section, we prove the idea of flow-of-ranks. The upper bound is proved via a straightforward application of singular value inequalities while the lower bound requires a careful construction.

*Proof of Theorem 4.*  We prove the two statements separately.

**The upper bound.** Fix a head index $1 \leq i \leq h$ and let

$$\mathbf{Y}^{(i)} = \mathbf{W}_V^{(i)}\mathbf{U}\,\text{softmax}\left(\mathbf{T}^{(i)}\right), \qquad \mathbf{T}^{(i)} = \frac{\mathbf{U}^\top \mathbf{W}_Q^{(i)\top} \mathbf{W}_K^{(i)} \mathbf{U}}{\sqrt{d_h}}.$$

Then, every entry of $\mathbf{T}^{(i)}$ has a magnitude no greater than 1 because every column of $\mathbf{U}$ has a 2-norm no greater than 1 and $\|\mathbf{W}_Q^{(i)\top}\mathbf{W}_K^{(i)}\|_2 \leq \sqrt{d_h}$. Hence, every entry of $\text{softmax}(\mathbf{T}^{(i)})$ has a magnitude between $L^{-1}e^{-2}$ and $L^{-1}e^2$. This means

$$\sigma_1\left(\text{softmax}\left(\mathbf{T}^{(i)}\right)\right) = \left\|\text{softmax}\left(\mathbf{T}^{(i)}\right)\right\|_2 \leq \left\|\text{softmax}\left(\mathbf{T}^{(i)}\right)\right\|_F \leq e^2.$$

Moreover, since $\left\|\mathbf{W}_V^{(i)}\right\|_2 \leq 1$, we have that

$$\sigma_j\left(\mathbf{Y}^{(i)}\right) \leq e^2 \sigma_j(\mathbf{U}), \qquad 1 \leq j \leq d_h.$$

Fix some $1 \leq j \leq d$ and let $j_h = \lfloor (j-1)/h \rfloor + 1$. When we concatenate $\mathbf{Y}^{(i)}$ to form $\mathbf{Y}$, using the singular value inequality that $\sigma_{i+j-1}(\mathbf{A}+\mathbf{B}) \leq \sigma_i(\mathbf{A}) + \sigma_j(\mathbf{B})$ and defining $\mathbf{Y}^\mathcal{I}$ to be the concatenation of $\mathbf{Y}^{(i)}$ for all $i \in \mathcal{I}$, we have

$$\sigma_j(\mathbf{Y}) \leq \sigma_{j_h}\left(\mathbf{Y}^{(h)}\right) + \sigma_{j-j_h+1}\left(\mathbf{Y}^{[h-1]}\right) \leq \sigma_{j_h}\left(\mathbf{Y}^{(h)}\right) + \sigma_{j_h}\left(\mathbf{Y}^{(h-1)}\right) + \sigma_{j-2j_h+2}\left(\mathbf{Y}^{[h-2]}\right)$$

$$\leq \cdots \leq \left(\sum_{i=2}^h \sigma_{j_h}\left(\mathbf{Y}^{(i)}\right)\right) + \sigma_{j-(h-1)(j_h-1)}\left(\mathbf{Y}^{(1)}\right) \leq \sum_{i=1}^h \sigma_{j_h}\left(\mathbf{Y}^{(i)}\right) \leq e^2 h\,\sigma_{j_h}(\mathbf{U}).$$

Hence, applying the same inequality again, we have that

$$\sigma_k(\mathbf{Z}) = \sigma_k\left(\mathbf{U} + \frac{1}{\sqrt{N}}\mathbf{Y}\right) \leq \sigma_{k-j+1}(\mathbf{U}) + \sigma_j\left(\frac{1}{\sqrt{N}}\mathbf{Y}\right) \leq \sigma_{k-j+1} + \frac{e^2 h}{\sqrt{N}}\sigma_{\lfloor (j-1)/h \rfloor + 1}. \quad (26)$$

Moreover, from the triangle inequality, we have

$$\sigma_1(\mathbf{Z}) \geq \sigma_1(\mathbf{U}) - \sigma_1\left(\frac{1}{\sqrt{N}}\mathbf{Y}\right) \geq \sigma_1(\mathbf{U}) - \frac{1}{\sqrt{N}}e^2 h\sigma_1(\mathbf{U}) \geq \frac{1}{2}\sigma_1(\mathbf{U}). \quad (27)$$

Combining eq. (26) and (27), we prove the theorem.

**The lower bound.** Let $1 = \sigma_1 \geq \cdots \geq \sigma_d > 0$ be given. Define the input matrix to be

$$\mathbf{U} = \begin{bmatrix} \text{diag}(\sigma_1,\ldots,\sigma_d) & \mathbf{0}_{d\times(L-d)} \end{bmatrix}.$$

For every $1 \leq i \leq h$, let $\mathbf{W}_V^{(i)}$ be

$$\mathbf{W}_V^{(i)} = \begin{bmatrix} \mathbf{I}_{d_h-1} & \mathbf{0}_{(d_h-1)\times(d-d_h+1)} \\ \mathbf{0}_{1\times(d_h-1)} & \mathbf{0}_{1\times(d-d_h+1)} \end{bmatrix}.$$

Set $\varepsilon = \sqrt{N}\,\sigma_d/(5h)$. For every $1 \leq i \leq h$, let $\mathbf{P}^{(i)} \in \mathbb{R}^{L\times L}$ be the matrix so that $\mathbf{P}^{(i)}(1:d_h,((i-1)d_h+1):id_h) = \mathbf{I}_{d_h}$, $\mathbf{P}^{(i)}(d_h,k) = 1$ for every $1 \leq k < (i-1)d_h$ and every $k > id_h$, and is zero elsewhere. For a sufficiently large $\alpha > 0$ determined later on, let

$$\mathbf{W}_Q^{(i)} = \alpha \begin{bmatrix} \text{diag}(\sigma_1^{-1},\ldots,\sigma_{d_h}^{-1}) & \mathbf{0}_{d_h\times(d-d_h)} \end{bmatrix},$$

$$\mathbf{W}_K^{(i)} = \left[ \begin{array}{c|c|c} \begin{matrix} \mathbf{0}_{(d_h-1)\times(i-1)d_h} \\ \begin{bmatrix} \sigma_1^{-1} & \cdots & \sigma_{(i-1)d_h}^{-1} \end{bmatrix} \end{matrix} & \text{diag}(\sigma_{(i-1)d_h+1}^{-1},\ldots,\sigma_{id_h}^{-1}) & \begin{matrix} \mathbf{0}_{(d_h-1)\times(h-i)d_h} \\ \begin{bmatrix} \sigma_{id_h+1}^{-1} & \cdots & \sigma_d^{-1} \end{bmatrix} \end{matrix} \end{array} \right].$$

Then, we have

$$\mathbf{T}^{(i)} := \mathbf{U}^\top \mathbf{W}_Q^{(i)}{}^\top \mathbf{W}_K^{(i)} \mathbf{U}$$

$$= \alpha \begin{bmatrix} \mathbf{I}_{d_h} & \mathbf{0}_{d_h \times (L-d_h)} \end{bmatrix}^\top \begin{bmatrix} \mathbf{0}_{(d_h-1)\times(i-1)d_h} & \mathbf{I}_{d_h} & \mathbf{0}_{(d_h-1)\times(h-i)d_h} & \mathbf{0}_{d_h \times (L-d)} \\ \mathbf{1}_{1\times(i-1)d_h} & & \mathbf{1}_{1\times(h-i)d_h} & \end{bmatrix} = \alpha \mathbf{P}^{(i)}.$$

Therefore, by picking $\alpha$ sufficiently large, we have

$$\|\mathbf{G}^{(i)} - \tilde{\mathbf{G}}^{(i)}\|_2 \leq \epsilon, \quad \mathbf{G}^{(i)} := \mathrm{softmax}(\mathbf{T}^{(i)}), \quad \tilde{\mathbf{G}}^{(i)}(j,k) = \begin{cases} 1, & k = j + (i-1)d_h, \\ 1, & j = d_h \text{ and } k \leq (i-1)d_h, \\ 1, & j = d_h \text{ and } k > id_h, \\ L^{-1}, & k > d, \\ 0, & \text{otherwise.} \end{cases} \tag{28}$$

Note that we have

$$\mathbf{W}_V^{(i)} \mathbf{U} \tilde{\mathbf{G}}^{(i)} = \begin{bmatrix} \mathrm{diag}(\sigma_1, \ldots, \sigma_{h_d-1}) & 0 & \mathbf{0}_{(d_h-1)\times(L-d_h)} \\ \mathbf{0}_{1\times(d_h-1)} & 0 & \mathbf{0}_{1\times(L-d_h)} \end{bmatrix}$$

$$\times \begin{bmatrix} \mathbf{0}_{(d_h-1)\times(i-1)d_h} & \mathbf{I}_{d_h} & \mathbf{0}_{(d_h-1)\times(h-i)d_h} & L^{-1}\mathbf{1}_{d_h\times(L-d)} \\ \mathbf{1}_{1\times(i-1)d_h} & & \mathbf{1}_{1\times(h-i)d_h} & \\ \mathbf{0}_{(L-d_h)\times(i-1)d_h} & \mathbf{0}_{(L-d_h)\times d_h} & \mathbf{0}_{(L-d_h)\times(h-i)d_h} & L^{-1}\mathbf{1}_{(L-d_h)\times(L-d)} \end{bmatrix}$$

$$= \begin{bmatrix} \underbrace{\mathbf{0}_{d_h\times(i-1)d_h} \; \mathrm{diag}(\sigma_1,\ldots,\sigma_{h_d-1},0) \; \mathbf{0}_{d_h\times(h-i)d_h}}_{\mathbf{Y}_{\mathrm{lead}}^{(i)}} & \underbrace{\begin{bmatrix} L^{-1}\mathrm{diag}(\sigma_1,\ldots,\sigma_{h_d-1})\mathbf{1}_{(d_h-1)\times(L-d)} \\ \mathbf{0}_{1\times(L-d)} \end{bmatrix}}_{\mathbf{Y}_{\mathrm{trail}}^{(i)}} \end{bmatrix}.$$

Note that the mulithead attention output is given by

$$\mathbf{Y} = \mathrm{MH\text{-}Attention}(\mathbf{U}; \mathbf{W}_Q, \mathbf{W}_K, \mathbf{W}_V, h) = \begin{bmatrix} \mathbf{Y}_{\mathrm{lead}} & \mathbf{Y}_{\mathrm{trail}} \end{bmatrix} + \mathbf{E},$$

$$\mathbf{Y}_{\mathrm{lead}} = \begin{bmatrix} \mathbf{Y}_{\mathrm{lead}}^{(1)}{}^\top & \cdots & \mathbf{Y}_{\mathrm{lead}}^{(h)}{}^\top \end{bmatrix}^\top, \qquad \mathbf{Y}_{\mathrm{trail}} = \begin{bmatrix} \mathbf{Y}_{\mathrm{trail}}^{(1)}{}^\top & \cdots & \mathbf{Y}_{\mathrm{trail}}^{(h)}{}^\top \end{bmatrix}^\top,$$

where

$$\|\mathbf{E}\|_2 \leq \sum_{i=1}^h \|\mathbf{W}_V^{(i)} \mathbf{U} \tilde{\mathbf{G}}^{(i)} - \mathbf{W}_V^{(i)} \mathbf{U} \tilde{\mathbf{G}}^{(i)}\|_2 \leq \sum_{i=1}^h \|\mathbf{W}_V^{(i)} \mathbf{U}\|_2 \|\mathbf{G}^{(i)} - \tilde{\mathbf{G}}^{(i)}\|_2 \leq h\varepsilon,$$

and the layer output is given by

$$\mathbf{Z} = \mathbf{U} + \frac{1}{\sqrt{N}} \mathbf{Y} = \begin{bmatrix} \mathbf{Z}_{\mathrm{lead}} & \mathbf{Z}_{\mathrm{trail}} \end{bmatrix} + \frac{1}{\sqrt{N}} \mathbf{E},$$

$$\mathbf{Z}_{\mathrm{lead}} = \mathrm{diag}(\sigma_1, \ldots, \sigma_d) + \frac{1}{\sqrt{N}} \mathbf{Y}_{\mathrm{lead}}, \qquad \mathbf{Z}_{\mathrm{trail}} = \frac{1}{\sqrt{N}} \mathbf{Y}_{\mathrm{trail}}.$$

Since $\mathbf{Z}_{\mathrm{lead}}$ is a diagonal matrix with nonnegative entries, its singular values equal its diagonal entries, which are

$$\begin{array}{ccccc} \sigma_1 + \frac{1}{\sqrt{N}}\sigma_1 & \sigma_2 + \frac{1}{\sqrt{N}}\sigma_2 & \cdots & \sigma_{d_h-1} + \frac{1}{\sqrt{N}}\sigma_{d_h-1} & \sigma_{d_h} \\ \sigma_{d_h+1} + \frac{1}{\sqrt{N}}\sigma_1 & \sigma_{d_h+2} + \frac{1}{\sqrt{N}}\sigma_2 & \cdots & \sigma_{d_h+(d_h-1)} + \frac{1}{\sqrt{N}}\sigma_{d_h-1} & \sigma_{d_h+d_h} \\ \vdots & \vdots & \ddots & \vdots & \vdots \\ \sigma_{(h-1)d_h+1} + \frac{1}{\sqrt{N}}\sigma_1 & \sigma_{(h-1)d_h+2} + \frac{1}{\sqrt{N}}\sigma_2 & \cdots & \sigma_{(h-1)d_h+(d_h-1)} + \frac{1}{\sqrt{N}}\sigma_{d_h-1} & \sigma_{(h-1)d_h+d_h} \end{array}. \tag{29}$$

Moreover, since $\mathbf{Z}_{\mathrm{lead}}$ is a submatrix of $\mathbf{Z}$, the singular values of $\mathbf{Z}$ are no smaller than the singular values of $\mathbf{Z}_{\mathrm{lead}}$. Given $1 \leq i \leq h$ and $1 \leq j \leq d_h$, let $\xi_{i,j} = \sigma_{(i-1)d_h+j} + \mathbb{1}_{\{j\neq d_h\}}\sigma_j/\sqrt{N}$ be the element in row $i$ and column $j$ in the array in eq. (29). Then, we have that

$$\xi_{i,j} \leq \xi_{i',j'}, \qquad \text{for any } i \geq i' \text{ and } j \geq j'.$$

Hence, $\xi_{i,j}$ is no greater than the $(i \times j)$th singular value of $\mathbf{Z}_{\text{lead}}$. That means for every $1 \leq i \leq h$, the $k$th singular value of $\mathbf{Z}_{\text{lead}}$ satisfies

$$\sigma_k(\mathbf{Z}_{\text{lead}}) \geq \xi_{i,\lceil k/i \rceil} = \sigma_{(i-1)d_h + \lceil k/i \rceil} + \frac{\mathbb{1}_{\lceil k/i \rceil \neq d_h}}{\sqrt{N}} \sigma_{\lceil k/i \rceil}.$$

Using Weyl's inequality, we have

$$\sigma_k(\mathbf{Z}) \geq \sigma_k(\mathbf{Z}_{\text{lead}}) - \frac{1}{\sqrt{N}}\sigma_1(\mathbf{E}) \geq \sigma_k(\mathbf{Z}_{\text{lead}}) - \frac{h\varepsilon}{\sqrt{N}}$$

$$\geq \left(1 - \frac{1}{5}\right)\left(\sigma_{(i-1)d_h + \lceil k/i \rceil} + \frac{\mathbb{1}_{\lceil k/i \rceil \neq d_h}}{\sqrt{N}}\sigma_{\lceil k/i \rceil}\right).$$

Moreover, we have

$$\sigma_1(\mathbf{Z}) \leq \sigma_1(\mathbf{Z}_{\text{lead}}) + \sigma_1(\mathbf{Z}_{\text{trail}}) + \frac{1}{\sqrt{N}}\sigma_1(\mathbf{E}) \leq \left(1 + \frac{1}{\sqrt{N}}\right) + \frac{1}{\sqrt{N}}\|\mathbf{Y}_{\text{trail}}\|_F + \frac{1}{\sqrt{N}}h\varepsilon$$

$$\leq 2 + \frac{1}{\sqrt{N}}\sqrt{L \times d \times L^{-2}} + \frac{1}{5} \leq \frac{16}{5}.$$

Combining the two inequalities above, we obtain the theorem. $\qquad\square$

The proof of Corollary 1 is a straightforward manipulation of the ceiling and floor operators.

*Proof of Corollary 1.* Set $j = \lfloor (hk+1)/(h+1) \rfloor$, we have

$$\sigma_{k-j+1} = \sigma_{k-\lfloor (hk+1)/(h+1) \rfloor + 1} \leq \sigma_{\lceil k-(hk+1)/(h+1)+1 \rceil} = \sigma_{\lceil (k+h)/(h+1) \rceil},$$

and

$$\sigma_{\lfloor (j-1)/h \rfloor + 1} = \sigma_{\lfloor (\lfloor (hk+1)/(h+1) \rfloor - 1)/h \rfloor + 1} \leq \sigma_{\lfloor ((hk+1)/(h+1)-1)/h \rfloor + 1} = \sigma_{\lfloor (k+1)/(h+1) \rfloor + 1}.$$

The corollary follows from Theorem 4. $\qquad\square$

# G    WATCHING THE WEIGHTS OF LARGE MODELS

The central claims made in this paper are heavily based on the low-rank structures of weight matrices in large-scale time series foundation models. In this section, we provide more empirical analysis of the singular values of these matrices.

## G.1    COMPARING THE WEIGHTS OF TSFMS AND LLMS

For the first set of comparisons, we consider three models: T5, Chronos, and Chronos-Bolt. While T5 is an LLM, Chronos and Chronos-Bolt are TSFMs. The three models we compare have the same base size, and each model contains 12 encoder layers and 12 decoder layers. From each layer, we take out a matrix $\mathbf{W}$, which is either the query projection matrix $\mathbf{W}_Q \in \mathbb{R}^{768 \times 768}$ or the first matrix of the MLP layer $\mathbf{W}_{\text{MLP}} \in \mathbb{R}^{3072 \times 768}$. We apply an SVD to the weight matrix $\mathbf{W} = \mathbf{U}\mathbf{\Sigma}\mathbf{V}^\top$ and construct a histogram out of all relative singular values in $\text{diag}(\mathbf{\Sigma})/\mathbf{\Sigma}_{1,1}$.

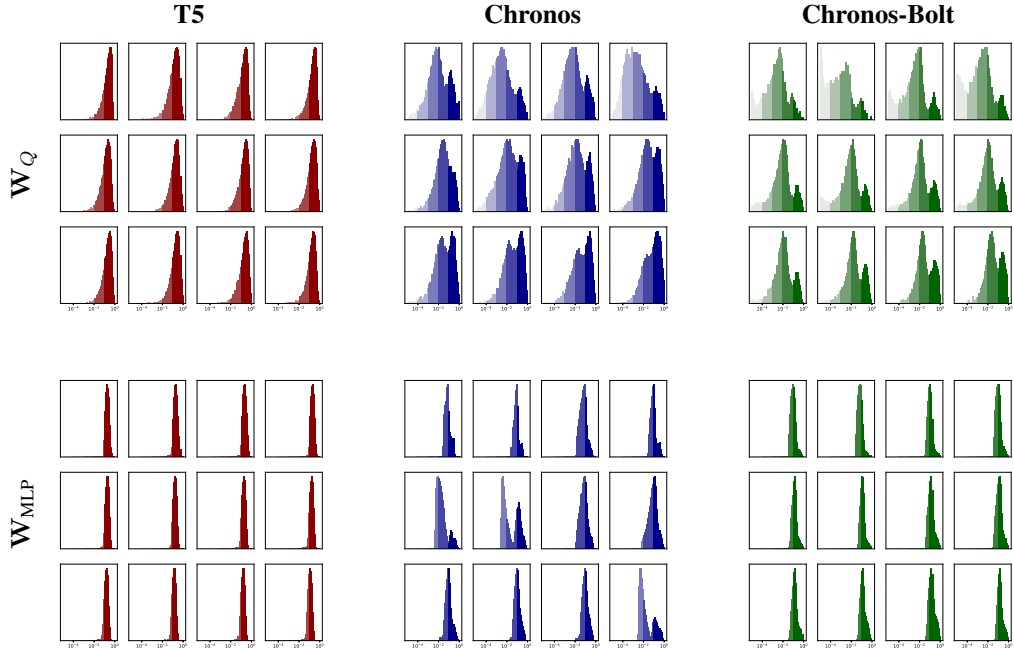

**Figure 10:** The distribution of all relative singular values in a weight matrix $\mathbf{W}$ in a T5, Chronos, or Chronos-Bolt model. We use five progressively fainter face colors to indicate the range where the relative singular values are in the (1e-5, 1e-4], (1e-4, 1e-3], (1e-3, 1e-2], (1e-2, 1e-1], and (1e-1, 1e0] ranges, respectively. Each model contains 12 encoder layers, and we arrange the corresponding 12 weight matrices in row-major order. Note that all histograms are made on a semilog-x scale.

From Figure 10, we make some observations that align with the major claims made in the main manuscript:
1. The relative singular values decay faster for Chronos and Chronos-Bolt and they decay much slower for T5. The reason is that TSFMs leverage low-rank embeddings and do not require high-rank attention matrices, which is not the case for T5, which inevitably needs a high-rank embedding.
2. In Chronos and Chronos-Bolt, when the layers get deeper, we generally have a higher-rank structure. This aligns with the flow-of-ranks idea: as an input is pushed through more nonlinear attention and MLP layers, its rank gets higher, and we need higher-rank attention matrices.
3. The attention matrices are generally more compressible than MLP matrices. This is not surprising, because the attention matrices in these models are square matrices, making low-rank structures much easier to emerge than in a rectangular MLP matrix.

## G.2 WATCHING THE WEIGHTS OF TSFMS DURING TRAINING

Another mysterious finding that we used in the main manuscript is that the attention matrices in a pretrained Chronos or Chronos-Bolt model all demonstrate low-rank structure; however, while Theorem 3 only suggests that the attention matrices *can* be written in a low-rank form, it does not preclude these matrices from having a high-rank form. To understand why the attention matrices happen to exhibit a low-rank structure, we watch the training dynamics of these attention matrices, where we pretrain a small Chronos model and record the six weight matrices in its six encoder layers at a few different training steps. Unlike Figure 10, in Figure 11, we do not show the relative singular values $\sigma_j/\sigma_1$. The reason will be clear later.

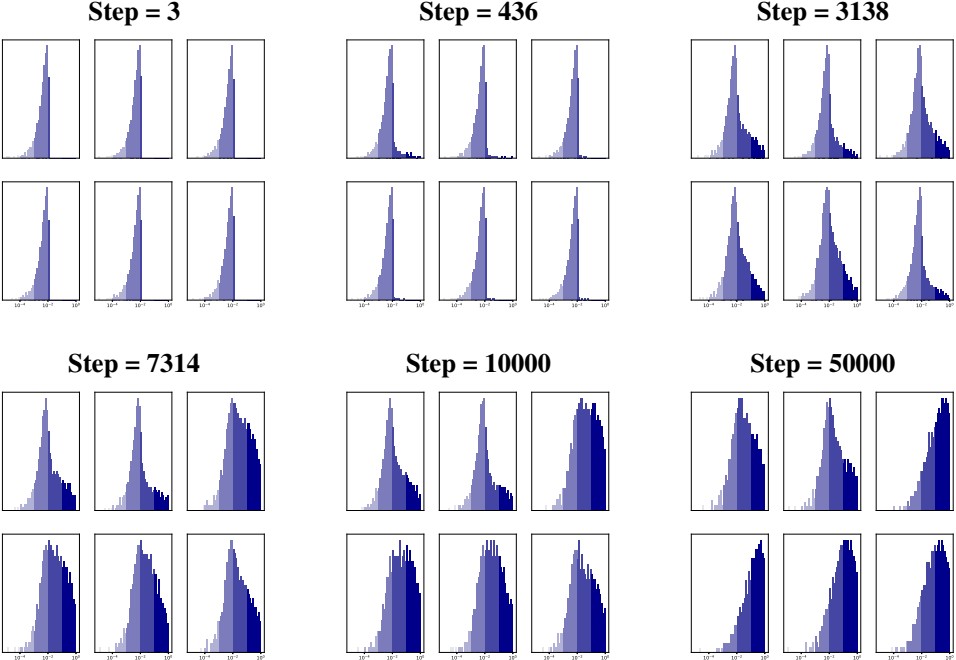

**Figure 11:** The distribution of all absolute singular values in a weight matrix $\mathbf{W}_Q$ in a Chronos model over training. The model contains 6 encoder layers, and we arrange the corresponding 6 weight matrices in row-major order. Note that all histograms are made on a semilog-x scale.

We note that the initialization of Chronos typically relies on a very small scaling factor. That is, the weights are all initialized to be small. From Figure 11, we see that the weight matrices are eventually learned to be larger: that is, by looking at the absolute singular values instead of the relative ones, we see that the leading singular value, i.e., the norm of the weight matrix $\mathbf{W}_Q$, eventually gets larger. When they get larger, we note that the low-rank structure evolves. This "learning the leading singular direction" interpretation is more plausible than if the weight matrix $\mathbf{W}_Q$ is initialized large, because it has no incentive to "forget the residual ranks" when it does not need to.

## G.3 WATCHING THE WEIGHTS IN A MULTIHEAD ATTENTION

In Figure 9, we showed the "correlation matrix" of a three-head attention matrix $\mathbf{W}_Q$. In Figure 12, we show more of these matrices when a different number of heads $h$. For each number of heads $h$, we can see clearly $h$ low-rank blocks on the diagonal, corresponding to the in-head low-rank attention.

One interesting observation we can make is that the off-diagonal parts of the heatmaps, while much more yellowish than the diagonal parts, also have block structures. This is expected: if the rows in each head are very colinear, then when you compare rows between two different heads, they should share a similar angle. That is, we have a duality that not only holds for attention matrices but also

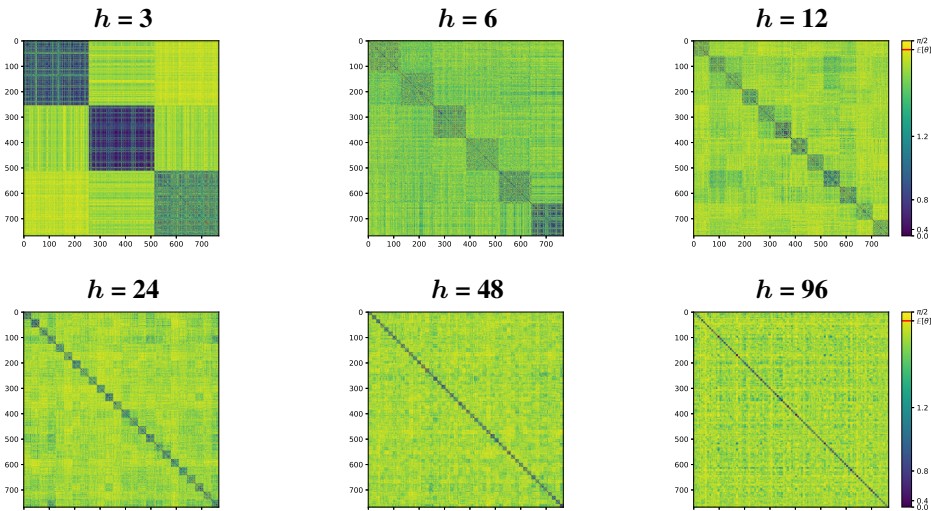

**Figure 12:** We pretrain a Chronos model with a different number of heads $h$. As $h$ changes, we show the angle between every pair of rows of $\mathbf{W}_Q$ in the first encoder layer.

holds in general: the darker a diagonal block is, the clearer the edges of off-diagonal blocks in the same row of blocks will be.

The off-diagonal parts of attention matrices, while much more orthogonal than the rows within each head, are still far from random. This means the heads themselves are also somewhat correlated, which is not surprising given that the row spaces of query, key, and value matrices are subspaces of the potentially low-dimensional row space of the input and that the models are trained with a single objective. There are two forces pulling against each other: a head-dependent random initialization, which "orthogonalizes" different heads, and a head-independent training objective that tries to align these heads.

There is a third trend (and the second duality) that is also very interesting: darker diagonal blocks seem to correspond to lighter off-diagonal blocks. There is one potential explanation for that: if the diagonal blocks are very dark, that means each head only attends to a tiny bit of information of the input. In order to obtain good results, other heads must incorporate the remaining bits of the input, resulting in very different sketchings and larger angles. On the other hand, if the diagonal blocks are brighter, that means it already contains a fair amount of information about the input, and lots of the input information is shared across heads. Hence, they should be much more correlated during training.

# H    VISUALIZATION OF TOKENIZERS

In section 2, we consider a large corpus of inputs. Here, we perform two case studies to look into how each of the embeddings works. In particular, we select two time-series input data:

- A sinusoidal wave $\sin(t)$.

- A random Gaussian input, where each entry is i.i.d. $\mathcal{N}(0,1)$.

Let $\mathbf{X} \in \mathbb{R}^{d \times L}$ be the embedded input. For each pair of vectors $\mathbf{x}_i$ and $\mathbf{x}_j$ of $\mathbf{X}$, we compute the their correlation:

$$\theta_{i,j} = \arccos\left(\frac{|\mathbf{x}_i^\top \mathbf{x}_j|}{\|\mathbf{x}_i\|_2 \|\mathbf{x}_j\|_2}\right).$$

These correlation matrices are shown in Figure 13. There are two interesting observations to make: first, for WaveToken, the correlation matrix given a sinusoidal input has clearly four blocks — corresponding to the low-frequency wavelets and high-frequency ones. For Time MOE, the angles are so much darker because we apply a continuous embedding on a one-dimensional input space, leading to an ultimately low-rank structure.

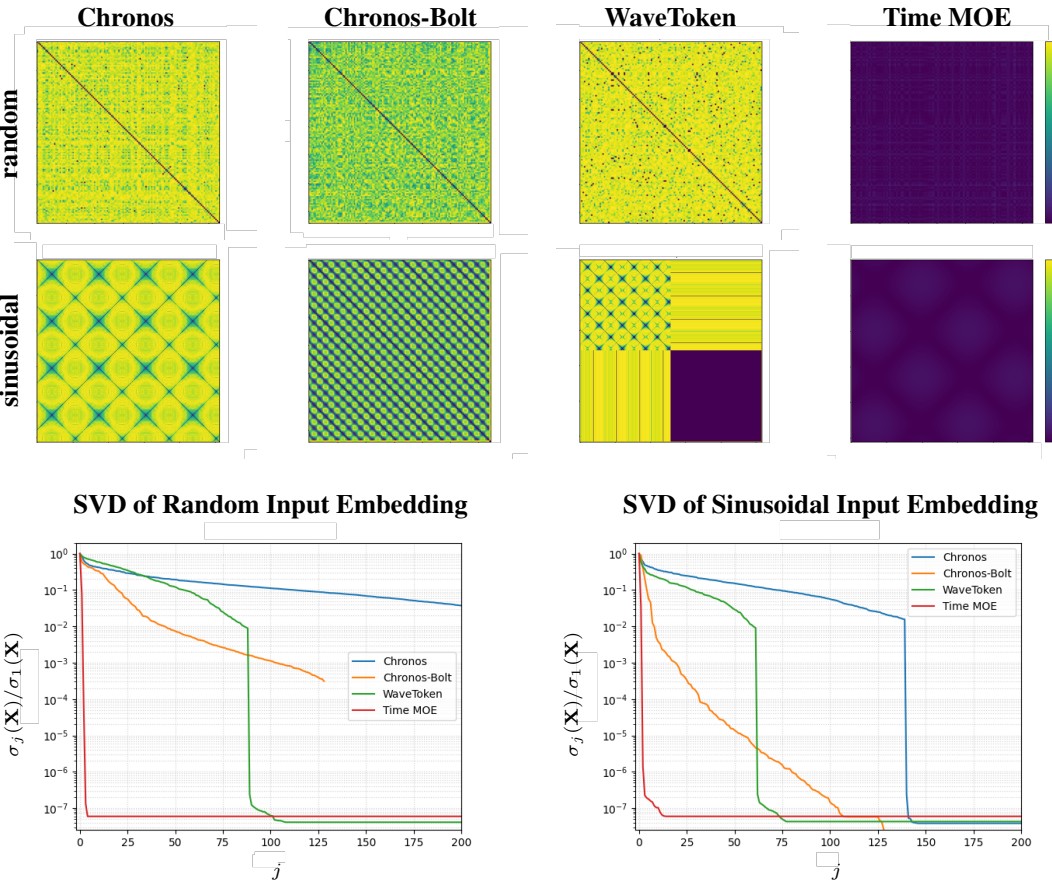

**Figure 13:** The heat maps show the correlation matrix of the embedded input with a random context or a sinusoidal one, using four different embedding strategies. The two line plots show the relative singular values of the embedded matrix $\mathbf{X}$.

We also observe that the numerical rank of the embedded input $\mathbf{X}$ is generally higher when the context is random than sinusoidal. This is not surprising either, because the temporal relationship within a random context is much more complex than that in a sinusoidal one.

## I   MORE ON THE CHEBYSHEV EMBEDDING

In Figure 4, we show an experiment where we increase the rank of a fixed input embedding and watch the numerical ranks of pretrained Chronos models with that embedding. Here, we further explain how we can control the rank of this fixed input embedding function using Chebyshev polynomials. To motivate our design, we first consider how we can compute a rank-1 embedding function $\mathbf{\Phi}_1 : \mathbb{R} \to \mathbb{R}^d$. In this case, what $\mathbf{\Phi}_1$ needs to do is to map the real line linearly onto a one-dimensional subspace of $\mathbb{R}^d$. That is, we can set $\mathbf{\Phi}_1(x) = x\mathbf{u}$ for some fixed unit vector $\mathbf{u} \in \mathbb{R}^d$.

Now, how can we use this idea to design a rank-$k$ embedding $\mathbf{\Phi}_k(x)$? One way to do that is by considering the following extension of $\tilde{\mathbf{\Phi}}_k$:

$$\mathbf{\Phi}_k : \mathbb{R} \to \mathbb{R}^d, \qquad x \mapsto f_1(x)\mathbf{u}_1 + \cdots + f_k(x)\mathbf{u}_k.$$

As long as the functions $f_1, \ldots, f_k$ are linearly independent, the image of the real line, $\mathbf{\Phi}_k(\mathbb{R})$, is a subset of a $k$-dimensional subspace of $\mathbb{R}^d$. The only question that remains is: how to choose $f_1(x), \ldots, f_k(x)$. Perhaps the easiest way to choose such a basis is by making them monomials: $f_j(x) = x^j$. However, the monomial basis often suffers from many numerical stability issues and is not ideal for the embedding, e.g., for a large $j$, the set $\{x, \ldots, x^j\}$ is very ill-conditioned. To choose a well-conditioned basis, we use Chebyshev polynomials, which are orthogonal on $[-1, 1]$. That is, we set $f_j(x) = T_j(x/x_{\max})$, where $x_{\max}$ is the maximum number considered in quantization, which equals 15 in the case of Chronos. Given a set of points $x_1, \ldots, x_L$ to embed, we assemble the embedded matrix $\mathbf{X} \in \mathbb{R}^{d \times L}$ as follows:

1. Sample orthonormal random column vectors $\mathbf{u}_1, \ldots, \mathbf{u}_k \in \mathbb{R}^{d \times 1}$.

2. Compute row vectors $\mathbf{f}_1, \ldots, \mathbf{f}_k \in \mathbb{R}^{1 \times L}$, where the $i$th entry of $\mathbf{f}_j$ is $T_j(x_i/x_{\max})$.

3. Compute the outer product $\mathbf{X} = \sum_{j=1}^k \mathbf{u}_j \mathbf{f}_j$.

We show the visualization of our Chebyshev embeddings in Figure 14, where as see that as we increase the rank $k$, the embedding becomes more "complicated."

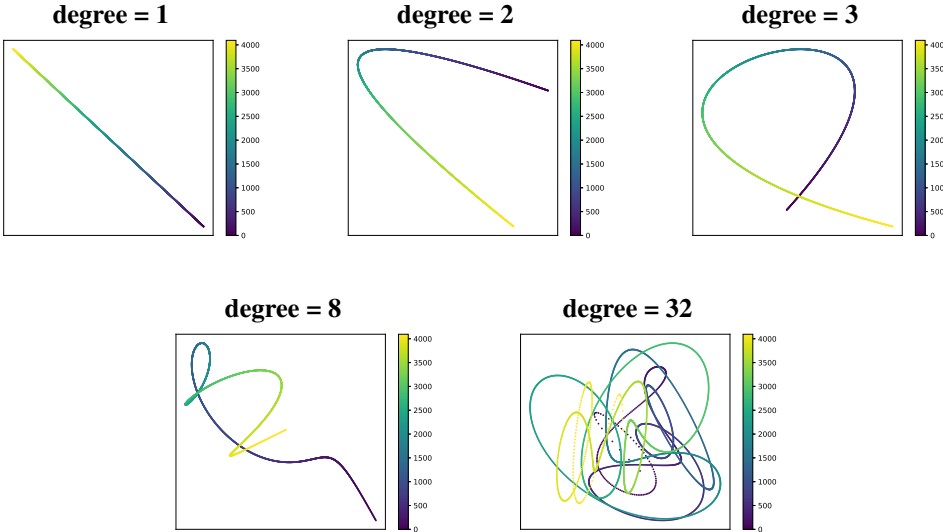

**Figure 14:** Visualization of the Chebyshev embeddings. For each embedding, we visualize the first two dimensions out of the 768 in the hidden space. There are 4096 points embedded in the hidden space, whose corresponding values in the input domain range from $-15$ to $15$.

## J  NUMERICAL EXPERIMENTS TO CORROBORATE OUR THEOREM

In this appendix, we provide numerical experiments, simulated in MATLAB R2024b, that verify the theoretical statements we made in the main manuscript.

### J.1  TWO NUMERICAL EXPERIMENTS ON THEOREM 3

To verify Theorem 3, we fix the hidden dimension to be $d = 512$ and the vocabulary size $N = 4096$. We create a random embedded matrix $\mathbf{X} \in \mathbb{R}^{d \times N}$ with singular values $\mathbf{s} \in \mathbb{R}^d$ ranging from $e^{-5}$ to $e^0$ and uniformly distributed on a logarithmic scale. This matrix is computed by randomly sampling an orthogonal matrix $\mathbf{U} \in \mathbb{R}^{d \times d}$ and a matrix $\mathbf{V} \in \mathbb{R}^{N \times d}$ with orthonormal columns, via QR-decomposing random matrices, and setting $\mathbf{X} = \mathbf{U} \operatorname{diag}(\mathbf{s}) \mathbf{V}^\top$.

Next, we randomly sample three attention matrices $\mathbf{W}_Q, \mathbf{W}_K, \mathbf{W}_V$, and for each reduced-order $\tilde{d} = 1, \ldots, d - 1$, we use the constructive formulas given in the proof of Theorem 3 to compute the reduced matrices $\tilde{\mathbf{W}}_Q, \tilde{\mathbf{W}}_K, \tilde{\mathbf{W}}_V$. We set the input to be the entire vocabulary matrix $\mathbf{X}$ and compute the output of the original attention layer, defined by $\mathbf{W}_Q, \mathbf{W}_K, \mathbf{W}_V$, as well as that of the reduced one, defined by $\tilde{\mathbf{W}}_Q, \tilde{\mathbf{W}}_K, \tilde{\mathbf{W}}_V$. We evaluate the Frobenius norm of the difference between the two outputs.

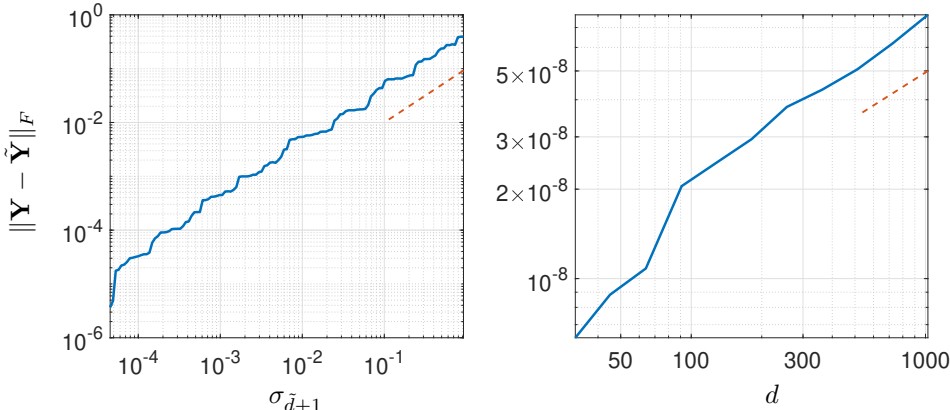

**Figure 15:** The left panel shows the relationship between the reduced-order $\tilde{d}$ and the approximation error of a randomly sampled attention layer applied to a randomly sampled input matrix $\mathbf{X}$ with controlled singular values $\sigma_1, \ldots, \sigma_d$. The reference line has a slope of 1 in the log-log plot. The right panel shows the relationship between the dimension of the hidden space $d$ and the approximation error of a fixed-degree reduced-order model. For each hidden space $d$, we randomly resample the attention matrices and embedded matrix $\mathbf{X}$, holding its leading singular values unchanged. The reference line has a slope of $1/2$ in the log-log plot.

Figure 15 shows two controlled experiments by changing a different variable. On the left, we change the truncation degree $\tilde{d}$, which in turn controls the singular value $\sigma_{\tilde{d}+1}$. The reference line has a slope of 1, revealing a linear relationship between $\|\mathbf{Y} - \tilde{\mathbf{Y}}\|_F$ and $\sigma_{\tilde{d}+1}$, as indicated in Theorem 3. On the right, we change the size of the hidden space $d$, and the reference line has a slope of $1/2$, which verifies the $\sqrt{d}$ factor in the statement of Theorem 3.

### J.2  A NUMERICAL EXPERIMENT ON THEOREM 6

The essence of Theorem 6 is that a sparse multi-head sketching is more effective than a dense single-head sketching. We use an experiment to verify that. In our setting, we set $d = 2048$ and $L = 4096$, and we randomly sample an input matrix $\mathbf{X} \in \mathbb{R}^{d \times L}$ with exponentially decaying singular values, as shown in the left panel of Figure 16. Our sampling method is the same as the one outlined in the previous experiment.

We set our reduced rank to $\tilde{d} = 4$. If we just use a single head to sketch a rank-$\tilde{d}$ space from the row space of $\mathbf{X}$, then its accuracy is lower-bounded by $\sigma_{\tilde{d}+1}$, which is still a large number. To explore the potential of sparse head-dependent sketching, we increase the number of heads from $h = 1$ to $128$ and keep it an integral divisor of $d$. For each $h$, we use the sparse sketching; that is, we assemble a random matrix

$$\mathbf{W}_{2,h} = \begin{bmatrix} \mathbf{W}_{2,h}^{(1),\top} & \cdots & \mathbf{W}_{2,h}^{(1),\top} \end{bmatrix}^{\top}, \qquad \mathbf{W}_{2,h}^{(i)} \in \mathbb{R}^{\tilde{d} \times d}.$$

The matrix $\mathbf{W}_{2,h}$ is sparse in the sense that it has exactly $d_h = d/h$ non-zero entries, whose positions are randomly chosen, and for each non-zero position, we sample its value i.i.d. from $\mathcal{N}(0, 1)$. Note that given this sparse design, $\mathbf{W}_{2,h}$ has the same number of non-zero entries for any $h$.

Now, our question is: what is the different between the row space of $\mathbf{W}_{2,h}\mathbf{X}$ and that of $\mathbf{X}$? In other words, how good is the sketching using $\mathbf{W}_{2,h}$? To this end, we clearly have that $\mathcal{R}(\mathbf{W}_{2,h}\mathbf{X}) \subset \mathcal{R}(\mathbf{X})$, where we use the notation $\mathcal{R}$ for the row space of a matrix. Hence, the remaining question is how much in $\mathcal{R}(\mathbf{X})$ is not filled by $\mathcal{R}(\mathbf{W}_{2,h}\mathbf{X})$. This can be measured by projecting $\mathcal{R}(\mathbf{X})$ onto $\mathcal{R}(\mathbf{W}_{2,h}\mathbf{X})$ and measure the loss by the projection:

$$d(\mathcal{R}(\mathbf{X}), \mathcal{R}(\mathbf{W}_{2,h}\mathbf{X})) = \left\| \mathrm{orth}(\mathbf{X}^{\top}\mathbf{W}_{R,h}^{\top})\,\mathrm{orth}(\mathbf{X}^{\top}\mathbf{W}_{R,h}^{\top})^{\top}\mathbf{X}^{\top} - \mathbf{X}^{\top} \right\|_2. \tag{30}$$

We show this distance in the right panel of Figure 16, and we relate this to the singular values of $\mathbf{X}$. In that sense, since the vertical rules in the left panel are almost evenly spaced, it shows that the "effective rank" of $\mathcal{R}(\mathbf{W}_{2,h}\mathbf{X})$ grows proportionally with respect to $h$, indicating that the quality of a sparse multi-head sketching is comparable to the quality of a dense multi-head sketching.

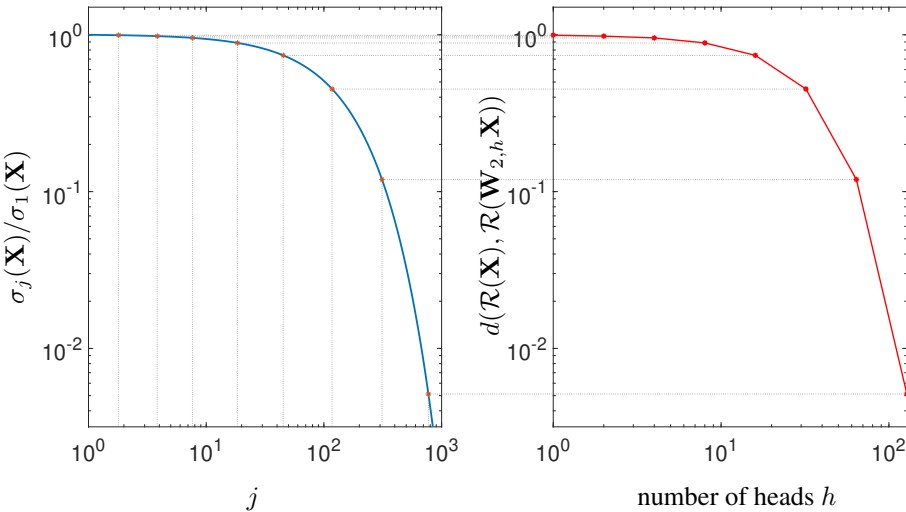

**Figure 16:** On the left, we show the singular values of the input matrices. On the right, we compute the "sketching error" defined in eq. (30), as we change the number of heads in a sparse sketching. We map the sketching errors back to the singular value plot to indicate the index $j$ such that our sketching error achieves an error below $\sigma_{j+1}$.

### J.3    A NUMERICAL EXPERIMENT ON THEOREM 4

Our final numerical experiment considers the flow-of-ranks. Theorem 4 suggests that a larger number of heads also facilitates the flow-of-rank, and this is hard to validate empirically with pretrained Chronos models. In our targeted numerical experiment, we fix an input matrix $\mathbf{X}$ with predefined exponentially decaying singular values, which is, again, randomly sampled from the product of a random orthogonal matrix, a predefined diagonal matrix, and the transpose of a random matrix with orthonormal columns (see the previous two subsections). Then, we randomly select $\mathbf{W}_Q, \mathbf{W}_K, \mathbf{W}_V$ and compute the output of the attention mechanism defined by these three matrices together with a number of heads parameter $h$. Figure 17 gives us that the output matrix

$$\mathbf{Y} = \mathrm{MH\text{-}Attention}(\mathbf{X}; \mathbf{W}_Q, \mathbf{W}_K, \mathbf{W}_V, h) + \mathbf{X}$$

is higher-rank as $h$ increases.

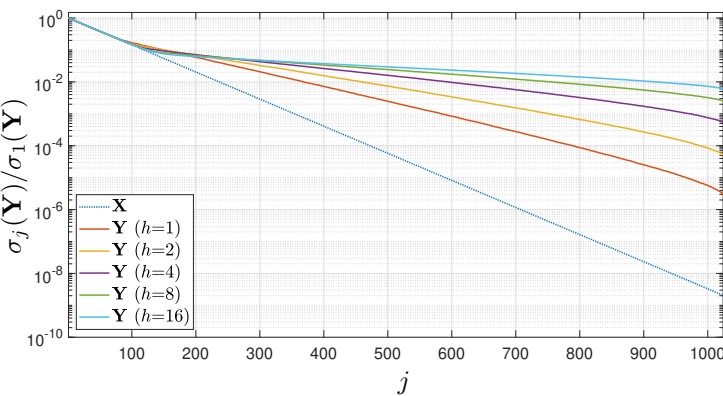

**Figure 17:** Relative singular values of the input matrix $\mathbf{X}$ and output matrices $\mathbf{Y} = $ MH-Attention$(\mathbf{X}; \mathbf{W}_Q, \mathbf{W}_K, \mathbf{W}_V, h) + \mathbf{X}$ as we change the number of heads $h$. We see that the numerical rank of $\mathbf{Y}$ increases with $h$.

# K    ADDITIONAL EXPERIMENTS AND DISCUSSIONS

**Pretraining a Compressed Chronos-Bolt and a Compressed Moirai.** In addition to Chronos, we also pretrain a compressed Chronos-Bolt model. To show the promise of compression, we start with an already-small Chronos-Bolt (small) model, based on the T5 (small) architecture. The table and figures in Table 3 can be read in the same way as those in Figure 7. In particular, we see that for both MASE and WQL, the compressed Chronos-Bolt models completely form the Pareto frontier on our evaluation benchmark. That is, given any local method or pretrained foundation model, there exists a compressed Chronos-Bolt model that is simultaneously faster and more accurate.

**Table 3:** Results of pretraining a compressed Chronos-Bolt (small) model. We compare the performance scores *relative to the original pretrained model*. The last row is the baseline.

| $\tilde{d}_0$ | $\alpha$ | Inference | | In-Domain | | Zero-Shot | |
|---|---|---|---|---|---|---|---|
| | | Time | Space | WQL $\downarrow$ | MASE $\downarrow$ | WQL $\downarrow$ | MASE $\downarrow$ |
| 2 | 0.25 | 0.517 | 0.802 | 1.005 | 1.004 | 1.013 | 0.999 |
| 3 | 0.50 | 0.601 | 0.821 | 1.005 | 0.996 | 1.009 | 1.001 |
| 5 | 0.70 | 0.740 | 0.840 | 0.980 | 1.003 | 0.991 | 1.010 |
| 64 | 0.00 | 1.000 | 1.000 | 1.000 | 1.000 | 1.000 | 1.000 |

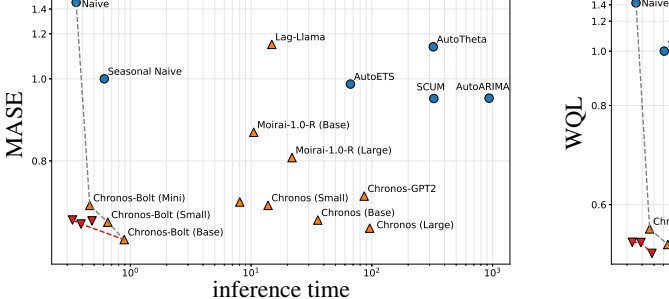 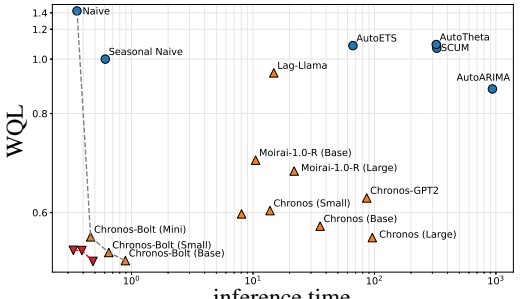

While it is hard to find a rule of thumb that works for any task, we propose a guideline that should work fine in most cases. Let $D$ be the number of layers and $d$ the hidden size. Set the per-layer target rank

$$r_\ell = r_1 + (r_D - r_1)\left(\tfrac{\ell-1}{D-1}\right)^\alpha, \quad \ell = 1, \dots, D,$$

where we can set $r_{\min}$ = the median numerical rank of a small sample of input embeddings (or 16 if unknown), $r_{\max} = d/2$, and $\gamma = 0.5$. This design guarantees that $r_\ell$ grows smoothly and monotonically from $r_{\min}$ to $r_{\max}$ (concave in depth), aligning capacity with the observed flow-of-ranks while keeping early layers compact.

**Table 4:** Results of pretraining a compressed Moirai-1.0-R-base model. We compare the performance scores *relative to the original pretrained model*. We show prediction losses when the flow-of-ranks is or is not used. The last row is the baseline.

| Size Ratio | With Flow-of-ranks | | | | Without Flow-of-ranks | | | |
|---|---|---|---|---|---|---|---|---|
| | $\tilde{d}_0$ | $\alpha$ | WQL $\downarrow$ | MASE $\downarrow$ | $\tilde{d}_0$ | $\alpha$ | WQL $\downarrow$ | MASE $\downarrow$ |
| 0.250 | 8 | 0.34 | 1.001 | 1.014 | 16 | 0.00 | 1.069 | 1.050 |
| 0.500 | 10 | 0.58 | 0.996 | 1.007 | 32 | 0.00 | 1.038 | 1.036 |
| 1.000 | - | - | - | - | 64 | 0.00 | 1.000 | 1.000 |

## L    THE USE OF LARGE LANGUAGE MODELS (LLMS)

LLMs are used for polishing the writing and word choices of a few sections in the main text. They are not used in the conceptualization and implementation of research.

