# OpenReview forum: "Understanding Transformers for Time Series: Rank Structure, Flow-of-ranks, and Compressibility"
_ICLR.cc/2026/Conference — ICLR 2026 Poster_

### Official Review · Reviewer_u1Ge · 2025-10-24

**Soundness:** 4
**Presentation:** 3
**Contribution:** 3
**Rating:** 8
**Confidence:** 3

**Summary:**

This paper analyzes Transformers for time-series data through the lens of rank structure. The authors show that time-series embeddings are inherently low-rank, unlike those in text or vision. They prove that this structure induces low-rank attention matrices, introducing the concept of flow-of-ranks to describe how rank gradually increases with layer depth due to nonlinear mixing. They demonstrate that time-series foundation models, such as Chronos, can be compressed by up to 65% in inference time and 81% in memory without accuracy loss.

**Strengths:**

1. The paper combines linear-algebraic theory with well-designed experiments confirming the predicted low-rank behavior in TSFMs.
2. The authors introduce a new analytical lens (flow-of-ranks) that connects data modality to model design. The findings have real design implications for TSFMs.

**Weaknesses:**

1. The main validation focuses on Chronos. Testing on other TSFMs (like TimesFM and Time-MoE) would strengthen the generality claim.
2. The comparison to prior compression methods (LoRA) is missing, which makes it unclear how much gain stems from modality vs. technique.

**Questions:**

1. How does the proposed compression compare quantitatively with existing low-rank or sparse-attention baselines (e.g., LoRA, Linformer)?
2. Does the low-rank structure persist after fine-tuning a compressed model on downstream tasks?
3. Can the flow-of-ranks pattern be empirically confirmed on other TSFMs beyond Chronos?
4. How would the theory extend to multivariate or irregularly sampled time series?
5. Could you provide a simple practical guideline (e.g., rank schedule formula) for designing TSFMs from scratch?

---

> ### Author Response · Authors · 2025-11-15
> **Official Comment by Authors (1/2)**
>
> We thank the reviewer for the careful review and insightful comments.
>
>
> * **W1. Compression of Models Beyond Chronos**
>
>
>   We thank the reviewer for pointing out this limitation. We agree that the empirical evaluation can be strengthened by compressing a model from outside the Chronos family. Please see the results in Table 3 of the revised manuscript or in our general response. Our results show that Moirai is also very compressible, supporting our claims on the compressibility of TSFMs with more generality. In addition, it provides an ablation of flow-of-ranks, which shows that a model compressed in a layer-dependent way performs significantly better than one whose layers are compressed in the same way.
>
>
> * **W2. Comparison to LoRA**
>
>
>    We highlight the differences between our approach and LoRA. Our method targets the intrinsic compressibility of the attention operators; that is, we *replace* attention matrices with low-rank factors justified by the low-dimensional structure of TSFM inputs, which yields compute-efficient inference with theoretical guarantees. In contrast, LoRA is a parameter-efficient fine-tuning method that keeps the backbone dense and learns a low-rank *update*. It typically does not reduce inference cost even though it reduces trainable parameters and storage. Thus, the two approaches address different goals and are complementary: a compressed backbone can still be adapted with LoRA. We have clarified this distinction in the revised paper.
>
>
> * **Q1. Comparison to LoRA and Sparse Attention**
>
>
>    We have clarified the comparison to LoRA above. We appreciate the suggestion to compare with sparse/linear attention. Our approach is orthogonal. We compress the operator side by factorizing the attention *projection matrices* based on provable low-rank structure of TSFM embeddings. This leaves the softmax attention rule unchanged, and reduces the $\mathcal{O}(Ld^2)$ projection cost. In contrast, sparse/linear attention methods approximate the sequence-side interaction matrix to reduce the $\mathcal{O}(L^2)$ cost. Consequently, the two families target different bottlenecks and are complementary: in principle, a compressed backbone can be paired with sparse/linear attention for long sequences.
>
>
> * **Q2. Low-rank Structure After Fine-tuning**
>
>
>   The rank structure of TSFMs after fine-tuning is certainly an interesting question. Since our analysis depends only on the low-dimension nature of the time-series co-domain, but not on any temporal structure, we naturally expect that the low-rank structure is preserved under fine-tuning. To verify this, we fine-tuned Moirai-1.0-R-base on the ETTh1 dataset for 100 epochs. Below are the averaged $\varepsilon$-ranks of the attention projection matrices:
>
>   | $\varepsilon$ | pretrained | fine-tuned |
>   |:-----------:|:----------:|:---------:|
>   | $10^{-0.5}$ | 13.167 | 19.250 |
>   | $10^{-1.0}$ | 145.250 | 141.417 |
>   | $10^{-1.5}$ | 318.250 | 284.833 |
>   | $10^{-2.0}$ | 436.833 | 378.750 |
>
>
>   We see that fine-tuning does not significantly increase the ranks of attention matrices in this example.
>
> * **Q3. Flow-of-ranks Beyond Chronos**
>
>   Yes, this is a good point. To see flow-of-ranks beyond Chronos, we have done some additional experiments on Moirai, WaveToken, and VisionTS. We observed flow-of-ranks in all of them. Results are summarized in Figure 6 of the revised manuscript, and a discussion can be found in our general response.

---

> ### Author Response · Authors · 2025-11-15
> **Official Comment by Authors (2/2)**
>
> * **Q4. Multivariate Scenario and Irregular Samples**
>
>
>    Yes, our results generalize to the multivariate case. First, note that section 2 is the only place where the univariate assumption is used. When the input is multivariate, one can simply view the channel axis as yet another patch dimension, so that Theorem 2 can still be applied. This holds as long as the number of variables is considerably smaller than the hidden dimension of the Transformer, which is indeed the case for most TSFMs. Our supplementary experiments on compressing Moirai also corroborate this. We have added a discussion in the footnote in Section 2.
>
>
>   Our results are sampling-agnostic since we are only concerned with the low-rank structure induced by the low-dimensional co-domain of a time series. In fact, our theorems concern only discrete inputs instead of continuous time series, which makes the results applicable to even irregularly sampled data.
>
>
> * **Q5. Low-rank Structure After Fine-tuning**
>
>
>   We propose a guideline that should work fine for most tasks. Let $D$ be the number of layers and $d$ the hidden size. Set the per-layer target rank
>   $$
>   r_\ell = r_{1}+(r_{D}-r_{1})\Big(\tfrac{\ell-1}{D-1}\Big)^{\alpha},\quad \ell=1,\dots,D,
>   $$
>   where we can set $r_{\min}$ = the median numerical rank of a small sample of input embeddings (or 16 if unknown), $r_{\max}=d/2$, and $\alpha = 0.5$. This design guarantees that $r_\ell$ grows smoothly and monotonically from $r_{\min}$ to $r_{\max}$ (concave in depth). This aligns capacity with the observed flow-of-ranks and keeps early layers compact. We have added this discussion to Appendix K of the revised paper.
>
> We are happy to answer any follow-up question(s) that the reviewer may have.

---

### Official Review · Reviewer_uH8K · 2025-10-30

**Soundness:** 3
**Presentation:** 3
**Contribution:** 2
**Rating:** 6
**Confidence:** 3

**Summary:**

The paper titled “UNDERSTANDING TRANSFORMERS FOR TIME SERIES: RANK STRUCTURE, FLOW-OF-RANKS, AND COMPRESSIBILITY” analyzes Transformer models for specifically time series (TSFMs). They show that in this scenario, the transformer models ( as a consequence of the data being passed to it) possess a uniquely low-rank structure compared to similar architectures for text or vision. They postulate thus due to the continuous nature of time-series embeddings. This low-rank input structure leads to that the Attention layer matrices being highly compressible. The authors introduce the concept of "flow-of-ranks," which describes how the numerical rank of a representation gradually increases with model depth due to nonlinear mixing, explaining why earlier layers are more amenable to compression. By leveraging these insights, the researchers demonstrate that large TSFMs like Chronos are severely over-parameterized and can be significantly compressed, achieving up to a 65% reduction in inference time and 81% in memory without losing predictive accuracy.

**Strengths:**

The low rank property of time-series may prove very useful in their application using transformers, in terms of designing models with fewer parameters. Theoretically, this low-rank property is proven for continuous embeddings, with guaranteed polynomial or exponential decay of singular values for smooth or analytic functions, respectively (Theorems 1 and 2).
The work provides the first general theoretical results (Theorem 3) connecting low-rank input embeddings to the compressibility of the internal Attention matrices (W_Q, W_K, W_V)
The paper introduces and quantifies the concept of "flow-of-ranks," which explains how  non-linear components (like activations, residual connections, and normalization) across deep layers gradually increase the rank of a representation (Theorem 4)

**Weaknesses:**

The majority of the empirical validation and compression experiments focus almost exclusively on the Chronos family of Time Series Foundation Models. While there are references to other models the core compression techniques and deep rank analysis (flow-of-ranks, impact of heads) are primarily demonstrated on Chronos. This limits the generality of the practical findings and compression results to other TSFM architectures
While the paper's core claims about rank structure are presented as a modality-dependent framework, the empirical evidence is often constrained to a small number of specialized experiments. How do we know this structure will prevail in other transformer architectures or other datasets.
The core theoretical analysis (Theorems 1 and 2) is derived for the univariate time series case. The authors mention in passing that this is extendible to few-variate time series but a detailed discussion is lacking

**Questions:**

na

---

> ### Author Response · Authors · 2025-11-15
>
> We thank the reviewer for the careful review and insightful comments.
>
>
> * **Compression of Models Beyond Chronos Family**
>
>
>    We show our results are general and hold on TSFMs outside of the Chronos family. We add an additional empirical evaluation on compressing Moirai-1.0-R-base. Please see the results in Table 3 of the revised manuscript or in our general response. Our results show that Moirai is also very compressible, supporting our claims on the compressibility of TSFMs with more generality. In addition, it provides an ablation of flow-of-ranks, which shows that a model compressed in a layer-dependent way performs significantly better than one whose layers are compressed in the same way.
>
>
> * **Low-rank Structures in Other Transformers/Datasets**
>
>    It is certainly interesting to test if the empirical claims made in our paper hold for other models. To show that the results hold for more TSFMs, we have done additional experiments in Figure 6 of the revised manuscript. We computed the ranks of three additional TSFMs that are fairly different from Chronos in both the architecture and the training dataset. Please find the summary of the results in our general response. Our results show that many TSFMs beyond the Chronos family enjoy low-rank embeddings and a compressible architecture. In addition, we do still see the flow-of-ranks in earlier layers.
>
>
> * **Generalizing Results to Multivariate case**
>
>
>    Yes, our results generalize to the multivariate case. First, note that section 2 is the only place where the univariate assumption is used. When the input is multivariate, one can simply view the channel axis as yet another patch dimension, so that Theorem 2 can still be applied. This holds as long as the number of variables is considerably smaller than the hidden dimension of the Transformer, which is indeed the case for most TSFMs. Our supplementary experiments on compressing Moirai also corroborate this. We have added a discussion in the footnote in Section 2.
>
> We are happy to answer any follow-up question(s) that the reviewer may have.

---

### Official Review · Reviewer_eiE9 · 2025-11-01

**Soundness:** 3
**Presentation:** 3
**Contribution:** 3
**Rating:** 4
**Confidence:** 2

**Summary:**

This paper investigates why Transformers trained on time series data behave differently from those trained on text or vision. The authors analyze the rank structure of embeddings and attention matrices to explain why time-series foundation models (TSFMs)—such as Chronos are highly compressible without losing much accuracy.
Several key points include: time series are naturally low-rank, and flow-of-rank perspective.

**Strengths:**

1. To my understanding, transformer theories in time series are generally lacking or provides limited practical guidance. This paper serves as a good entry point to understand how time series data differs from other modalities.
2. The results are intuitive to me, making this paper easy to follow.

**Weaknesses:**

1. This paper considers univariate time series. which is limited as several TSFMs can handle any-variate inputs.
2. The paper assumes input data X is 1-rank (or low-rank). I think this is a pretty strong assumption, which may not hold in many high-dimensional data.

**Questions:**

1. Can the authors explain footnote 1? Does it mean that if we have $n$-variate data, $x$ is then rank-$m$, where $n = m$? Is it possible that $m < n$?
2. I wonder how naive Thm 1 and Thm 2 are? Since this paper mainly shows existence proof, if the input data is low-rank, it seems like its straightforward that we can find low-rank weights to model it. Are there any counterexamples?
3. Are there any other features in the data assumption that makes it a time series? Or would the result hold for all low-rank input data?

Overall, I think this paper will be a good contribution to the field and am happy to adjust my score if the authors address my concerns?

---

> ### Author Response · Authors · 2025-11-15
>
> We thank the reviewer for their careful review and insightful comments.
>
> * **W1. Multivariate TSFMs**
>
>
>   Our theory is not limited to univariate TSFMs. In fact, our theory handles patched inputs, which are already multi-dimensional. To see this, note that our central argument in Section 2 is that the embedding $\boldsymbol{\Phi}: \mathbb{R}^k \rightarrow \mathbb{R}^d$ maps a $k$-dimensional patch space onto a low-dimensional submanifold $\boldsymbol{\Phi}(\mathbb{R}^k)$ in the hidden space because we usually have that $k \ll d$. For multivariate time series, the input embedding is $\boldsymbol{\Phi}: \mathbb{R}^{k \times c} \rightarrow \mathbb{R}^d$, where $c$ is the number of variates. As long as $k\cdot c \ll d$, the central argument still applies.
>
>
>   To empirically validate this argument, we have compressed Moirai-1.0-R-base, which is a multivariate TSFM. Please see the results in Table 3 of the revised manuscript or in our general response. Our results show that Moirai is also very compressible, supporting our claims on the compressibility of TSFMs with more generality. In addition, it provides an ablation of flow-of-ranks, which shows that a model compressed in a layer-dependent way performs significantly better than one whose layers are compressed in the same way.
>
>
> * **W2. Low-rank Inputs**
>
>
>   Suppose we have a multivariate time series $f: [0,T] \rightarrow \mathbb{R}^c$. The input of the entire TSFM is a sampled version of $f$, which we denote as $\mathbf{X}$. If no patching is used, then $\mathbf{X}$ is of shape $c \times L$, where $L$ is the number of samples we collected. If patching is used, then $\mathbf{X}$ is of shape $(c \cdot k) \times (L/k)$, where $k$ is the patch size. For several TSFMs, $c$ and $c \cdot k$ (which establish an upper bound of $\text{rank}(\mathbf{X})$) are typically much smaller than the hidden dimension. The low-rank property that we assumed for $\mathbf{X}$ is merely a product of its shape, and is reasonable.
>
>
>   For other data modalities, the input data is typically high-rank. This is what sets time series data apart from other modalities, and is why we propose in our paper a different design of TSFMs from, say, an LLM. We acknowledge that there are time series applications where the number of variates is comparable or greater than the hidden dimension, in which case the input matrix may not be low-rank. We thank the reviewer for suggesting this and have added this remark in Section 2 to improve the clarity.
>
>
> * **Q1. Footnote Clarification**
>
>
>   We have clarified the footnote in the revised version. In addition to our response to **W2**, we note that for a $c$-variate input, $\mathbf{X} \in \mathbb{R}^{c \times L}$ may indeed have a numerical rank smaller than $c$, but this depends on the specific data. Since we study foundation models rather than individual datasets, we assume only that $\text{rank}(\mathbf{X}) \leq c$ rather than $< c$, without making any data-dependent assumptions. The reason why the analysis directly applies to a $c$-variate TSFM is that $c$ can be viewed as another patch dimension. Since our analysis applies to any small patch size $k$, it also applies to any small number of variates $c$.
>
> * **Q2. Interpretation of Theorem 1 and 2**
>
>
>   To clarify: Theorems 1–2 do not assert “existence of low-rank weights.” They establish that when the embedding itself is low-rank by giving smoothness-based decay rates for the singular values of the embedded inputs, which we use to justify that TSFM inputs $\mathbf{U}$ are low-rank in the first place. This role is purely about the input geometry and is independent of model weights.
>
>
>   Theorem 3 then addresses the modeling question: given low-rank inputs, it provides a uniform, data-independent guarantee that attention can be approximated by low-rank matrices. The result is not naive, given that attention is non-linear in general. Importantly, Theorem 3 (part 2) also establishes a counter-example and proves a matching lower bound to show when such compression is impossible. Thus, beyond existence, we quantify how much rank reduction is justified and when it fails.
>
>
> * **Q3. Beyond Time Series Data**
>
>
>   This is a good point on the generality of our results beyond time series data, and you are correct that many theorems are not limited to the time series setting. If the input $\mathbf{U}$ into an attention layer is low-rank, then Theorem 3 and 4 are applicable and are independent of the data modality. Theorems 1-4 are purely numerical and do not explicitly depend on the time series input. Theorem 1 and 2, require that the input matrix be low-rank. For time series, this is guaranteed by the shape of the input matrix, but it is not for other high-rank data modalities, e.g., text and image (See Figure 2(a)).
>
> We are happy to answer any follow-up question(s) that the reviewer may have.

---

> > ### Comment · Reviewer_eiE9 · 2025-11-17
> >
> > Dear authors,
> >
> > Thank you very much for the detailed response.
> >
> > I think the authors addressed most of my concerns and I understand the paper a lot better now.
> > Personally, I still have two comments to the paper regarding authors response:
> > - It seems like the scope of this paper is larger than time series (except for the embedding part) and is applicable to high-dimensional data in general. I feel like a nice short discussion on this matter will enable wider application to the theories in this paper.
> > - While I do agree the authors response on Q2, I feel like low rank approximation of transformers/attention is being studied heavily in recent years, for example, [1, Appendix B] and it will be nice to include a short discussion on existing studies.
> >
> > However, overall I think the authors utilize the rebuttal session well, I recognize this paper's contribution in the deep learning community.
> > Thus, I'm willing to raise my rating and confidence score accordingly.
> >
> >
> >
> > [1] Scatterbrain: Unifying Sparse and Low-rank Attention Approximation [NeurIPS 2021]

---

> > > ### Author Response · Authors · 2025-11-18
> > >
> > > Dear Reviewer eiE9,
> > >
> > > Thank you for your careful review of our rebuttal and for the additional comments. We found them helpful and have made some further revisions:
> > >
> > > * We have highlighted that our analysis of the low-rank attention weight matrices is more general than a tool for time series. We explicitly mentioned this in the introduction (line 94-95) and also after Theorem 3 (line 308-310).
> > >
> > > * We have also added a comprehensive discussion of the development of low-rank approximation techniques for Transformers in our related work section (line 804-820).
> > >
> > > We are happy that you liked the paper, and hope that with these further edits, you would consider raising your rating, as mentioned. Please let us know if you have any other questions.

---

> > > > ### Author Response · Authors · 2025-11-25
> > > > **Official Comment by Authors**
> > > >
> > > > Dear Reviewer eiE9,
> > > >
> > > > We hope this message finds you well. We would like to kindly follow up on the rating update you mentioned in your post-rebuttal comments. We appreciate your thoughtful feedback and the time you have devoted to our paper. Please let us know if any additional clarification would be helpful. Thank you!
> > > >
> > > > Warm regards,
> > > >
> > > > Authors

---

> > > > > ### Comment · Reviewer_eiE9 · 2025-11-26
> > > > >
> > > > > Dear authors,
> > > > >
> > > > > I have increased my score to 6, good luck.

---

> > > > > > ### Author Response · Authors · 2025-11-26
> > > > > >
> > > > > > We truly appreciate your thoughtful feedback, which has strengthened the manuscript. Thank you very much for taking the time to reevaluate our submission.
> > > > > >
> > > > > > Authors

---

### Official Review · Reviewer_Qkww · 2025-11-01

**Soundness:** 3
**Presentation:** 3
**Contribution:** 3
**Rating:** 8
**Confidence:** 2

**Summary:**

This paper presents a rigorous linear-algebraic analysis of time series Transformer models. It first examines the ranking structure of time series embeddings, revealing unique low-rank characteristics that distinguish them from other modalities. The authors then theoretically infer the potential low-rank nature of the attention matrices in time series Transformers. Meantime, the authors introduce the concept of “flow-of-ranks” to describe how representation ranks evolve and increase across Transformer layers due to nonlinear transformations. Finally, leveraging these insights, the paper proposes two effective compression strategies for time series foundation models, achieving up to 65% reduction in inference time and 81% reduction in memory usage on the Chronos model, without compromising accuracy.

**Strengths:**

1. Strong Theoretical Grounding: Theorems 1 and 2 formally connect patch size and embedding smoothness to low-rank structure. Theorem 3 crucially links low-rank inputs to compressible attention layers, while Theorem 4 quantifies the "flow-of-ranks." These theories provide novel insights to guide time series foundation model design.

2.  The translation from theory to practice is seamless and compelling. Each theoretical claim is supported by corresponding empirical evidence, making the overall theory framework more convincing and practically relevant. And the results are striking, showing that TSFMs are significantly more over-parameterized than LLMs and can be compressed dramatically. The layer-dependent rank schedule is a simple yet powerful idea derived directly from the "flow-of-ranks."

3. Clarity and Organization: Despite the complex mathematical content, the paper is well-structured and readable. The flow from data structure to single-layer analysis to depth-dependent phenomena and finally to applications is logical and easy to follow.

**Weaknesses:**

1. The core experiments only focus on a single architecture family. All experiments are conducted on Chronos and Chronos-Bolt, which are based on the T5 architecture. While the principles are argued to be general, validation on other TSFM architectures (e.g., TimesFM, Moirai) would have further strengthened the claim of universality.

2. The paper provides elegant existence proofs—such as the low-rank properties of time series embeddings and the W_Q/K/V matrices—based on the core assumption that time series embeddings are intrinsically low-rank. However, this assumption may be an artifact of current TSFM design choices (e.g., small patch sizes and simple MLP-based embedding layers). As more tokenization methods emerge (e.g., VisionTS [1], Wavelet-based Tokenization [2]), it remains unclear whether these conclusions will still hold.

[1] Chen M, Shen L, Li Z, et al. Visionts: Visual masked autoencoders are free-lunch zero-shot time series forecasters[J]. arXiv preprint arXiv:2408.17253, 2024.

[2] Masserano L, Ansari A F, Han B, et al. Enhancing foundation models for time series forecasting via Wavelet-based tokenization[J]. arXiv preprint arXiv:2412.05244, 2024.

**Questions:**

1. Could the observed low-rank and compressibility properties be interpreted as universal characteristics of time series Transformer models? If so, does this imply that current architectures are inherently low-rank and may therefore lack sufficient expressiveness to capture more complex temporal dynamics?

2. The success of pre-trained compressed model suggests that standard TSFMs are severely over-parameterized. Does this imply that the common practice of scaling up model size for time series is misguided ?  Is the low rank of TSFMs a inherant feature, or is it a sign that we are not yet challenging them with tasks of sufficient complexity?

3. Theorem 3 suggests that the attention matrix may be less compressible when dealing with more complex or higher-rank input time series (e.g., noisy financial data). Are there datasets of this nature that could be used to empirically test the boundaries of the proposed low-rank assumption?

---

> ### Author Response · Authors · 2025-11-15
>
> We thank the reviewer for their careful review and insightful comments.
>
> * **W1. Experiments on additional TSFMs**
>
>
>   We show additional empirical evaluation on compressing a TSFM outside the Chronos family, i.e., Moirai-1.0-R-base. Please see the results in Table 3 of the revised manuscript and in our general response above. Our results show that Moirai is also very compressible, supporting our claims on the compressibility of TSFMs with more generality. In addition, it provides an ablation of flow-of-ranks, which shows that a model compressed in a layer-dependent way performs significantly better than one whose layers are compressed in the same way.
>
>
> * **W2. Other tokenization methods**
>
>
>   Yes, the compressibility of most TSFMs comes from the intrinsically low-dimensionality of the input domain of the embedding function. To show that the results hold for more TSFMs, we have run additional experiments in Figure 6 of the revised manuscript to show how our analysis applies to various embedding strategies, including those used in WaveToken and VisionTS. We computed the ranks of Moirai, WaveToken, and VisionTS: they are all fairly different from Chronos in both the architecture and the training dataset. Please find the summary of the results in our general response. Our results show that while most TSFMs enjoy low-rank embeddings and a compressible architecture, exceptions like VisionTS exist. In addition, we do still see the flow-of-ranks in earlier layers.
>
> * **Q1. Interpretation of Low-rank Structures**
>
>
>   The observed low-rank structure of attention matrices in TSFMs arises naturally from uni- or few-variate time-series inputs. This finding primarily indicates that the early Transformer layers are larger than necessary for low-rank inputs. It does not suggest that the TSFM architecture lacks expressivity. In fact, our identified flow-of-ranks phenomenon shows that the rank of the hidden representations increases across layers. This shows how the model transforms a low-dimensional complex problem into a higher-dimensional and more linearly separable one. This is analogous to the Koopman operator, which lifts nonlinear dynamics into a high-dimensional space where they become quasi-linear. We have added a brief discussion of this analogy in Section 4 of the revised manuscript.
>
>
> * **Q2. Design of TSFMs**
>
>
>   Yes, one finding from our work is that TSFMs, unlike foundation models for other data modalities, are easily overparameterized (at least in their first few layers). As Theorem 3 shows, the first few layers of a Transformer in a TSFM provably need low ranks only. Flow-of-ranks shows that later layers can handle higher-rank inputs. Therefore, it is not totally hopeless to scale up TSFMs to handle more complicated tasks, but our work shows that it should be done in a layer-dependent manner to avoid severe overparameterization.
>
>
> * **Q3. Data-dependent Rank Analysis**
>
>
>   Your question points out a subtle distinction between the temporal domain of a time series and the domain of the input embedding of a TSFM that we will clarify. You are correct that if a time series $f$ has a simple temporal structure (e.g., a constant function), then its embedding is low-rank. Our paper shows that the embedding of any time series input is low-rank, regardless of its temporal structure.
>
>   To see this, we view a time series as a function $f: [0,T] \rightarrow \mathbb{R}^c$, where $c$ is the number of variates, then an embedding layer is a function $\boldsymbol{\Phi}: \mathbb{R}^{c \times k} \rightarrow \mathbb{R}^{d}$, where $k$ is the patch size and $d$ is the Transformer's hidden dimension. Theorem 3 shows that if $\boldsymbol{\Phi}(\mathbb{R}^{c \times k})$ is low-dimensional, then attention matrices need only be low-rank. Note that the assumption of $\boldsymbol{\Phi}(\mathbb{R}^{c \times k})$ being low-rank is only an artifact of the low-dimensionality of $\mathbb{R}^{c \times k}$, and is universally true for any $f$ independent of its temporal structures. We have added a clarification on this to Section 3.
>
> We are happy to answer any follow-up question(s) that the reviewer may have.

---

### Public Comment · ~Peiwang_Tang1 · 2025-11-14
**A Question about Theorem 2**

Consider the input embedding defined by a two-layer residual MLP. For any $\varepsilon > 0$, the formula indicates that the maximum rank is $(1 + \varepsilon^{-2})k$. Since $\varepsilon$ is usually a very small positive number (typically less than 1), it seems that when $\varepsilon$ is less than $\sqrt{k/(d - k)}, (1 + \varepsilon^{-2})k$ is much larger than $d$. For example, when $k = 16$ and $d = 768$, $\varepsilon$ only needs to be less than $0.1458$ for this to hold (Apparently, $\varepsilon\$ is usually smaller than 0.1458). I wonder if I might be missing something here, as it seems counterintuitive that the $\varepsilon$-numerical rank of the MLP embedding is constrained by the patch size $k$ rather than the hidden dimension $d$ in such cases. Could you kindly shed some light on how to reconcile this apparent discrepancy? I would truly appreciate your insights.

---

> ### Author Response · Authors · 2025-11-15
>
> Dear Peiwang,
>
>
> Thank you for the great question! You are right that the bound can look vacuous if one plugs in a very small tolerance $\varepsilon$ together with a moderate width $d$: since it upper-bounds the $\varepsilon$-rank by $(1+\varepsilon^{-2})k$, whenever $(1+\varepsilon^{-2})k>d$ the statement effectively reduces to the trivial cap $\leq d$. We are happy to clarify the following points:
>
>
> * **How to read the bound.** The theorem is best viewed in a dimension-free, asymptotic sense: fix the patch size $k$ and a numerically meaningful tolerance $\varepsilon$, then let the width $d$ grow. The conclusion is that the number of singular values above the threshold is $\mathcal{O}(k)$ (i.e., independent of $d$), so making the model arbitrarily wide is unnecessary for representing embedded patches at that tolerance.
>
>
> * **Worst-case analysis.** The proof gives a worst-case upper bound. Typical embeddings do not saturate it; empirically, spectra decay much faster and effective ranks remain well below both $d$ and $(1+\varepsilon^{-2})k$. See Figure 3. The key takeaway is the linear dependence on $k$ and the lack of growth with $d$.
>
>
> * **On the apparent "vacuity."** A precise reading of the equation in Theorem 2 is
>   $$
>     \text{LHS} \leq \min\{\,d,\ (1+\varepsilon^{-2})k\,\}.
>   $$
>   Thus, for extremely small $\varepsilon$, the minimum is capped by $d$; this does not contradict the theorem, it reflects the nature of $\varepsilon$-rank at very fine tolerances.
>
>
> * **Stable rank (tolerance-free view).** If one prefers a tolerance-free notion, the stable rank $\\|X\\|_F^2/\\|X\\|_2^2$ enjoys a sharper analysis here. The same argument underlying our proof in Appendix C yields a stable-rank bound proportional to $k$ (and again independent of $d$), aligning with the intuition that a residual MLP lifts a $k$-dimensional patch space without creating $d$ effective degrees of freedom at the top scale.
>
>
> We appreciate the question and have added clarifications to Section 2.

---

### Author Response · Authors · 2025-11-15
**General Response**

We appreciate the reviewers' valuable feedback to help improve the clarity and generality of our manuscript. We are glad that all the reviewers found our work mathematically rigorous and an important contribution to understanding TSFMs. To summarize:
 * **Reviewer Qkww** acknowledges that our "theories provide novel insights to guide time series foundation model design and that the translation from theory to practice is seamless and compelling."
 * **Reviewer eiE9** highlights that our paper is "easy to follow" with "intuitive results" that serve "as a good entry point to understand how time series data differs from other modalities".
 * **Reviewer uH8K** acknowledges that our work "provides the first general theoretical results (Theorem 3) connecting low-rank input embeddings to the compressibility of the internal Attention matrices" and "introduces and quantifies the concept of "flow-of-ranks."
 * **Reviewer u1Ge** importantly highlights that our "findings have real design implications for TSFMs".

We provide a common response below, and address minor, reviewer-specific comments in our individual reviewer responses. We have also updated the manuscript with the changes highlighted in red in the rebuttal version.


## **Applicability to TSFMs beyond the Chronos family**
 We have added additional experiments on compressing Moirai-1.0-R-base in Table 3, which we also show in the below table:

 | Size Ratio | $\tilde{d}\_0$ | $\alpha$ | WQL | MASE | $\tilde{d}\_0$ | $\alpha$ | WQL | MASE |
 |:-----------:|:----------:|:---------:|:------:|:-------:|:-------------:|:---------:|:------:|:-------:|
 | | **w/ flow-of-ranks**  |  |  |  | **w/o flow-of-ranks** |  |  |  |
 | 0.250 | 8 | 0.34 | 1.001 | 1.014 | 16 | 0.00 | 1.069 | 1.050 |
 | 0.500 | 10 | 0.58 | 0.996 | 1.007 | 32 | 0.00 | 1.038 | 1.036 |
 | 1.000 | – | – | – | – | 64 | 0.00 | 1.000 | 1.000 |

 For Moirai-1.0-R-base, we show that one can also compress the attention matrices by at least $75%$ percent without affecting the performance. The results show that our method is applicable more broadly than just to Chronos and Chronos-Bolt, and is more generally observed on other TSFMs, e.g., Moirai. The results also show an ablation of the flow-of-ranks design proposed in our paper: given the same compression ratio, a model compressed with flow-of-ranks performs significantly better than a model that does not.


## **Flow-of-Ranks holds on Various TSFMs**
We have done some additional experiments that reveal the ranks of attention matrices in more TSFMs, including Moirai, WaveToken, and VisionTS in Figure 6. Our results reveal the following:

  * Moirai, which uses a patch-based input embedding still has low-rank structures. There is also a clear flow-of-rank pattern in the weight matrices throughout layers.

  * WaveToken, which uses a wavelet transform to decompose the context into different frequency-modes, inputs a time series (i.e., the concatenation of wavelets in the time domain) rather than the wavelet coefficients. The input domain is a one-dimensional space, and is quantized for embedding. We also observe low-rank structures. (See Figure 2(a)) (More empirical analysis can also be found in Appendix H.)

  * VisionTS first transforms a time series into an image and operates on the patches. Since the patch space usually has a higher dimension, we observe that the attention matrices are closer to full-rank than those in other TSFMs.

  These results show that many TSFMs indeed have low-rank structures in their attention matrices, and our flow-of-ranks is a general phenomenon.


## **Multivariate TSFMs**
We have revised footnote 1 to indicate how the analysis in our paper seamlessly applies to multivariate TSFMs. Note that our central argument in Section 2 is that the embedding $\boldsymbol{\Phi}: \mathbb{R}^k \rightarrow \mathbb{R}^d$ maps a $k$-dimensional patch space onto a low-dimensional submanifold $\boldsymbol{\Phi}(\mathbb{R}^k)$ in the hidden space because we usually have that $k \ll d$. For multivariate time series, the input embedding is $\boldsymbol{\Phi}: \mathbb{R}^{k \times c} \rightarrow \mathbb{R}^d$, where $c$ is the number of variates. As long as $k\cdot c \ll d$, the central argument still applies.


## **Autocorrelation-agnostic**
We added clarifications in Sections 3 and 4 by separating the temporal domain from the embedding domain and showing that the low-rank assumption follows from the low-dimensional patch embedding, which is independent of temporal structure, stationarity, or any specific autocorrelation pattern.

---

### Meta-Review · Area_Chair_CKAQ · 2025-12-25

**Summary:**

All reviewers acknowledge the contributions of this paper. For most of the questions, the authors' responses seemed reasonable.

**Reviewer Concerns:**

Reviewer eiE9 responded to the authors' rebuttal and raised the score accordingly. The other reviewers did not respond, but for most of the questions, the authors' responses seemed reasonable. However, I believe there is a key issue that requires further clarification: The majority of the empirical validation and compression experiments focus almost exclusively on the Chronos family of Time Series Foundation Models (TSFM). Even though the authors conducted further validation on the Moirai-1.0-R-base model, this is far from sufficient. TSFM is in a very early stage; existing models are essentially unusable, and many factors influence performance. Exploring its quantification at this point is premature, and the conclusions may even be meaningless.

**Reviewer Scores:**

The initial scores from reviewers Qkww, uH8K, and u1Ge are relatively high, and they all acknowledge the contributions of this paper. It is unlikely that the scores will be raised further.

---

### Decision · Program_Chairs · 2026-01-26

Accept (Poster)